# Near-Optimal Private Linear Regression via Iterative Hessian Mixing

**Omri Lev** [1]  **Moshe Shenfeld** [2]  **Vishwak Srinivasan** [1]  **Katrina Ligett** [2]  **Ashia C. Wilson** [1]

## Abstract

We study differentially private ordinary least squares (DP-OLS) with bounded data $(X, Y)$ via sketching-based mechanisms. While Gaussian sketching approaches have been explored for DP-OLS (Sheffet, 2017), they are typically viewed as less competitive than the Adaptive Sufficient Statistics Perturbation (AdaSSP) method (Wang, 2018), which directly perturbs the sufficient statistics $(X^\top X, X^\top Y)$. This method was shown to be close to information-theoretically optimal, while also exhibiting strong empirical performance. In this work, we propose *Iterative Hessian Mixing* (IHM), an algorithm that builds on Gaussian-sketching approaches to DP-OLS and is inspired by the Iterative Hessian Sketch (Pilanci & Wainwright, 2016). We prove that IHM is differentially private and provide utility guarantees in the form of excess empirical risk bounds. These bounds improve upon those of AdaSSP by removing a multiplicative factor that can be as large as the square root of the data dimension. The design of the IHM is based on new accuracy guarantees that we present for prior Gaussian sketching approaches for DP-OLS, which clarify when these methods are expected to perform well and how IHM circumvents their inherent limitations. We also conduct a rigorous empirical evaluation on a large suite of datasets, demonstrating that IHM consistently outperforms prior baselines, including AdaSSP.[1]

## 1. Introduction

Machine learning models trained on personal data are now ubiquitous, making it increasingly important to safeguard the privacy of individuals whose data contribute to these systems. Differential privacy (DP) (Dwork et al., 2006) has emerged as the standard for privacy-preserving analysis, providing a formal guarantee that the inclusion or exclusion of any individual in the training data has only a limited effect on the model's output distribution. DP has been adopted in practice by major technology platforms and statistical agencies (Erlingsson et al., 2014; Apple Research, 2017; Ding et al., 2017; Facebook Research, 2020; Snap Security, 2022; Ponomareva et al., 2023), demonstrating its feasibility at scale. Nevertheless, enforcing DP introduces both computational and statistical challenges. Stronger privacy guarantees typically degrade model accuracy, creating a fundamental trade-off between privacy and utility. This tension motivates the search for computationally efficient, differentially private, and statistically accurate algorithms. In this paper, we study the fundamental statistical problem of *ordinary least squares* (OLS), where the goal is to estimate a linear predictor that best fits covariates $X_i \in \mathbb{R}^d$ to responses $Y_i \in \mathbb{R}$. Mathematically, given data $\{(X_i, Y_i)\}_{i=1}^n$, this is achieved by generating the regressor $\theta^*$

$$L(\theta) := \|Y - X\theta\|_2^2, \qquad \theta^* = \underset{\theta \in \mathbb{R}^d}{\mathrm{argmin}}\, L(\theta). \quad (1)$$

When $X$ has full rank, this solution has the closed form $\theta^* = (X^\top X)^{-1} X^\top Y$. We study the differentially private version of this problem (reffered as *DP-OLS*), which involves finding $\widehat{\theta}$ that solves (1) subject to the constraint that the algorithm producing $\widehat{\theta}$ needs to satisfy formal privacy definitions with respect to $(X, Y)$.

Whenever there are known bounds on $\|x_i\|$ and $|y_i|$, some solutions for the DP-OLS problem work by publishing a private version of the sufficient statistics $(X^\top X,\ X^\top Y)$ and use them to calculate $\theta^*$ (Vu & Slavkovic, 2009; Dwork et al., 2014b; Foulds et al., 2016; Wang, 2018; Tang et al., 2024; Ferrando & Sheldon, 2025). Amongst these, AdaSSP (Wang, 2018), which adds an adaptive regularization to these private sufficient statistics to avoid invertibility issues, stands out for its strong empirical performance and theoretical guarantees; it matches the information-theoretic lower bounds up to logarithmic terms and thus is regarded as the

[1]Department of EECS, Massachusetts Institute of Technology, US [2]School of Computer Science and Engineering, The Hebrew University of Jerusalem, IL. Correspondence to: Omri Lev <omrilev@mit.edu>.

*Proceedings of the 43rd International Conference on Machine Learning*, Seoul, South Korea. PMLR 306, 2026. Copyright 2026 by the author(s).

[1]Code available at https://github.com/omrilev1/HessianMix.

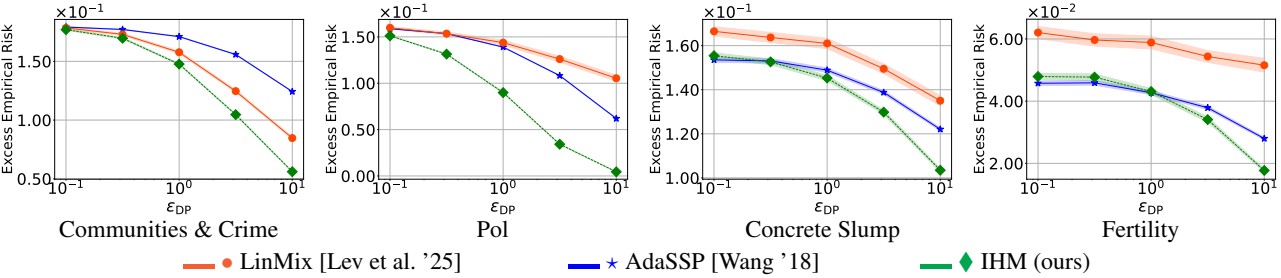

*Figure 1.* Comparison of IHM with two DP-OLS baselines: AdaSSP and the previous methods that rely on Gaussian sketches, on four representative datasets. We describe the baseline algorithms in more detail in Section 4, and the simulation details in Section 5.

leading baseline for the DP-OLS problem in the setup where $(X, Y)$ are fixed and where we have known bounds on $\|x_i\|$ and $|y_i|$ (Liu et al., 2022; 2023; Brown et al., 2024b). However, as presented in (Wang, 2018) and as we discuss in the later parts of this work, the upper bounds on the performance of AdaSSP involve terms that, although logarithmic, can be large due to the presence of quantities that may be exponentially small. In particular, the bounds incur a multiplicative factor $\log(1/\varrho)$ where $\varrho$ is the failure probability of the algorithm. Thus, whenever the failure probability is kept exponentially small in problem-related quantities, these factors might incur a prohibitive slack relative to the lower bounds.

Other strategies for solving the DP-OLS problem have been studied in the past; see Sheffet (2017; 2019); Varshney et al. (2022); Amin et al. (2023); Brown et al. (2024a) for example. Of particular relevance to this paper are the works of Sheffet (2017; 2019) that posit using mechanisms that rely on left-multiplication of the data $(X, Y)$ by a random Gaussian matrix (also known as a *Gaussian sketch*) as a viable solution for the DP-OLS problem. Mathematically, these methods first generate the noisy sketched data $(SX + \eta\xi, SY + \eta\zeta)$ for i.i.d. Gaussian matrices $S, \xi$ and $\zeta$ and for a calibrated $\eta$ chosen for ensuring $(\varepsilon, \delta)$-DP, and then use these variables for solving the OLS problem. We note that beyond the DP-OLS setting, related sketching ideas in the context of linear regression have also been used in alternative settings such as federated learning and distributed computing (Karakus et al., 2017; Prakash et al., 2020; Sun et al., 2022; Anand et al., 2021). Recent work by Lev et al. (2025) gives stronger privacy guarantees for a mechanism based on Gaussian sketches, which consequently led to improved estimation errors in comparison to Sheffet (2017), and which in several cases are superior relative to AdaSSP. However, it remains unclear for which settings sketching-based solutions can be used as a replacement for the AdaSSP approach, and what are their error guarantees in comparison to the bounds presented by Wang (2018).

## 1.1. Our Contributions

In this work, we introduce a new algorithm for DP-OLS, called *Iterative Hessian Mixing* (IHM) (Algorithm 1). This method adapts Gaussian sketching and is inspired by the iterative Hessian sketch algorithm of Pilanci & Wainwright (2016), which, in the context of linear regression, works by sketching only inside the Gram matrix $X^\top X$, and then involving these sketches in an iterative algorithm that produces a sequence of estimates $\{\widehat{\theta}_t\}$ whose distance to $\theta^*$ shrinks geometrically with the iteration count $t$. Our method provides a differential privacy counterpart of this classical method, and we show that in the context of DP, sketching only inside $X^\top X$ allows for substantially reducing the level of the additive noise in certain cases, and further avoiding error amplification, which usually occurs when sketching $Y$. We provide a theoretical analysis of this algorithm, showing (1) DP guarantees and (2) its excess empirical risk guarantees in the setting where the dataset $(X, Y)$ is fixed, deriving bounds that parallel the excess empirical risk bounds derived by Wang (2018). We then empirically demonstrate its performance on the same collection of linear regression datasets used by Wang (2018), showing that it consistently outperforms the previous methods of Sheffet (2017); Wang (2018); Lev et al. (2025) (see Figure 1, Figure 2, and extended discussion in Section 5). Additionally, we provide a new accuracy analysis of the previously existing methods that rely on Gaussian sketches, which can be applied in multiple settings currently uses the Gaussian sketch. By doing this, we characterize the limitations of these previous methods, showing when they are expected to outperform and underperform the AdaSSP baseline. Then, we show that in most regimes IHM has improved guarantees in comparison to both AdaSSP and to the Linear Mixing procedure from Lev et al. (2025) (which is an improved version of the method of Sheffet (2019)).

**Organization.** In Section 2.1 and Section 2.2 we provide background about DP and DP-OLS. In Section 3 we introduce the Iterative Hessian Mixing (IHM), its theoretical properties, and the intuition behind it. Utility guarantees for existing DP-OLS methods appear in Section 4.1 and Sec-

tion 4.2; in Section 4.3, we compare these utility guarantees to those of IHM. We demonstrate the empirical performance of IHM in Section 5, and conclude with discussion of avenues for future work in Section 6.

## 2. Background and Problem Setup

**Notation.** Random variables are in sans-serif (e.g., $\mathsf{X}, \mathsf{y}$), and their realizations in serif (e.g., $X, y$). We denote $\{1, 2, \ldots, n\} := [n]$. The $\ell_2$ norm of $v \in \mathbb{R}^d$ is $\|v\|$. For a positive semi-definite matrix $M$, $\|v\|_M$ is the weighted norm $\sqrt{\langle v, Mv \rangle}$. The all-zeros column vector of length $d$ is $\vec{0}_d$. The $k \times k$ identity matrix is $\mathbb{I}_k$ and $\mathcal{N}(0, \mathbb{I}_{k_1 \times k_2})$ denotes a $k_1 \times k_2$ matrix of i.i.d. standard Gaussian entries. Throughout, we refer to $\theta^*$ as the OLS fit from (1) and to $\theta^*(v)$ as the Ridge regressor $\left(X^\top X + v\mathbb{I}_d\right)^{-1} X^\top Y$. Given a matrix $X$, we denote $\lambda_{\min}^X := \lambda_{\min}(X^\top X)$ and $\lambda_{\max}^X := \lambda_{\max}(X^\top X)$. Given vector $Y$, we denote $\lambda_{\min}^{XY} := \lambda_{\min}((X, Y)^\top(X, Y))$. The per-coordinate C-level clipping operation applied over a vector $\overline{v}$ is denoted by $\mathrm{clip}_{\mathsf{C}}(\overline{v})$. Specifically, for each entry $v_i$ in $\overline{v}$ its clipped version is $\mathrm{clip}_{\mathsf{C}}(v_i) = v_i \cdot \min\{1, \mathsf{C}/|v_i|\}$. The notation $\widetilde{O}(\cdot)$ hides factors that are logarithmic in quantities appearing polynomially in the bound. In Appendix A, we give additional notation.

### 2.1. Differential Privacy

A differentially private mechanism, roughly speaking, ensures that the distributions of the outputs returned for *neighboring datasets* are similar. Datasets considered in this paper are assumed to be elements of $\mathbb{R}^{n \times d}$ i.e., matrices with $n$, $d$-dimensional real-valued rows. Two datasets $X, X'$ are called *neighbors* if $X'$ is formed by replacing a row in $X$ with $\vec{0}_d$ or vice-versa – this is referred to as *zero-out* neighboring where the removal of a row is analogous to replacing it with $\vec{0}_d$, and we use $X \simeq X'$ to denote this relation. With this setup in place, we now define the notion of differential privacy used in this work.

**Definition 1** (($\varepsilon, \delta$)-DP (Dwork et al., 2006))**.** A randomized mechanism $\mathcal{M}$ is said to satisfy ($\varepsilon, \delta$)-*differential privacy* if for all $X, X'$ such that $X' \simeq X$ and all measurable subsets $\mathcal{S} \subseteq \mathrm{Range}(\mathcal{M})$,

$$\mathbb{P}(\mathcal{M}(X) \in \mathcal{S}) \leq e^\varepsilon \cdot \mathbb{P}(\mathcal{M}(X') \in \mathcal{S}) + \delta.$$

($\varepsilon, \delta$)-DP satisfies key properties such as maintaining its guarantees under post-processing and graceful degradation under composition. In particular, the post-processing property ensures that if a mechanism $\mathcal{M}$ satisfies this privacy definition, then so does $g \circ \mathcal{M}$ for any (possibly randomized) function $g$ (Dwork et al., 2014a).

### 2.2. Problem Setup: DP-OLS

Recall that the focus of this work is DP-OLS, where the (non-private) OLS problem is defined by (1). Given a dataset $(X, Y)$ where $X \in \mathbb{R}^{n \times d}$ and $Y \in \mathbb{R}^n$, our goal is to calculate its private variant, which is defined as the problem of estimating a differentially-private linear predictor from this data, according to the ($\varepsilon, \delta$)-DP definition. In this work, we place the following concrete assumptions[2] on the data:

- ($\mathbf{A}_1$) *Bounded domain:* $\|x_i\| \leq \mathsf{C}_X$ and $|y_i| \leq \mathsf{C}_Y$ for all $i \in [n]$.

- ($\mathbf{A}_2$) *Overdetermined system:* $n \geq d$.

We assume knowledge of the values $\mathsf{C}_X$ and $\mathsf{C}_Y$ as done in several prior analyses of linear regression (both in the private and non-private settings, see Shamir (2015); Sheffet (2017; 2019); Wang (2018)). Throughout, we normalize the covariates $X$ so that $\mathsf{C}_X = 1$ without loss of generality.[3] Following standard baselines (Bassily et al., 2014; Wang, 2018), our accuracy measure is the *excess empirical risk*, which, given a dataset $(X, Y)$ defined by the difference $\mathcal{R}(X, Y, \theta) := L(\theta) - L(\theta^*)$ where $L(\theta)$ and $\theta^*$ were defined in (1). In the rest of this work, we assume that $(X, Y)$ is fixed, and omit it from $\mathcal{R}$ unless specified otherwise.

---

**Algorithm 1** Iterative Hessian Mixing (IHM)

---

**Input:** Dataset: $X \in \mathbb{R}^{n \times d}, Y \in \mathbb{R}^n$; noise parameters: $(\gamma, \tau, \sigma)$; hyperparameters $k \in \mathbb{N}, T \in \mathbb{N}$ and $\mathsf{C} \in \mathbb{R}_+$;
1: Initialize $\widehat{\theta}_0 \leftarrow \vec{0}_d$.
2: Set $\eta \leftarrow \texttt{CalibrateMixingNoise}\left(X, 1, \gamma, \tau, \frac{\gamma}{\sqrt{k}}\right)$ (Algorithm 2).
3: **for** $t = 0, \ldots, T-1$ **do**
4:    Sample $\mathsf{S}_t \sim \mathcal{N}(0, \mathbb{I}_{k \times n}), \xi_t \sim \mathcal{N}(0, \mathbb{I}_{k \times d}), \zeta_t \sim \mathcal{N}(\vec{0}_d, \mathbb{I}_d)$.
5:    Calculate $\widetilde{X}_t = \mathsf{S}_t X + \eta \xi_t$.
6:    Calculate $\widetilde{G}_t = X^\top \mathrm{clip}_{\mathsf{C}}(Y - X\widehat{\theta}_t) - \eta^2 \widehat{\theta}_t + \sigma\zeta_t$.
7:    Update $\widehat{\theta}_{t+1} = \widehat{\theta}_t + \left(\frac{1}{k}\widetilde{X}_t^\top \widetilde{X}_t\right)^{-1} \widetilde{G}_t$.
8: **Output:** $\widehat{\theta}_T$

---

## 3. Our Method: Iterative Hessian Mixing

In this section, we introduce our proposed method, IHM, which is presented in Algorithm 1. As mentioned in Section 1, our method is inspired by the iterative Hessian sketch (IHS) of Pilanci & Wainwright (2016) to solve (1). The IHS

---

[2]From a privacy perspective, rather than assuming these properties, they can be enforced by clipping the input data.

[3]This is without loss of generality as for $X' = \frac{X}{\mathsf{C}_X}$ and $\theta' = \mathsf{C}_X\theta$, we have $X\theta = X'\theta'$. Thus, any guarantee proved under $\mathsf{C}_X = 1$ extends to general $\mathsf{C}_X$ by applying the inverse rescaling.

generates a sequence $\{\widehat{\theta}_t\}_{t\geq 0}$ that satisfies the recursion

$$\widehat{\theta}_0 = \vec{0}_d, \tag{2}$$

$$\widehat{\theta}_{t+1} = \underset{\theta \in \Upsilon}{\arg\min}\left\{\frac{1}{2k}\left\|\mathsf{S}_t X(\theta - \widehat{\theta}_t)\right\|^2 - \theta^\top (X^\top (Y - X\widehat{\theta}_t))\right\}$$

where $\mathsf{S}_t \sim \mathcal{N}(0, \mathbb{I}_{k \times n})$ is sampled independently at each iteration and $\Upsilon$ is a convex subset of $\mathbb{R}^d$. The differences between Algorithm 1 and (2) results in IHM being DP:

1. The additive noise $\eta\xi_t$ added to the sketched data $\mathsf{S}_t X$ was proved to provide DP for the $\eta$ calculated inside the algorithm (Sheffet, 2017; Lev et al., 2025);

2. The calculation of $\widetilde{G}_t$ involves clipping and additive Gaussian noise, which provides DP as in classical adjustments done when making classical optimizers private (for example, as done in the private version of SGD, DP-SGD (Abadi et al., 2016)).

Our main theorem, formalized below, states that for any $k, \mathsf{C}$ and $T$, there exist choices of $\gamma, \sigma, \tau$ that result in IHM being $(\varepsilon, \delta)$-DP with respect to $(X, Y)$. Additionally, we present high-probability bounds on the excess empirical risk $\mathcal{R}$ that hold for a specific choice of $k, T$, and $\mathsf{C}$. As we show later, our bounds improve over the currently known state-of-the-art guarantees established in (Wang, 2018) by a factor that can be as large as $\sqrt{d}$.

**Theorem 1.** *Let $\widehat{\theta}$ be the output of Algorithm 1. Then, there exist noise parameters $(\gamma, \sigma, \tau)$ such that:*

1. *Algorithm 1 is $(\varepsilon, \delta)$-DP with respect to $(X, Y)$.*

2. *Let $\varrho \in (0, 1]$ be a target failure probability and let*

$$g^X := \left(\frac{\varepsilon d}{\sqrt{\log(1/\delta)}} \cdot \left(\lambda^X_{\max} - \lambda^X_{\min}\right)\right)^{2/3},$$

$$\gamma_{\mathrm{h}} := \frac{\sqrt{\max\left\{d, \log\left(1/\varrho\right), g^X\right\}\log(1/\delta)}}{\varepsilon},$$

*and $\mathsf{C}^2_{\mathrm{h}} := \mathsf{C}^2_Y + \|\theta^*\|^2$. Then, there exists $k, T, \mathsf{C}$ such that with probability at least $1 - \varrho$*

$$\mathcal{R}\big(\widehat{\theta}\big) \leq \widetilde{O}\left(\gamma_{\mathrm{h}} \cdot \mathsf{C}^2_{\mathrm{h}} \cdot \min\left\{1, \frac{\gamma_{\mathrm{h}}}{\lambda^X_{\min}}\right\}\right). \tag{3}$$

As we show in Section 4.3, the bound (3) has a similar structure to the guarantees of the currently existing schemes. However, in many cases it improves them in its dependence on the failure probability $\varrho$ and the dimension $d$.

**Proof Outline for Theorem 1.** The privacy guarantee is due the algorithm performing $2T + 1$ calls to individual private mechanisms: one call for computing $\eta$ (Line

2), $T$ Gaussian sketching mechanism applied over the extended matrix $[X^\top, \eta\mathbb{I}_d]^\top$ in Line 5, and $T$ calls to the Gaussian mechanism in Line 6. The privacy guarantees of each of these steps are given in Lemma 1 and Lemma 2 in Appendix C. Composing these yields the stated privacy guarantee (Dwork et al., 2014a, Thm. 3.16).

The accuracy guarantee relies on the accuracy analysis of the IHS (Pilanci & Wainwright, 2016) which we introduce first. Expanding the OLS objective as

$$\frac{1}{2}\|Y - X\theta\|^2 = \frac{1}{2}\|Y\|^2 + \frac{1}{2}\|X\theta\|^2 - Y^\top X\theta,$$

IHS applies a Gaussian sketch to $X$ inside $\|X\theta\|^2$, resulting in $\|\mathsf{S}X\theta\|^2$, and then solves the minimization problem with this new objective. As shown in Pilanci & Wainwright (2016), the excess empirical risk of this solution can be bounded simply in terms of $\|X\theta^*\|$ instead of $L(\theta^*)$; notably the former is small when $\|\theta^*\|$ is small. The new sketched objective can be iteratively minimized, analogous to Newton's method, and this is precisely given in (2). Performing this $T$ iterations, and with specific choices of the sketching dimension $k$ yields the following high-probability guarantee on $\mathcal{R}$

$$\mathcal{R}\big(\widehat{\theta}_T\big) \leq \chi^{2T} \|X\theta^*\|^2$$

for $\chi \in (0, 1]$ that decreases with an increase in $k$. This is appealing since this guarantee decays geometrically with $T$. To operationalize this in proving Theorem 1, we note that our iterations are similar to those presented in (2), with $X$ replaced by the concatenated matrix $[X^\top, \eta\mathbb{I}_d]^\top$. Before using Pilanci & Wainwright (2016, Theorem 2) to obtain a bound on the excess empirical error, we control the effect of the two key differences to the IHS setup highlighted previously: the additive Gaussian noise $\sigma\zeta_t$ and the clipping operation. For the former, we use concentration guarantees to suitably bound the contribution from the additive Gaussian noise $\sigma\zeta_t$ with high probability for a careful choice of $k$. For the latter, we also demonstrate that the clipping operation does not occur with high probability. The associated high probability events are detailed in Definition 2, along with lemmas showing that they occur with high probability. The complete proof along with explicit choices of the parameters $(\gamma, \sigma, \tau, k, \mathsf{C}, T)$ is given in Appendix C.

**Remark 1.** In private setup, the fact that the IHS applies the sketch only to $X$ provides an additional accuracy improvement: since the privacy guarantees of the mechanism that rely on the Gaussian-sketch improve with the minimal eigenvalue of its input matrix (Lemma 2 and extended discussion in Lev et al. (2025)), applying the sketch only to $X$ lets us to substantially reduce the amount of noise in cases where $X$ is well-conditioned. As we show later, this is in contrast to the previous methods that rely on Gaussian

sketching for DP, where the sketch is applied to the concatenation $(X, Y)$, making the noise dependent on $\lambda_{\min}^{XY}$, which is rarely large in practical settings. Evidently, (3) is monotonically decreasing in $\lambda_{\min}^X$.

### 3.1. Choosing Parameters for IHM

The guarantee (3) was established for a specific choice of hyperparameters, which usually requires knowledge of data-dependent quantities like $g^X$ and $\|\theta^*\|$. Here we explain how this should be done in practice.

1. **Sketch Size $k$.** Our analysis relies on invoking classical results from the sketching literature, which formulate constraints on the sketch size $k$ for their guarantees to hold. Following these, for any number of iterations $T$ the sketch size required to satisfy $k \geq a_1 \cdot \max\left\{d, \log(a_2 T/\varrho)\right\}$ for some universal constants $a_1, a_2$. In practice, the actual factors $a_1$ and $a_2$ can be chosen using similar global rules used in the non-private Hessian sketch literature (see, e.g., Pilanci & Wainwright (2016, Section 3.1)). Additionally, for preventing clipping throughout the iterations, we further have the condition $k \geq a_3 \cdot g^X$ for some universal constant $a_3$. We note that this condition is active whenever $\varepsilon\left(\lambda_{\max}^X - \lambda_{\min}^X\right) \geq \sqrt{d \log(1/\delta)}$. We note that this quantity requires knowledge of the eigenvalues $\lambda_{\min}^X$ and $\lambda_{\max}^X$. Since our algorithm already calculates a private version of $\lambda_{\min}^X$ (inside the function `CalibrateMixingNoise`), $g^X$ can be privately estimated by calculating a private estimate of $\lambda_{\max}^X$, whose sensitivity is $\mathsf{C}_X^2$. Practically, the condition on $g^X$ is usually satisfied by the other conditions placed on $k$, and we choose $k$ based on the choices made in (Pilanci & Wainwright, 2016).

2. **Number of Iterations $T$.** Our proof strategy suggests that the parameter $T$ should be set to $\log\left(\max\left\{1, \frac{\lambda_{\min}^X}{\gamma_h}\right\}\right)$ up to constant factors. In the worst-case, $\lambda_{\min}^X = \frac{n}{d}$, and the optimal $T$ grows at most like $O\left(\log\left(\frac{n\varepsilon}{\max\{d, \log(1/\varrho)\}\log(1/\delta)}\right)\right)$. Similarly to $k$, the actual $T$ can be chosen based on private estimations of $\lambda_{\max}^X$ and $\lambda_{\min}^X$. However, in many typical settings and practical ranges of $\varepsilon$, this quantity is bounded from above by a constant, and, as we show in Section 5, a constant choice of $T$ usually suffices for close-to-optimal performance on both regimes of $\lambda_{\min}^X$.

3. **Clipping Level $\mathsf{C}$.** For preventing clipping, $\mathsf{C}$ needs to satisfy $\mathsf{C} > a_4 \cdot \max\left\{\mathsf{C}_Y, \|\theta^*\|\right\}$ for some universal constant $a_4$. In practice, we typically lack bounds on $\|\theta^*\|$. However, since $\|Y\|_\infty$ is bounded, it is reasonable to assume that there exists a finite $B = O(1)$ such that $\mathsf{C} = B\mathsf{C}_Y$ ensures no clipping.

As we demonstrate via our experiments, both $k, \mathsf{C}$ and $T$ can be set globally and do not require data-dependent tuning of the kind often needed for methods such as DP-SGD, even in synthetic datasets chosen deliberately to have large $g^X$ and $\lambda_{\min}^X$ (see Appendix J and Appendix K).

## 4. Comparison Between Methods for DP-OLS

Equipped with Theorem 1, we now compare the utility guarantees of different existing approaches for this DP-OLS setting. Specifically, we compare AdaSSP with the LinMix method of Lev et al. (2025) (Algorithm 5 in Appendix G) that uses the Gaussian sketches for DP-OLS and with IHM (Algorithm 1). As we will show, IHM retains the benefits of both schemes, providing guarantees that are superior to these prior baselines.

### 4.1. Adaptive Sufficient Statistics Perturbation

The leading baseline for DP-OLS with fixed and bounded $(X, Y)$ is AdaSSP (see Algorithm 6 in Appendix G) which functionally can be expressed as:

$$\widehat{\theta}_a = \left(X^\top X + \eta\xi + \widehat{\lambda}\mathbb{I}_d\right)^{-1}\left(X^\top Y + \sigma\zeta\right)$$

with $\eta = \Theta\left(\frac{\sqrt{\log(1/\delta)}}{\varepsilon}\right), \sigma = \Theta\left(\frac{\sqrt{\log(1/\delta)}\mathsf{C}_Y}{\varepsilon}\right)$, $\zeta$ being a $d$-dimensional Gaussian vector with i.i.d. standard Gaussian entries and $\xi$ being a symmetric matrix whose upper triangular elements are i.i.d. standard Gaussian. The role of $\eta\xi$ and $\sigma\zeta$ is to ensure that $X^\top X$ and $X^\top Y$ are kept private. The parameter $\widehat{\lambda}$ can essentially be set to $0$ (Dwork et al., 2014b), though in practice it was observed that these methods can benefit from a regularization added to avoid non-invertible cases, and AdaSSP exploits this. This approach is viable for DP-OLS due to the following reasons: (a) under $\mathbf{A}_1$, the worst-case $\ell_2$ change in the quantities $X^\top X$ and $X^\top Y$ when evaluated on neighboring datasets can be uniformly bounded, and (b) using their private versions in (1) yields a solution that is private by post-processing. The utility guarantees of AdaSSP are in the next proposition.

**Proposition 1.** *(Wang, 2018, Appendix. B) Let $\varrho \in (0, 1]$ be a target failure probability and let $\gamma_a := \frac{\sqrt{d \log(1/\varrho) \log(1/\delta)}}{\varepsilon}$ and $\mathsf{C}_a^2 := \mathsf{C}_Y^2 + \|\theta^*\|^2$. Then, w.p. at least $1 - \varrho$ the output of the AdaSSP algorithm $\widehat{\theta}_a$ satisfies*

$$\mathcal{R}\left(\widehat{\theta}_a\right) \leq \widetilde{O}\left(\gamma_a \cdot \mathsf{C}_a^2 \cdot \min\left\{1, \frac{\gamma_a}{\lambda_{\min}^X}\right\}\right) \qquad (4)$$

### 4.2. DP-OLS via Gaussian Sketches

The application of Gaussian sketches in DP-OLS was initiated by Sheffet (2017; 2019). In optimization, randomized sketching is used to compress large quadratic objectives

while approximately preserving their structure, enabling fast and provably accurate approximations (Mahoney et al., 2011; Woodruff, 2014; Pilanci & Wainwright, 2015). In DP-OLS, these approaches rely on solutions of the form

$$\widehat{\theta}_{\text{Sketch}} = (\widetilde{X}^\top \widetilde{X})^{-1} \widetilde{X}^\top \widetilde{Y} \; ; \quad \begin{cases} \widetilde{X} = \mathsf{S}X + \eta\xi \\ \widetilde{Y} = \mathsf{S}Y + \eta\zeta \end{cases} \quad (5)$$

where $\mathsf{S} \sim \mathcal{N}(0, \mathbb{I}_{k \times n}), \xi \sim \mathcal{N}(0, \mathbb{I}_{k \times d}), \zeta \sim \mathcal{N}(\vec{0}_k, \mathbb{I}_k)$ and $\eta$ set to ensure that the sketched data $(\widetilde{X}, \widetilde{Y}) := (\mathsf{S}X + \eta\xi, \mathsf{S}Y + \eta\zeta)$ is $(\varepsilon, \delta)$-DP (Sheffet, 2017; 2019; Lev et al., 2025). We note that $\widehat{\theta}_{\text{Sketch}}$ relies on the quantities $(\mathsf{S}X + \eta\xi, \mathsf{S}Y + \eta\zeta)$, for which the privacy guarantees improves whenever the minimal eigenvalue $\lambda_{\min}^{XY}$ grows (Sheffet, 2019; Lev et al., 2025). Excess empirical risk guarantees for $\widehat{\theta}_{\text{Sketch}}$ can be derived by noting that for the non-private but sketched formulation of the OLS problem below

$$\widehat{\theta} := \underset{\theta}{\operatorname{argmin}} \; \|\mathsf{S}Y - \mathsf{S}X\theta\|^2$$

there exists choice of $k$ such that for $\mathsf{S} \sim \mathcal{N}(0, \mathbb{I}_{k \times n})$

$$\mathcal{R}(\widehat{\theta}) \leq (2\chi + \chi^2) L(\theta^*) \quad (6)$$

as proved by Pilanci & Wainwright (2015, Theorem 1). Note that (6) scales linearly with $L(\theta^*)$, which in some cases could be large (e.g., due to low signal-to-noise ratio). As we show in Theorem 2, $\widehat{\theta}_{\text{Sketch}}$ suffers from a similar dependence in $L(\theta^*)$ and which might limits its performance.

While Lev et al. (2025) assesses the Gaussian sketching mechanism for DP-OLS empirically, it remains unclear when they outperform AdaSSP or match its theoretical guarantees. The next theorem presents a new utility analysis that highlights their tradeoffs relative to AdaSSP. It concerns a variant of LinMix algorithm of Lev et al. (2025, Algorithm 2) which we state explicitly as Algorithm 5 in Appendix G. Starting from a prescribed sketching dimension $k$, the algorithm privately computes the required noise level based on $\lambda_{\min}^{XY}$. When no noise is needed, it internally enlarges the sketch size up to the maximum value compatible with the privacy constraints, without adding additional noise. This yields improved performance when $\lambda_{\min}^{XY} \gg 1$.

**Theorem 2.** *Let $\widehat{\theta}_{\text{m}}$ be the output of Algorithm 5 and let $\varrho \leq \delta$ be a target failure probability. Let $\gamma_{\text{m}} := \frac{\sqrt{\max\{d, \log(1/\varrho)\} \log(1/\delta)}}{\varepsilon}$ and $\mathsf{C}_{\text{m}}^2 := (1 + \mathsf{C}_Y^2)(1 + \|\theta^*\|^2)$. Then, there exists $k$ such that w.p. at least $1 - \varrho$*

$$\mathcal{R}(\widehat{\theta}_{\text{dmix}}) \leq \begin{cases} O\left(\left(\gamma_{\text{m}} - \frac{\lambda_{\min}^{XY}}{1 + \mathsf{C}_Y^2}\right) \mathsf{C}_{\text{m}}^2 + L(\theta^*)\right) & \text{if } \lambda_{\min}^{XY} < \frac{\gamma_{\text{m}} \mathsf{C}_{\text{m}}^2}{1 + \|\theta^*\|^2} \\ O\left(\frac{\gamma_{\text{m}} \mathsf{C}_{\text{m}}^2 L(\theta^*)}{\lambda_{\min}^{XY}(1 + \|\theta^*\|^2)}\right) & \text{otherwise.} \end{cases}$$

$$(7)$$

**Proof Sketch** Using (6) for (5) similarly to the way it applied in (Lev et al., 2025) yields

$$\mathcal{R}(\widehat{\theta}_{\text{m}}) \leq (1 + \chi)^2 \left(\mathcal{R}(\theta^*(\eta^2)) + \eta^2 \|\theta^*(\eta^2)\|^2\right) + (\chi^2 + 2\chi) \cdot (\eta^2 + L(\theta^*))$$

for any $(k, \chi)$ such that (6) holds and for any $\eta \geq 0$ that is defined in the functional solution template (5). Then, the proof proceeds by substituting the values of $\chi$ and $\eta$ derived in the different cases and bounding $\mathcal{R}(\theta^*(\eta^2))$ and $\|\theta^*(\eta^2)\|^2$ via classical arguments for the excess empirical risk and solution norm difference between the Ridge and the OLS fits; see Appendix D.

Although Theorem 2 is stated for Algorithm 5, its close structural similarity to the algorithm of Sheffet (2017) means that analogous conclusions apply to these as well, except that whenever $\lambda_{\min}^{XY} \geq \frac{\gamma_{\text{m}} \mathsf{C}_{\text{m}}^2}{1 + \|\theta^*\|^2}$ their excess error saturates at $\sqrt{\frac{\max\{c_0 d, c_2^{-1} \log(2c_1/\varrho)\}}{k}} \cdot L(\theta^*)$.

### 4.3. Comparison of Guarantees

On a high level, we identify four qualitative aspects in which the guarantees can be compared, and which are listed below.

**Leading Multiplier.** Since $\frac{\gamma_{\text{a}}}{\gamma_{\text{m}}} = \sqrt{\min\{d, \log(1/\varrho)\}}$ (up to $\widetilde{O}$ terms), we have $\gamma_{\text{m}} \leq \gamma_{\text{a}}$, giving the sketching-based approach a smaller leading multiplier. Moreover, since $\frac{\gamma_{\text{h}}}{\gamma_{\text{m}}} = \sqrt{\max\left\{1, \frac{g^X}{\max\{d, \log(1/\varrho)\}}\right\}}$ (up to $\widetilde{O}$ terms), this improvement is retained by IHM whenever $g^X = O(\max\{d, \log(1/\varrho)\})$. Thus, whenever $g^X$ is small, this multiplier retains the sketching benefits and improves the multiplier relative to AdaSSP. Empirically, in many typical cases $g^X$ is smaller than $\max\{d, \log(1/\varrho)\}$, so IHM usually retains the same benefit as linear mixing, giving an improvement that can be as large as $\sqrt{d}$.

**Residual Sensitivity.** We note that (7) contains a term proportional to $L(\theta^*)$. This term might behave as $\Theta(n)$ and dominate the bound. As we prove in Appendix F, $\frac{L(\theta^*)}{\lambda_{\min}^{XY}(1 + \|\theta^*\|^2)} \geq 1$. Thus, it is irreducible, even when $\lambda_{\min}^{XY} \gg 1$. On the contrary, the asymptotic bound (3) does not contain any additional term. The closed-form (non-asymptotic) bound contains the term $\left(\sqrt{2c_0 d/k}\right)^T \|\theta^*(\eta)\|_{X^\top X + \eta^2 \mathbb{I}_d}^2$, that decays geometrically in $T$ whenever $k > 2c_0 d$ and which further decreases by the noise level $\eta$. Practically, in most typical ranges of $\varepsilon$, it can be made sufficiently small after a few iterations.

**Eigenvalue Dependence.** The bound (7) is monotonically decreasing in $\lambda_{\min}^{XY}$, in contrast to (4) which decreases with

| Method | Multiplier $\gamma$ | Scale Term C | Eigenvalue Term | Additional Term |
|--------|---------------------|--------------|-----------------|-----------------|
| AdaSSP | $\frac{\sqrt{d \log(1/\varrho) \log(1/\delta)}}{\varepsilon}$ | $\|\theta^*\|^2 + C_Y^2$ | $\frac{\gamma_a}{\lambda_{\min}^X}$ | $0$ |
| LinMix | $\frac{\sqrt{\max\{d, \log(1/\varrho)\} \log(1/\delta)}}{\varepsilon}$ | $(1 + \|\theta^*\|^2)(1 + C_Y^2)$ | $\frac{L(\theta^*)}{\lambda_{\min}^{XY}(1+\|\theta^*\|^2)}$ | $L(\theta^*) - \lambda_{\min}^{XY}\left(1 + \|\theta^*\|^2\right)$ |
| IHM | $\frac{\sqrt{\max\{d, \log(1/\varrho), g^X\} \log(1/\delta)}}{\varepsilon}$ | $\|\theta^*\|^2 + C_Y^2$ | $\frac{\gamma_h}{\lambda_{\min}^X}$ | $0$ |

*Table 1.* Comparison of guarantees. The constants $\gamma_h, \gamma_a$ and $g^X$ were defined in Theorem 1 and Proposition 1 and are given by $\gamma_a = \frac{\sqrt{d \log(1/\varrho) \log(1/\delta)}}{\varepsilon}, \gamma_h = \frac{\sqrt{\max\{d, \log(1/\varrho), g^X\} \log(1/\delta)}}{\varepsilon}$ and $g^X = \left(\frac{\varepsilon d}{\sqrt{\log(1/\delta)}} \cdot \left(\lambda_{\max}^X - \lambda_{\min}^X\right)\right)^{2/3}$.

$\lambda_{\min}^X$. As we show in Appendix F, $\lambda_{\min}^{XY} \leq \lambda_{\min}^X$. Thus, this leads to a benefit for AdaSSP, especially in instances in which $\lambda_{\min}^X \gg \lambda_{\min}^{XY}$. However, (3) decays with $\lambda_{\min}^X$, thus enjoys similar performance improvement as AdaSSP for well-conditioned designs $X$.

**Scale.** The scale of (7) is $C_m^2 = (1 + C_Y^2)(1 + \|\theta^*\|^2)$. Comparing to (4), $C_a^2$ increased by $1 + C_Y^2 \|\theta^*\|^2$. Then, this increase is problematic in two regimes: whenever $C_Y^2 \|\theta^*\|^2 \ll 1$ it holds that $C_m^2 \approx 1 + C_a^2 \geq C_a^2$. Additionally, whenever $C_Y^2 \|\theta^*\|^2 \gg 1$ then $C_m^2 \approx C_a^2 + C_Y^2 \|\theta^*\|^2$. On the other hand, $C_h^2 = C_a^2$, giving similar behavior for IHM and AdaSSP.

Table. 1 summarizes these differences. Comparing LinMix and AdaSSP, we note that whenever $L(\theta^*) = O(\gamma_m C_m^2)$, $C_Y^2 \|\theta^*\|^2 = \Theta\left(\max\{C_Y^2, \|\theta^*\|^2\}\right)$ and $\lambda_{\min}^X \ll 1$ (so $\lambda_{\min}^{XY} \leq \lambda_{\min}^X \ll 1 + C_Y^2$), LinMix enjoys from an improved guarantee due to the avoidance of the $\log(1/\varrho)$ term in $\gamma_m$, and is expected to outperform AdaSSP, although it have a slightly worse scale. Comparing this to IHM, under the previously mentioned constraints on $g^X$, IHM enjoys the benefits of both worlds and provides the benefits of both schemes. As we show in Section 5, it usually outperforms both baselines.

**Remark 2.** The guarantee (7) was established by using a specific value of $k$, and under the assumption that we only have access to a private version of $\lambda_{\min}^{XY}$. However, one might ask if this can be improved via an optimized (possibly data-dependent) choice of $k$, and a more refined way to exploit $\lambda_{\min}^{XY}$. In Appendix E we show that similar behavior and conclusions hold even under a (non-realistic) data-dependent choice of $k$ and exact access to $\lambda_{\min}^{XY}$; thus, these are indeed fundamental drawbacks of the current methods rather than artifacts of a suboptimal choice of parameters. Moreover, note that this constant choice was made so that the leading term in the guarantee improves over that of AdaSSP. As explained, it further avoids the term $g^X$ appearing in $\gamma_h$. We note that even under the optimized data-dependent choice from Appendix E, the leading multiplier does not improve asymptotically. In particular, in the favor-

able regime for LinMix, namely, $L(\theta^*) = O(\gamma_m C_m^2)$ and $\lambda_{\min}^X \ll 1$, the asymptotic guarantee remains unchanged.

## 5. Empirical Results

We evaluate IHM on the 33 linear regression datasets gathered from real-world settings used in (Wang, 2018) and two synthetic datasets to compare three methods: AdaSSP (Wang, 2018), LinMix (Lev et al., 2025), and IHM. The parameters chosen for each of the schemes are in Table. 5.

| Algorithm | Parameters |
|-----------|------------|
| LinMix | $k = 2.5 \cdot \max\{d, \log(2/\varrho)\}$ |
| | $k = 6 \cdot \max\{d, \log(4T/\varrho)\}$ |
| IHM | $T = 3$ |
| | $C = C_Y$ |

For IHM, the factor 6 was picked based on the choice of sketch size from (Pilanci & Wainwright, 2016, Section 3.1). We further used $\varrho = \delta/10$ where $\delta = 1/n^2$. For IHM, $\gamma$ was calculated analytically via the calculation from (Lev et al., 2025) (also described in line 1 of Algorithm 4 in Appendix G) with sketch size $kT$ and target of $(\varepsilon/2, \delta/2)$. The parameter $\sigma$ was calculated analytically based on the exact formula for the Gaussian mechanism from (Balle & Wang, 2018), for a target of $(\varepsilon/2, \delta/4)$ after composition over $T$ steps. For AdaSSP, we further used the analytic formula from (Balle & Wang, 2018) to calculate the amount of noise needed for $\lambda_{\min}^X$, for $X^\top X$ and for $X^\top Y$, with $(\varepsilon/3, \delta/3)$ for each of these. Additional experimental details are given in Appendix H. In Appendix J, we provide experiments for (a) dataset in which $\lambda_{\max} - \lambda_{\min}$ grows with $n$, and (b) dataset satisfying $\lambda_{\min}^X = n/d$, the largest attainable value; in this case, the optimal number of iterations from Appendix C.1.2 (hidden inside the $\widetilde{O}$ terms of (3)) is maximized. For both cases, IHM provides similar gains over the alternatives with similar hyperparameters. The simulations presented in Figure 1 and Figure 2 span the four cases discussed in Section 4: two datasets with a large residual $L(\theta^*)$ (the Pol and Protein datasets); two datasets with a large $\lambda_{\min}^X$ and a significantly smaller $\lambda_{\min}^{XY}$ (the Concrete Slump and the Servo datasets); two datasets in which $\|\theta^*\|^2 \ll 1$ (the Fertility and the Pen-

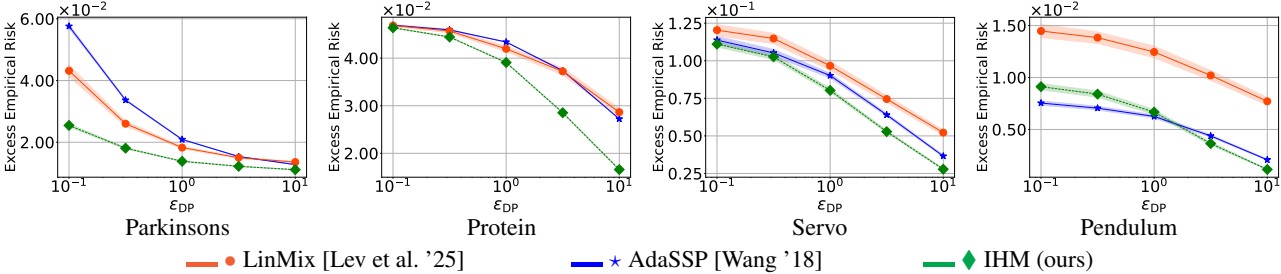

*Figure 2.* Performance of the IHM compared with AdaSSP (Wang, 2018) and with the Gaussian-sketch–based method of (Lev et al., 2025) across four representative datasets. Columns here and in Figure 1 are grouped by dataset characteristics: (1) datasets favorable to prior sketching methods (Sheffet, 2017; Lev et al., 2025); (2) datasets with large residuals; (3) datasets with non-negligible $\lambda_{\min}^X$; and (4) datasets where $\|\theta^*\| \ll 1$, in which classical sketching methods incurs an unfavorable scale term. Across these datasets—and consistently over the full experimental suite (Appendix I–Appendix L)— IHM matches or outperforms all baselines.

dulum datasets), and another two datasets in which none of these settings hold, correspond to a regime which is in favor of the linear mixing algorithm (the Crime and the Parkinsons datasets). The key parameters that characterize the different datasets are in Table. 2. We report the normalized (divided by $n$) excess empirical risk and 95% confidence intervals across $\varepsilon \in [0.1, 10]$, and which were chosen as this represents a typical range of operation. In all cases $\delta = 1/n^2$ and $\varrho = \delta/10$. Across selected datasets (see Figure 1 and Figure 2) and in the full suite (Appendix I-Appendix K), IHM consistently outperforms or matches the alternatives, and LinMix outperforms AdaSSP only when the favorable conditions in Section 4 hold simultaneously. Moreover, beyond certain cases (discussed in Appendix L and Appendix K), $T = 3, k = 6 \cdot \max\{d, \log(4T/\varrho)\}$ and $C = C_Y$ achieve a consistent improvement over the alternatives.

## 6. Discussion and Future Work

We developed a new algorithm for DP-OLS, which relies on Gaussian sketches. We compared the performance of this algorithm to that of two classical methods: AdaSSP (Wang, 2018) and the previous methods that rely on Gaussian sketches (Sheffet, 2017). This was enabled due to a new utility analysis we provide for the methods from Sheffet (2017), which clarifies their fundamental disadvantages. The IHM avoids the major drawbacks of these prior approaches and, as we show, provides both theoretical and empirical gains across a wide range of regimes and across diverse real-world datasets.

This work opens several directions for future research. First, we note that (3) contains the $g^X$ term and the $\widetilde{O}$ notation hides factors which might be large in practice. Developing techniques that perform direct accuracy analysis in the presence of clipping or reduce the excess factors (for example, via different allocation of $\varepsilon$ across the iterations) could improve the performance of IHM. Second, IHM can be thought of as a private version of the Newton sketch

algorithm (Pilanci & Wainwright, 2017) applied over the quadratic loss $\|Y - X\theta\|^2$. Developing a private version of the Newton sketch algorithm, which can be applied to general models, is another potential future research direction, currently under investigation. Finally, replacing Gaussian sketches with structured random projections that support fast matrix–vector multiplication while preserving differential privacy could further improve computational efficiency, following the line of Pilanci & Wainwright (2015; 2016). Exploring such structured sketches and adapting both the iterative Hessian mixing framework and the prior linear mixing framework from Sheffet (2017) to these is the topic of ongoing work.

## Acknowledgements

OL thanks Martin Wainwright for drawing his attention to the iterative Hessian sketch and for several stimulating discussions about this work. OL and AW were supported in part by the Simons Foundation Collaboration on Algorithmic Fairness under Award SFI-MPS-TAF-0008529-14, and VS and AW were supported in part by Assicurazioni Generali S.p.a. through MIT Award 036189-00006. MS and KL were supported in part by ERC grant 101125913, Simons Foundation Collaboration 733792, Israel Science Foundation (ISF) grant 2861/20, Apple, and a grant from the Israeli Council of Higher Education. Views and opinions expressed are, however, those of the author(s) only and do not necessarily reflect those of the European Union or the European Research Council Executive Agency. Neither the European Union nor the granting authority can be held responsible for them.

## Impact Statement

Differential privacy is increasingly adopted in sensitive domains such as healthcare and finance. Differentially private linear regression is a common component in analytics pipelines in these settings. Our work improves methods for

this task while providing formal privacy guarantees. Deployment decisions should account for domain constraints, regulatory obligations, and potential downstream harms, and should include careful validation, monitoring, and stakeholder oversight.

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

## A. Additional notation

The set of positive integer numbers $\{1, 2, 3, \ldots, \}$ is denoted by $\mathbb{N}$. The operator norm of a matrix $A$ is denoted by $\|A\|_{\mathrm{op}}$ and is defined by $\sup_{v:\|v\|=1} \|Av\|$. The trace of a matrix $A$ is denoted by $\mathrm{Tr}(A)$, and its minimal and maximal eigenvalues by $\lambda_{\min}(A)$ and $\lambda_{\max}(A)$ respectively. The minimal and maximal eigenvalues of its Gram matrix are denoted by $\lambda_{\min}^A := \lambda_{\min}(A^\top A)$ and $\lambda_{\max}^A := \lambda_{\max}(A^\top A)$. We use the notation $A \succeq 0$ and $A \succ 0$ for denoting situations in which $A$ is Positive Semi-Definite (PSD) and in which $A$ is Positive Definite (PD). For $v \in \mathbb{R}^d$, its $\ell_\infty$ norm is $\|v\|_\infty$. The Loewner order is defined in the usual way, where $B \preceq A$ denotes $A - B \succeq 0$. We usually denote our dataset $\{x_i\}_{i=1}^n$ where each $x_i \in \mathbb{R}^d$ in the matrix form $X = (x_1, \ldots, x_n)^\top$. We write $y(x) = O(x)$ if $\exists c > 0, x_0 > 0$ such that $|y(x)| \leq cx$ for all $x \geq x_0$. We write $y(x) = o(x)$ if $\lim_{x \to \infty} \frac{y(x)}{x} = 0$. We write $y(x) = \Omega(x)$ if $\exists c > 0, x_0 > 0$ such that $y(x) \geq cx$ for all $x \geq x_0$. We write $y(x) = \Theta(x)$ if $y(x) = O(x)$ and $y(x) = \Omega(x)$.

## B. Differential Privacy via Gaussian Sketches

The tightest characterization of the privacy guarantees obtained by using the Gaussian sketch was established by Lev et al. (2025), and was derived by analyzing them via Rényi-DP (Mironov, 2017). Algorithm 3 and Proposition 2 provide a characterization of a mechanism that uses these sketches for providing differential privacy and their privacy guarantees.

**Proposition 2** ((Lev et al., 2025)). *For any $\delta \in (0, 1)$ and any input matrix $X \in \mathbb{R}^{n \times d}$ such that $\|x_i\| \leq 1$ for all $i \in [n]$ the output of Algorithm 3 with parameters $k \geq 1$, $\gamma > {}^5\!/{}_2$, $\eta > 0$ and $\tau \geq \sqrt{2\log({}^3\!/{}_\delta)}$ satisfies $(\widetilde{\varepsilon}(\eta, \gamma, k, \delta), \delta)$-DP, where*

$$\widetilde{\varepsilon}(\eta, \gamma, k, \delta) = \frac{\sqrt{2\log(3.75/\delta)}}{\eta} + \min_{1 < \alpha < \gamma} \left\{ \varphi(\alpha; k, \gamma) + \frac{\log(3/\delta) + (\alpha - 1)\log(1 - {}^1\!/{}_\alpha) - \log(\alpha)}{\alpha - 1} \right\} \tag{8}$$

*and where* $\varphi(\alpha; k, \zeta) := \frac{k\alpha}{2(\alpha-1)} \log\left(1 - \frac{1}{\zeta}\right) - \frac{k}{2(\alpha-1)} \log\left(1 - \frac{\alpha}{\zeta}\right)$.

As shown in (Lev et al., 2025), by setting $\eta = \frac{\gamma}{\sqrt{k}}$ we get that Algorithm 3 is $(\varepsilon, \delta)$-DP with

$$\varepsilon \leq \frac{k}{2\gamma^2} + \frac{2\sqrt{2k\log(4/\delta)}}{\gamma}. \tag{9}$$

## C. Proof of Theorem 1

The proof of the theorem relies on a collection of intermediate lemmas, which we present first. The proofs of these lemmas are provided in Appendix C.2; skip to Appendix C.1 for the main proof of the theorem. For the differential privacy guarantee, we use the notion of Rényi Differential Privacy (RDP) (Mironov, 2017) and its conversion to $(\varepsilon, \delta)$-DP (Canonne et al., 2020). The key mechanisms we use are the Gaussian Mixing (Lev et al., 2025) and the Gaussian mechanisms (Dwork et al., 2014a) whose RDP guarantees are presented first. The lemmas are presented in the following order: privacy-related lemmas, general algebraic lemmas, and, finally, lemmas related to the theorem's accuracy result.

For $A \in \mathbb{R}^{n \times d}$, $b \in \mathbb{R}^n$, and $a > 0$, the following definitions will be used throughout:

$$\theta^*(A, b) := (A^\top A)^{-1} A^\top b, \qquad \theta^*(A, b; a) := (A^\top A + a\mathbb{I}_d)^{-1} A^\top b.$$

**Lemma 1** (Mironov (2017, Corollary 3)). *The mechanism $\mathcal{M}(X) = f(X) + \sigma\xi$ for $f : \mathcal{X} \to \mathbb{R}^m$ and for a general domain $\mathcal{X}$ such that $\sup_{X \simeq X'} \|f(X) - f(X')\| \leq b$ and $\xi \sim \mathcal{N}(\vec{0}_m, \mathbb{I}_m)$ is $\left(\alpha, \frac{\alpha b^2}{2\sigma^2}\right)$-RDP for any $\alpha > 1$.*

**Lemma 2** (Lev et al. (2025, Lemma 1)). *Let $X \in \mathbb{R}^{n \times d}$ be such that $\max_{i \in [n]} \|x_i\|^2 \leq 1$ and assume that $\lambda_{\min}(X^\top X) \geq \overline{\lambda}_{\min}$. Then, the mechanism $\mathcal{M}(X) = \mathsf{S}X + \sigma\xi$ where $\mathsf{S} \sim \mathcal{N}(0, \mathbb{I}_{k \times n})$ and $\xi \sim \mathcal{N}(0, \mathbb{I}_{k \times d})$ is $(\alpha, \varphi(\alpha; k, \gamma))$-RDP for any $\alpha \in (1, \gamma)$ where $\gamma = \sigma^2 + \overline{\lambda}_{\min}$ and where*

$$\varphi(\alpha; k, \zeta) := \frac{k\alpha}{2(\alpha-1)} \log\left(1 - \frac{1}{\zeta}\right) - \frac{k}{2(\alpha-1)} \log\left(1 - \frac{\alpha}{\zeta}\right).$$

**Lemma 3** (Canonne et al. (2020, Proposition 12)). *If $\mathcal{M}$ satisfies $(\alpha, \varepsilon(\alpha))$-RDP, then it also satisfies $(\varepsilon_{\mathrm{DP}}, \delta)$-DP for any $0 < \delta < 1$, where $\varepsilon_{\mathrm{DP}} = \varepsilon(\alpha) + \log\left(1 - \frac{1}{\alpha}\right) - \frac{\log(\alpha\delta)}{(\alpha-1)}$.*

**Lemma 4.** *(Wang, 2018, Lemma. 5) For any $\theta \in \mathbb{R}^d$, $A \in \mathbb{R}^{n \times d}$ and $b \in \mathbb{R}^n$ it holds that*

$$\|b - A\theta\|^2 - \|b - A\theta^*(A, b)\| = \|\theta - \theta^*(A, b)\|_{A^\top A}^2.$$

**Lemma 5.** *For any $a \geq 0$, $A \in \mathbb{R}^{n \times d}$ and $b \in \mathbb{R}^n$ we have $\|\theta^*(A, b; a) - \theta^*(A, b)\|_{A^\top A + a\mathbb{I}_d}^2 \leq a\|\theta^*(A, b)\|^2$.*

**Lemma 6.** *Let $A \in \mathbb{R}^{n \times d}$, $B \in \mathbb{R}^{k \times n}$ such that $A^\top B^\top BA \succ 0$ and let $b \in \mathbb{R}^n$, $v \in \mathbb{R}^d$, $u \in \mathbb{R}^d$, $k \in \mathbb{R}$ and*

$$\widehat{\theta}(A, b, B, v, u) := v + \frac{1}{k}(A^\top B^\top BA)^{-1}\left(A^\top(b - Av) + u\right).$$

*Then,*

$$\widehat{\theta}(A, b, B, v, u) = \underset{\theta}{\operatorname{argmin}}\left\{\frac{1}{2k}\|BA(\theta - v)\|^2 - \theta^\top\left(A^\top(b - Av) + u\right)\right\} \tag{10}$$

**Lemma 7.** *Let $0 \leq c < 1$ and $e \geq 0$. Consider a positive sequence $\{a_t\}_{t \geq 0}$ defined according to the recursion*

$$a_{t+1} \leq c \cdot a_t + e \tag{11}$$

*for a general $a_0 \geq 0$. Then, for any $T \in \mathbb{N}$*

$$a_T \leq \frac{e}{1 - c} + a_0 \cdot c^T.$$

For the high-probability error bound, we begin with a set of "good" events, which we establish occur with high probability. When conditioned under these events, we obtain a collection of desirable bounds that, when combined, yield the guarantee in the theorem.

**Definition 2.** Given $\chi \in (0, 1]$, $\varrho \in (0, 1]$, $k \in \mathbb{N}$, $m \in \mathbb{N}$ and a matrix $A \in \mathbb{R}^{n \times d}$, we denote:

$$\mathscr{A}_1(A, \chi, k) := \left\{B \in \mathbb{R}^{k \times n} \mid \sup_{u \in \mathbb{R}^n, v \in \mathbb{R}^d} \frac{\left|u^\top\left(\frac{1}{k}B^\top B - \mathbb{I}_n\right)Av\right|}{\|u\| \cdot \|Av\|} \leq \frac{\chi}{2}\right\},$$

$$\mathscr{A}_2(A, \chi, k) := \left\{B \in \mathbb{R}^{k \times n} \mid \inf_{v \in \mathbb{R}^d} \frac{\|BAv\|^2}{k \cdot \|Av\|^2} \geq 1 - \chi\right\},$$

$$\mathscr{A}_3(\varrho, m) := \left\{v \in \mathbb{R}^m \mid \|v\|^2 \leq \left(\sqrt{2m} + \sqrt{4\log(1/\varrho)}\right)^2\right\}$$

**Lemma 8.** *Given $A \in \mathbb{R}^{n \times d}$ and $\mathsf{S} \sim \mathcal{N}(0, \mathbb{I}_{k \times n})$, there exists three universal constants $(c_0, c_1, c_2)$ such that for any $\chi \in (0, 1]$ satisfying $k\chi^2 \geq c_0 \cdot \operatorname{rank}(A)$ it holds that*

$$\mathbb{P}\left(\mathsf{S} \in \mathscr{A}_1(A, \chi, k) \cap \mathscr{A}_2(A, \chi, k)\right) \geq 1 - c_1 \cdot \exp\left\{-c_2 k\chi^2\right\}.$$

We further note that for a given $\varrho \in (0, 1]$ Lemma 8 guarantees that

$$\mathbb{P}\left(\mathsf{S} \in \mathscr{A}_1(A, \chi_0, k) \cap \mathscr{A}_2(A, \chi_0, k)\right) \geq 1 - \varrho$$

for $k \geq \max\left\{c_0 \cdot \operatorname{rank}(A), \frac{1}{c_2}\log(\frac{c_1}{\varrho})\right\}$ and where $\chi_0 = \sqrt{\frac{1}{k}\max\left\{c_0 \cdot \operatorname{rank}(A), \frac{1}{c_2}\log(\frac{c_1}{\varrho})\right\}}$.

**Lemma 9.** *Let $\xi \sim \mathcal{N}(\vec{0}_d, \mathbb{I}_d)$ and let $\varrho \in (0, 1]$. Then,*

$$\mathbb{P}\left(\xi \in \mathscr{A}_3(\varrho, d)\right) \geq 1 - \varrho^2.$$

Using Lemma 9, we note that it holds that $\|\xi\|^2 = O\left(\max\{d, \log(1/\varrho)\}\right)$ w.p. at least $1 - \varrho$.

The next lemma establishes guarantees for solutions of the form $\widehat{\theta}(A, b, B, v, u)$.

**Lemma 10.** *Given* $\chi \in \left(0, \frac{1}{2}\right]$, $A \in \mathbb{R}^{n \times d}$, $k \in \mathbb{N}$, $B \in \mathscr{A}_1(A, \chi, k) \cap \mathscr{A}_2(A, \chi, k)$, $b \in \mathbb{R}^n$ *and* $\theta_0 \in \mathbb{R}^d$ *we have*

$$\left\|\widehat{\theta}(A, b, B, \theta_0, \vec{0}_d) - \theta^*(A, b)\right\|_{A^\top A}^2 \leq \chi^2 \left\|\theta_0 - \theta^*(A, b)\right\|_{A^\top A}^2.$$

**Lemma 11.** *Given* $\chi \in (0, 1]$, $A \in \mathbb{R}^{n \times d}$ *and* $B \in \mathbb{R}^{k \times n}$ *such that* $B \in \mathscr{A}_1(A, \chi, k)$ *and* $A^\top A \succ 0$ *it holds*

$$\frac{1}{1 + \frac{\chi}{2}} \left(A^\top A\right)^{-1} \preceq \left(\frac{1}{k}(BA)^\top (BA)\right)^{-1} \preceq \frac{1}{1 - \frac{\chi}{2}} \left(A^\top A\right)^{-1} \tag{12}$$

*and furthermore for any vector* $v \in \mathbb{R}^d$

$$k^2 \left\|\left((BA)^\top (BA)\right)^{-1} v\right\|_{A^\top A}^2 \leq \frac{1}{\left(1 - \frac{\chi}{2}\right)^2} \cdot \frac{\|v\|^2}{\lambda_{\min}\left(A^\top A\right)}. \tag{13}$$

To aid us in choosing C, we derive a bound on $\left\|Y - X\widehat{\theta}_t\right\|_\infty$ for all $t \geq 0$ in terms of problem-dependent constants and under the high probability events $\mathcal{A}_1$, $\mathcal{A}_2$ and $\mathcal{A}_3$ and which we enforce to hold for all $t = 0, 1, \ldots, T - 1$. Based on this, we set C such that the clipping operation is a no-op under the high probability events that we already use to derive our bounds. The following lemma gives this bound. Recall that $\theta^*$ is the OLS solution based on $(X, Y)$.

**Lemma 12.** *Let* $Y \in \mathbb{R}^n$ *and* $X \in \mathbb{R}^{n \times d}$ *satisfying* $\mathrm{C}_X = 1$, *and define* $M := X^\top X$. *Suppose* $\{B_t\}_{t=0}^{T-1}$ *and* $\{v_t\}_{t=0}^{T-1}$ *are such that* $B_t \in \mathscr{A}_1(X, \chi, k) \cap \mathscr{A}_2(X, \chi, k)$ *and* $v_t \in \mathscr{A}_3(\varrho/2T, d)$ *for all* $t \in \{0, \ldots, T - 1\}$ *and for parameters* $\chi, \varrho \in (0, 1]$ *and* $k \in \mathbb{N}$. *Assume that*

$$\frac{\chi \sqrt{\kappa(M)}}{2 - \chi} < 1, \quad \text{where } \kappa(M) := \frac{\lambda_{\max}(M)}{\lambda_{\min}(M)}.$$

*If* $\{\widehat{\theta}_t\}_{t=0}^{T-1}$ *is defined according to the recursion*

$$\widehat{\theta}_{t+1}\left(\widehat{\theta}_t, \sigma v_t\right) = \widehat{\theta}_t + \left(\frac{1}{k}X^\top B_t^\top B_t X\right)^{-1} \left(X^\top \left(Y - X\widehat{\theta}_t\right) + \sigma v_t\right), \quad \widehat{\theta}_0 = \vec{0}_d,$$

*for* $\sigma \geq 0$, *then*

$$\left\|Y - X\widehat{\theta}_t\right\|_\infty \leq \mathrm{C}_Y + \left(1 + \left(\frac{\chi \sqrt{\kappa(M)}}{2 - \chi}\right)^t\right) \|\theta^*\| + \frac{2\sigma}{2 - \chi(1 + \sqrt{\kappa(M)})} \cdot \frac{\sqrt{2d} + \sqrt{4 \log(2T/\varrho)}}{\lambda_{\min}(M)}$$

### C.1. Proof of Theorem 1

*Proof.* We first prove the DP guarantee for the algorithm, and then its utility guarantee. Throughout the proof, given $(k, T, \mathrm{C})$ and for a target failure probability $\varrho$ we make the next choice of noise parameters

$$\gamma = \frac{\sqrt{kT}}{\sqrt{8 \log(5/\delta)} \left(-1 + \sqrt{1 + \frac{\varepsilon}{8 \log(5/\delta)}}\right)}, \quad \sigma = \frac{\sqrt{T}\mathrm{C}}{\sqrt{2 \log(4/\delta)} \left(-1 + \sqrt{1 + \frac{\varepsilon}{2 \log(4/\delta)}}\right)}, \quad \tau = \sqrt{2 \log \left(\max\left\{\frac{4}{\delta}, \frac{4}{\varrho}\right\}\right)}.$$

#### C.1.1. DIFFERENTIAL PRIVACY:

Let $(X', Y') \simeq (X, Y)$ i.e., $(X', Y')$ is a zero-out neighbor of $(X, Y)$. Then, since we set $\mathrm{C}_X = 1$, by Cauchy-Schwartz we get that

$$\left\|X'^\top \mathrm{clip}_\mathrm{C}(Y' - X'\widehat{\theta}_t) - X^\top \mathrm{clip}_\mathrm{C}(Y - X\widehat{\theta}_t)\right\| \leq \mathrm{C} \tag{14}$$

and thus the $\ell_2$ sensitivity of the quantity $X^\top \mathrm{clip}_C \left( Y - X\widehat{\theta}_t \right)$ is at most C irrespective of $\widehat{\theta}_t$.

We note that Algorithm 1 involves $2T + 1$ calls to private mechanisms: $T$ calls to the Gaussian mechanism for constructing $\widetilde{G}_t$, another $T$ calls for forming $\widetilde{X}_t$ using sketching, and one call for calculating $\eta$, which uses a private estimate $\widetilde{\lambda}$ of the minimal eigenvalue $\lambda_{\min}^X$. Our proof follows by using Lemma 1 and Lemma 2 for calculating the RDP guarantees for each of the steps that form $\widetilde{G}_t$ and $\widetilde{X}_t$, and uses the composition theorem for RDP to calculate the overall RDP guarantees of these steps, separately. Then, we convert these RDP guarantees to $(\varepsilon, \delta)$-DP, and sum them together with the $(\varepsilon, \delta)$-DP guarantee of the step that privately checks the minimal eigenvalue.

**Eigenvalue Test.** The calculations of the minimal eigenvalue are similar to those done in Lev et al. (2025, Appendix. D). Thus, for the choice $\eta = \gamma/\sqrt{k}$ their privacy guarantees are $\left( \frac{\sqrt{2k \log(5/\delta)}}{\gamma}, \frac{\delta}{4} \right)$-DP.

Moreover, for our $\tau$ it holds that $\mathbb{P}\left( \widetilde{\lambda} \leq \lambda_{\min}^X \right) \geq 1 - \frac{1}{2}\exp\left\{ -\frac{\tau^2}{2} \right\} \geq 1 - \frac{\delta}{4}$.

**Gaussian Mechanism.** By Lemma 1 and (14), each step has RDP guarantee of $\left( \alpha, \frac{\alpha C^2}{2\sigma^2} \right)$.

**Sketching.** Under the event that $\left\{ \widetilde{\lambda} \leq \lambda_{\min}^X \right\}$ we have $\lambda_{\min}^X + \eta^2 \geq \gamma$ for all the cases of the algorithm. Thus, by Lemma 2, the RDP guarantee of each of these steps has RDP guarantee of $(\alpha, \varphi(\alpha; k, \gamma))$ where $\varphi$ was defined in Lemma 2.

**Composition.** We compose the $T$ different steps of the Gaussian mechanism and the $T$ sketching steps in RDP. Then, we convert this to $(\varepsilon, \delta)$-DP via the conversion from (Canonne et al., 2020), and then compose with the step that privately tests for the minimal eigenvalue in $(\varepsilon, \delta)$-DP. For the $T$ steps of the Gaussian mechanism, we get

$$\widehat{\varepsilon}_{\text{Gauss}} = \min_{\alpha > 1} \left\{ \frac{\alpha T C^2}{2\sigma^2} + \frac{\log(4/\delta) + (\alpha - 1)\log(1 - 1/\alpha) - \log(\alpha)}{\alpha - 1} \right\}$$

$$\leq \min_{\alpha > 1} \left\{ \frac{\alpha T C^2}{2\sigma^2} + \frac{\log(4/\delta)}{\alpha - 1} \right\}$$

$$= \frac{\sqrt{2T \log(4/\delta)}C}{\sigma} + \frac{T C^2}{2\sigma^2}$$

and $\widehat{\delta}_{\text{Gauss}} = \delta/4$. For the $T$ sketching steps, we get

$$\widehat{\varepsilon}_{\text{GaussMix}} = \min_{1 < \alpha < \gamma} \left\{ T\varphi(\alpha; k, \gamma) + \frac{\log(4/\delta) + (\alpha - 1)\log(1 - 1/\alpha) - \log(\alpha)}{\alpha - 1} \right\}$$

and $\widehat{\delta}_{\text{GaussMix}} = \delta/4$. Then, we get that the overall process is $(\widehat{\varepsilon}, \widehat{\delta})$-DP where

$$\widehat{\delta} = \frac{\delta}{2} + \widehat{\delta}_{\text{Gauss}} + \widehat{\delta}_{\text{GaussMix}} = \delta$$

where the first $\delta/2$ is by summing the $\delta/4$ contributions from the test of the minimal eigenvalue and the probability of the event $\left\{ \widetilde{\lambda} \geq \lambda_{\min}^X \right\}$ and where $\widehat{\varepsilon}$ is

$$\widehat{\varepsilon} \leq \frac{\sqrt{2k \log(5/\delta)}}{\gamma} + \widehat{\varepsilon}_{\text{GaussMix}} + \widehat{\varepsilon}_{\text{Gauss}}$$

$$\leq \frac{\sqrt{2k \log(5/\delta)}}{\gamma} + \min_{1 < \alpha < \gamma} \left\{ T\varphi(\alpha; k, \gamma) + \frac{\log(4/\delta) + (\alpha - 1)\log(1 - 1/\alpha) - \log(\alpha)}{\alpha - 1} \right\}$$

$$+ \frac{\sqrt{2T \log(4/\delta)}C}{\sigma} + \frac{T C^2}{2\sigma^2}$$

$$\leq \frac{\sqrt{2kT \log(5/\delta)}}{\gamma} + \min_{1 < \alpha < \gamma} \left\{ T\varphi(\alpha; k, \gamma) + \frac{\log(4/\delta) + (\alpha - 1)\log(1 - 1/\alpha) - \log(\alpha)}{\alpha - 1} \right\}$$

$$+ \frac{\sqrt{2T \log(4/\delta)}C}{\sigma} + \frac{T C^2}{2\sigma^2}$$

where we note that the increase of the first term by a factor $\sqrt{T}$ in the last step results only in a constant factor increase in the overall optimized value, since the term inside the minimization grows with $T$, though it allows us to use the upper bounds derived in (Lev et al., 2025, Section. 4). Now, we note that for our choice of $\sigma$, the sum of the third and the fourth terms evaluates to $\varepsilon/2$. Moreover, the first two terms match the privacy guarantees from (Lev et al., 2025, Theorem. 1) with $k \leftarrow kT$ and for a target $\delta$ of $\delta/4$. Then, using (9), which, as shown in (Lev et al., 2025), is a consequence of Proposition 2, this term is upper bounded by $\frac{kT}{2\gamma^2} + \frac{2\sqrt{2kT\log(5/\delta)}}{\gamma}$ and for our choice of $\gamma$ this evaluates to $\varepsilon/2$. This concludes the proof. $\square$

### C.1.2. EXCESS EMPIRICAL RISK:

Our proof strategy involves obtaining a high probability bound on $\|\widehat{\theta}_{t+1} - \theta^*\|^2_{X^\top X + \eta^2 \mathbb{I}_d}$ in terms of $\|\widehat{\theta}_t - \theta^*\|^2_{X^\top X + \eta^2 \mathbb{I}_d}$ for all $t = 0, 1, \ldots, T-1$. Then, we translate this to a high probability bound on $\|\widehat{\theta}_T - \theta^*\|^2_{X^\top X}$, which by Wang (2018, Lemma. 5) corresponds to the excess empirical risk. Throughout, we denote the private eigenvalue that the algorithm uses (that is defined inside `CalibrateMixingNoise`) by $\widetilde{\lambda}$, and we use the shorthand $\overline{X}_\eta := (X^\top, \eta \mathbb{I}_d)^\top$ and $M := X^\top X + \eta^2 \mathbb{I}_d = \overline{X}_\eta^\top \overline{X}_\eta$. Moreover, we use the definitions of the sets $\mathscr{A}_1, \mathscr{A}_2$ and $\mathscr{A}_3$ from Definition 2.

*Step 1: Defining the High Probability Event:*

The analysis is conditioned on the intersection of the next events

$$\left\{ (\mathsf{S}_t, \xi_t) \in \mathscr{A}_1\left(\overline{X}_\eta, \chi, k\right) \cap \mathscr{A}_2\left(\overline{X}_\eta, \chi, k\right) \right\}, \quad \left\{ \zeta_t \in \mathscr{A}_3\left(\varrho/2T, d\right) \right\}, \quad \left\{ \widetilde{\lambda} \leq \lambda^X_{\min} \right\}$$

for $t = 0, \ldots, T-1$ and for some $\chi \in (0, 1]$ and for the target failure probability $\varrho \in (0, 1]$ and where $k$ and $T$ will be specified later. We denote the intersection between these events and over the different $T$ iterations by $\mathscr{A}_{\text{tot}}$, and the complement of this event by $\mathscr{A}^c_{\text{tot}}$.

First, recall that for our choice of $\tau$ and by Lemma 9

$$\mathbb{P}\left(\widetilde{\lambda} \leq \lambda^X_{\min}\right) \geq 1 - \frac{1}{2}\exp\left\{-\frac{\tau^2}{2}\right\} \geq 1 - \frac{\varrho}{4},$$

$$\mathbb{P}\left(\zeta_t \in \mathscr{A}_3\left(\varrho/2T, d\right)\right) \geq 1 - \left(\frac{\varrho}{2T}\right)^2, \quad \forall t = 0, \ldots, T-1.$$

Then, whenever $k \geq 8 \cdot \max\left\{c_0 d, \frac{1}{c_2}\log\left(\frac{4Tc_1}{\varrho}\right)\right\}$ where $c_0, c_1$ and $c_2$ are the constants from Lemma 8, and since under $\left\{\widetilde{\lambda} \leq \lambda^X_{\min}\right\}$ it holds that $\text{rank}\left(\overline{X}_\eta\right) = d$, Lemma 8 holds with respect to $(\mathsf{S}_t, \xi_t)$ for some $\chi \leq \frac{1}{\sqrt{8}}$ where $A \leftarrow \overline{X}_\eta$ and we have

$$\mathbb{P}\left((\mathsf{S}_t, \xi_t) \in \mathscr{A}_1(\overline{X}_\eta, \chi, k) \cap \mathscr{A}_2(\overline{X}_\eta, \chi, k)\right) \geq 1 - \frac{\varrho}{4T}, \quad \forall t = 0, \ldots, T-1.$$

By a union bound,

$$\mathbb{P}\left(\mathscr{A}_{\text{tot}}\right) = 1 - \mathbb{P}\left(\mathscr{A}^c_{\text{tot}}\right)$$

$$\geq 1 - \mathbb{P}\left(\widetilde{\lambda} > \lambda^X_{\min}\right) - \sum_{t=0}^{T-1}\mathbb{P}\left(\zeta_t \notin \mathscr{A}_3\left(\varrho/2T, d\right)\right) - \sum_{t=0}^{T-1}\mathbb{P}\left((\mathsf{S}_t, \xi_t) \notin \mathscr{A}_1(\overline{X}_\eta, \chi, k) \cap \mathscr{A}_2(\overline{X}_\eta, \chi, k)\right)$$

$$\geq 1 - \frac{\varrho}{4} - T \cdot \left(\frac{\varrho}{2T}\right)^2 - \frac{\varrho}{4}$$

$$\geq 1 - \varrho$$

where we used $T\left(\frac{\varrho}{2T}\right)^2 \leq \frac{\varrho}{2}$ which holds since $\frac{\varrho}{2} \leq 1$ and since $T \geq 1$.

*Step 2: No-Clipping Guarantee:*

We will now show that conditioned on $\mathscr{A}_{\text{tot}}$, there exist parameters $k, \mathsf{C}$ and $T$ under which clipping does not occur throughout the $T$ iterations of the algorithm. To that end, we first make the next definitions

$$\kappa^X(a) := \frac{\lambda^X_{\max} + a}{\lambda^X_{\min} + a}, \quad \psi := \max\left\{0, \gamma - \lambda^X_{\min}\right\}, \quad \kappa(A) := \frac{\lambda_{\max}(A)}{\lambda_{\min}(A)}.$$

The proof follows by first specifying conditions $k$ needs to satisfy given the number of iterations $T$. Under these conditions, and the analysis presented below, we show that clipping does not occur if $\mathsf{C} = O(\max\{\mathsf{C}_Y^2, \|\theta^*\|^2\})$. In the third part of the proof, we optimize the derived upper bound over the number of iterations $T$, providing a closed-form choice of the number of iterations.

We first note that by the normalization $\|Y\|_\infty \leq \mathsf{C}_Y$ and since $\widehat{\theta}_0 = \vec{0}_d$, whenever $\mathsf{C} \geq \mathsf{C}_Y$, clipping does not occur on the first iteration. Moreover, and for the sake of the analysis, we note that without the clipping operation, the update step of Line 7 is mathematically equivalent to the update

$$\widehat{\theta}_{t+1} = \widehat{\theta}_t + \left(\frac{1}{k}\overline{X}^\top \widetilde{\mathsf{B}}_t^\top \widetilde{\mathsf{B}}_t \overline{X}\right)^{-1} \left(\overline{X}^\top \left(\overline{Y} - \overline{X}\widehat{\theta}_t\right) + \sigma \zeta_t\right)$$

where $\overline{X} := (X^\top, \underbrace{\mathbb{I}_d, \dots, \mathbb{I}_d}_{\eta^2 \text{ times}})^\top$ and $\overline{Y} := (Y^\top, \underbrace{\vec{0}_d^\top, \dots, \vec{0}_d^\top}_{\eta^2 \text{ times}})^\top$ and where $\widetilde{\mathsf{B}}_t = \left(\mathsf{S}_t, \xi_t^{(1)}, \dots, \xi_t^{(\eta^2)}\right)$ for $\xi_t^{(i)} \overset{\text{iid}}{\sim}$
$\mathcal{N}(0, \mathbb{I}_{k \times d})$. We first note that the OLS regressor obtained by $\overline{X}$ and $\overline{Y}$ is $\theta^*(\eta^2)$ and moreover $\overline{X}^\top \overline{X} = X^\top X + \eta^2 \mathbb{I}_d = M$.
Then, whenever $\frac{\chi\sqrt{\kappa(M)}}{2-\chi} < 1$, which since $\kappa(M) = \kappa^X(\eta^2) = \kappa^X(\psi)$ corresponds to picking $k$ such that $k > \frac{c_0 d}{4} \cdot \kappa^X(\psi)$
[4], we use Lemma 12 with $v_t \leftarrow \zeta_t$ and $B_t \leftarrow \widetilde{\mathsf{B}}_t, X \leftarrow \overline{X}$ and $Y \leftarrow \overline{Y}$ and get

$$\left\|Y - X\widehat{\theta}_t\right\|_\infty$$

$$\overset{(a)}{\leq} \left\|\overline{Y} - \overline{X}\widehat{\theta}_t\right\|_\infty$$

$$\overset{(b)}{\leq} \mathsf{C}_Y + \left\|\theta^*\left(\eta^2\right)\right\| + \frac{1}{1 - \frac{\chi\sqrt{\kappa^X(\eta^2)}}{2-\chi}} \cdot \frac{\sigma}{1 - \frac{\chi}{2}} \cdot \frac{\sqrt{2d} + \sqrt{4\log\left(2T/\varrho\right)}}{\lambda_{\min}(M)} + \left(\frac{\chi\sqrt{\kappa^X(\eta^2)}}{2-\chi}\right)^t \left\|\theta^*\left(\eta^2\right)\right\|$$

$$\overset{(c)}{\leq} \mathsf{C}_Y + \|\theta^*\| + \frac{1}{1 - \frac{\chi\sqrt{\kappa^X(\eta^2)}}{2-\chi}} \cdot \frac{\sigma}{1 - \frac{\chi}{2}} \cdot \frac{\sqrt{2d} + \sqrt{4\log\left(2T/\varrho\right)}}{\gamma} + \left(\frac{\chi\sqrt{\kappa^X(\eta^2)}}{2-\chi}\right)^t \|\theta^*\|$$

$$\overset{(d)}{\leq} \mathsf{C}_Y + \|\theta^*\| + \frac{1}{1 - \frac{\chi\sqrt{\kappa^X(\eta^2)}}{2-\chi}} \cdot \frac{\mathsf{C}}{1 - \frac{\chi}{2}} \cdot \frac{\sqrt{2d} + \sqrt{4\log\left(2T/\varrho\right)}}{\sqrt{k}} + \left(\frac{\chi\sqrt{\kappa^X(\eta^2)}}{2-\chi}\right)^t \|\theta^*\|$$

$$\overset{(e)}{\leq} \mathsf{C}_Y + \|\theta^*\| + \frac{1}{1 - \frac{\sqrt{\kappa^X(\eta^2)}}{2\sqrt{b}-1}} \cdot \frac{\mathsf{C}}{2\sqrt{b}-1} + \frac{\sqrt{\kappa^X(\eta^2)}}{2\sqrt{b}-1} \|\theta^*\|$$

$$= \mathsf{C}_Y + \|\theta^*\| + \frac{\mathsf{C}}{2\sqrt{b}-1-\sqrt{\kappa^X(\eta^2)}} + \frac{\sqrt{\kappa^X(\eta^2)}}{2\sqrt{b}-1} \|\theta^*\|$$

where (a) is by the definition of $\overline{Y}$ and $\overline{X}$, (b) is by Lemma 12, (c) is since under $\mathscr{A}_{\text{tot}}$ it holds that $\lambda_{\min}(M) \geq \gamma$ and since $\|\theta^*(\eta^2)\| \leq \|\theta^*\|$, (d) is by substituting $\sigma$ and $\gamma$ and further using the inequality $\frac{-1+\sqrt{1+x}}{-1+\sqrt{1+z}} \leq \sqrt{\frac{x}{z}}$ which holds for all $z \geq x \geq 0$ and (e) is for $k$ satisfying $k \geq b \cdot \max\left\{c_0 d, \frac{1}{c_2}\log\left(\frac{4Tc_1}{\varrho}\right)\right\}$ and $k \geq b \cdot \left(\sqrt{2d} + \sqrt{4\log(2T/\varrho)}\right)^2$ for some constant $b$ and for which we further have $\chi < \frac{1}{\sqrt{b}}$ and further by using the upper bound $\left(\frac{\chi\sqrt{\kappa(\eta^2)}}{2-\chi}\right)^t \leq \frac{\chi\sqrt{\kappa(\eta^2)}}{2-\chi}$. We note that this is less than $\mathsf{C}$ whenever

$$\frac{\mathsf{C}_Y}{1 - \frac{1}{2\sqrt{b}-1-\sqrt{\kappa^X(\eta^2)}}} + \frac{1}{1 - \frac{1}{2\sqrt{b}-1-\sqrt{\kappa^X(\eta^2)}}}\left(1 + \frac{\sqrt{\kappa^X(\eta^2)}}{2\sqrt{b}-1}\right)\|\theta^*\| \leq \mathsf{C}$$

provided that $2\sqrt{b}-1-\sqrt{\kappa^X(\eta^2)} > 1$, and which since $\kappa^X(\eta^2) \geq 1$ corresponds to the previous condition $k \geq \widetilde{c} \cdot d \cdot \kappa^X(\eta^2)$, for some constant $\widetilde{c}$. Recalling that preventing clipping in the first iteration requires $\mathsf{C} \geq \mathsf{C}_Y$, we obtain the following

---

[4]since $\kappa^X(\psi)$ is monotonically decreasing $\psi$, and since $\psi$ is monotonically increasing in $k$ (via $\gamma$), we are guaranteed to have such a solution by choosing a large enough $k$

expression for C:

$$\mathrm{C} = \max\left\{1, \frac{3}{1 - \frac{1}{2\sqrt{b}-1-\sqrt{\kappa^X(\eta^2)}}}\right\} \cdot \max\left\{\mathrm{C}_Y, \|\theta^*\|\right\}$$

which was obtained by using the upper bound $1 + \frac{\sqrt{\kappa^X(\eta^2)}}{2\sqrt{b}-1} \leq 2$ which holds under the choice $2\sqrt{b} - 1 - \sqrt{\kappa^X(\eta^2)} > 1$. Then, we note that for every $b$ such that $\sqrt{b} \geq 2 + \sqrt{\kappa^X(\eta^2)}$ it holds that $\frac{3}{1 - \frac{1}{2\sqrt{b}-1-\sqrt{\kappa^X(\eta^2)}}} \leq 6$ and for which we get $\mathrm{C} = 6 \cdot \max\left\{\mathrm{C}_Y, \|\theta^*\|\right\}$ provided that

$$k \geq b \cdot \max\left\{\max\left\{c_0, \kappa^X(\psi)\right\} \cdot d, \frac{1}{c_2} \log\left(\frac{4Tc_1}{\varrho}\right), \left(\sqrt{2d} + \sqrt{4\log(4T/\varrho)}\right)^2\right\} \tag{15}$$

and for $b \geq \max\{8, \widetilde{c}\}$. Thus, under this choice for $k$, clipping is guaranteed not to occur during the iterations of the algorithm. To further simplify these conditions on $k$, recall that $\gamma = \Theta\left(\frac{\sqrt{kT \log(1/\delta)}}{\varepsilon}\right)$. Then, since

$$\kappa^X(\psi) = \frac{\lambda^X_{\max} + \psi}{\lambda^X_{\min} + \psi} = \frac{\lambda^X_{\max} - \lambda^X_{\min}}{\lambda^X_{\min} + \psi} + 1 = \frac{\lambda^X_{\max} - \lambda^X_{\min}}{\max\left\{\lambda^X_{\min}, \gamma\right\}} + 1 \leq \frac{\lambda^X_{\max} - \lambda^X_{\min}}{\gamma} + 1,$$

the condition on $k$ can be further read as

$$k \geq \widetilde{c} \cdot d \cdot \left(\frac{\left(\lambda^X_{\max} - \lambda^X_{\min}\right)\varepsilon}{\sqrt{kT \log(1/\delta)}} + 1\right). \tag{16}$$

Provided that $k \geq c_0 d$, we drop the $+1$ inside (16), and get the next additional constraint on $k$

$$k \geq \widetilde{c}_1\left(d \cdot \frac{\left(\lambda^X_{\max} - \lambda^X_{\min}\right)\varepsilon}{\sqrt{T \log(1/\delta)}}\right)^{2/3}$$

where $\widetilde{c}_1 = \widetilde{c}^{\,2/3}$ and which we enforce by requiring

$$k \geq \widetilde{c}_1\left(d \cdot \frac{\left(\lambda^X_{\max} - \lambda^X_{\min}\right)\varepsilon}{\sqrt{\log(1/\delta)}}\right)^{2/3} := \widetilde{c}_1 \cdot g_X$$

and where the overall choice of $k$ then needs to admit the constraint

$$k \geq b \cdot \max\left\{c_0 d, \frac{1}{c_2} \log\left(\frac{4Tc_1}{\varrho}\right), \left(\sqrt{2d} + \sqrt{4\log(4T/\varrho)}\right)^2, g_X\right\}$$

for some constant $b \geq \max\{8, \widetilde{c}_1\}$. From now on, we fix the value of $k$ on

$$k = b \cdot \max\left\{c_0 d, \frac{1}{c_2} \log\left(\frac{4Tc_1}{\varrho}\right), \left(\sqrt{2d} + \sqrt{4\log(4T/\varrho)}\right)^2, g_X\right\}.$$

Using this value of $k$ (which depends on the parameter $T$, which we have not specified yet), we now derive the explicit guarantees on the excess empirical risk $\mathcal{R}$. Based on the stated guarantee, we provide a choice of $T$ which minimizes them and yields the closed-form upper bound from (3).

*Step 3: Deriving Guarantees on the Error Under $\mathscr{A}_{\text{tot}}$:*

Continuing steps 1 and 2, our analysis now is done conditioned on $\mathscr{A}_{\text{tot}}$, and we further omit the clipping operation. Our goal now is to upper bound $\|\widehat{\theta}_T - \theta^*\|^2_{X^\top X}$ conditioned on the event $\mathscr{A}_{\text{tot}}$, and which by Lemma 4 corresponds to the excess empirical risk. To achieve this, we first derive a recursive characterization of $\|\widehat{\theta}_{t+1} - \theta^*\|^2_M$ in terms of $\|\widehat{\theta}_t - \theta^*\|^2_M$.

Throughout, we use $\widehat{\theta}_{t+1}(\eta, \sigma)$ to denote the update step from Line 7 in Algorithm 1. Specifically, given a vector $\theta$, we use the notation

$$\widehat{\theta}_{t+1}(\eta, \sigma; \theta) := \theta + \left(\frac{1}{k}\left((\mathsf{S}_t, \xi_t)\overline{X}_\eta\right)^\top \left((\mathsf{S}_t, \xi_t)\overline{X}_\eta\right)\right)^{-1}\left(X^\top(Y - X\theta) - \eta^2\theta + \sigma\zeta_t\right)$$

and note that $\widehat{\theta}_{t+1}(\eta, \sigma; \widehat{\theta}_t) = \widehat{\theta}_{t+1}$ and where, following our previous assumption, we have omitted the clipping operation from this update step. Moreover, by this definition, we note that for any $\theta$ it holds

$$\widehat{\theta}_{t+1}(\eta, \sigma; \theta) - \widehat{\theta}_{t+1}(\eta, 0; \theta) = \left(\frac{1}{k}\left((\mathsf{S}_t, \xi_t)\overline{X}_\eta\right)^\top \left((\mathsf{S}_t, \xi_t)\overline{X}_\eta\right)\right)^{-1}\sigma\zeta_t. \tag{17}$$

For a general $\eta > 0, \sigma > 0$, since $\|a + b\|^2 \le 2\left(\|a\|^2 + \|b\|^2\right)$ we have

$$\left\|\widehat{\theta}_{t+1}(\eta, \sigma) - \theta^*(\eta^2)\right\|_M^2$$
$$= \left\|\widehat{\theta}_{t+1}(\eta, \sigma; \widehat{\theta}_t(\eta, \sigma)) - \theta^*(\eta^2)\right\|_M^2$$
$$= \left\|\widehat{\theta}_{t+1}(\eta, \sigma; \widehat{\theta}_t(\eta, \sigma)) - \widehat{\theta}_{t+1}(\eta, 0; \widehat{\theta}_t(\eta, \sigma)) + \widehat{\theta}_{t+1}(\eta, 0; \widehat{\theta}_t(\eta, \sigma)) - \theta^*(\eta^2)\right\|_M^2$$
$$\le \underbrace{2\left\|\widehat{\theta}_{t+1}(\eta, \sigma; \widehat{\theta}_t(\eta, \sigma)) - \widehat{\theta}_{t+1}(\eta, 0; \widehat{\theta}_t(\eta, \sigma))\right\|_M^2}_{T_1} + \underbrace{2\left\|\widehat{\theta}_{t+1}(\eta, 0; \widehat{\theta}_t(\eta, \sigma)) - \theta^*(\eta^2)\right\|_M^2}_{T_2}.$$

By using (17) and Lemma 11, the term $T_1$ is bounded as

$$T_1 = 2\left\|\widehat{\theta}_{t+1}(\eta, \sigma; \widehat{\theta}_t(\eta, \sigma)) - \widehat{\theta}_{t+1}(\eta, 0; \widehat{\theta}_t(\eta, \sigma))\right\|_M^2$$
$$= 2\left\|\left(\frac{1}{k}\left((\mathsf{S}_t, \xi_t)\overline{X}_\eta\right)^\top \left((\mathsf{S}_t, \xi_t)\overline{X}_\eta\right)\right)^{-1}\sigma\zeta_t\right\|_M^2$$
$$\le \frac{2}{\left(1 - \frac{\chi}{2}\right)^2} \cdot \frac{\sigma^2\|\zeta_t\|^2}{\lambda_{\min}^X + \eta^2}$$

where the last inequality is due to Lemma 11 applied with $B \leftarrow (\mathsf{S}_t, \xi_t) \in \mathbb{R}^{k \times (n+d)}, A \leftarrow \overline{X}_\eta \in \mathbb{R}^{(n+d) \times d}, v \leftarrow \sigma\zeta_t$ and since $(\mathsf{S}_t, \xi_t) \in \mathscr{A}_1(\overline{X}_\eta, \chi, k)$. We note that $\theta^*(\eta^2)$ corresponds to the solution of the OLS system with design matrix $\overline{X}_\eta$ and response vector $\overline{Y} := (Y^\top, \vec{0}_d^\top)^\top$, since

$$\left(\overline{X}_\eta^\top \overline{X}_\eta\right)^{-1}\overline{X}_\eta^\top \overline{Y} = \left(X^\top X + \eta^2\mathbb{I}_d\right)^{-1}(X^\top, \eta\mathbb{I}_d)\begin{pmatrix}Y \\ \vec{0}_d\end{pmatrix} = \left(X^\top X + \eta^2\mathbb{I}_d\right)^{-1}X^\top Y = \theta^*(\eta^2).$$

Moreover, by Lemma 6 with $v = Y - X\widehat{\theta}_t(\eta, \sigma), \theta_0 = \widehat{\theta}_t(\eta, \sigma)$ and $\sigma = 0$ we note that

$$\widehat{\theta}_{t+1}(\eta, 0; \widehat{\theta}_t(\eta, \sigma)) = \operatorname*{argmin}_\theta \left\{\frac{1}{2k}\left\|(\mathsf{S}, \xi)\overline{X}_\eta\left(\theta - \widehat{\theta}_t(\eta, \sigma)\right)\right\|^2 - \theta^\top \overline{X}_\eta^\top \begin{pmatrix}Y - X\widehat{\theta}_t(\eta, \sigma) \\ -\eta\widehat{\theta}_t(\eta, \sigma)\end{pmatrix}\right\}$$
$$= \operatorname*{argmin}_\theta \left\{\frac{1}{2k}\left\|(\mathsf{S}, \xi)\overline{X}_\eta\left(\theta - \widehat{\theta}_t(\eta, \sigma)\right)\right\|^2 - \theta^\top \overline{X}_\eta^\top \left(\overline{Y} - \overline{X}_\eta\widehat{\theta}_t(\eta, \sigma)\right)\right\}.$$

Thus, applying Lemma 10 with our $\chi, A \leftarrow \overline{X}_\eta, B \leftarrow (\mathsf{S}_t, \xi_t)$ and $\theta_0 = \widehat{\theta}_t(\eta, \sigma)$ we know that

$$T_2 = 2\left\|\widehat{\theta}_{t+1}(\eta, 0; \widehat{\theta}_t(\eta, \sigma)) - \theta^*(\eta^2)\right\|_M^2$$
$$\le 2\chi^2 \cdot \left\|\widehat{\theta}_t(\eta, \sigma) - \theta^*(\eta^2)\right\|_M^2$$
$$= (\sqrt{2}\chi)^2 \cdot \left\|\widehat{\theta}_t(\eta, \sigma) - \theta^*(\eta^2)\right\|_M^2.$$

Together, these bounds on $T_1$ and $T_2$ and since $\zeta_t \in \mathscr{A}_3(\varrho/2T, d)$ implies that

$$
\left\| \widehat{\theta}_{t+1}(\eta, \sigma; \widehat{\theta}_t(\eta, \sigma)) - \theta^*(\eta^2) \right\|_M^2
$$
$$
\leq \frac{2}{\left(1 - \frac{\chi}{2}\right)^2} \cdot \frac{\sigma^2 \|\zeta_t\|^2}{\lambda_{\min}^X + \eta^2} + (\sqrt{2}\chi)^2 \left\| \widehat{\theta}_t(\eta, \sigma) - \theta^*(\eta^2) \right\|_M^2
$$
$$
\leq \frac{2}{\left(1 - \frac{\chi}{2}\right)^2} \cdot \frac{\sigma^2 \left(\sqrt{2d} + \sqrt{4\log(2T/\varrho)}\right)^2}{\lambda_{\min}^X + \eta^2} + (\sqrt{2}\chi)^2 \left\| \widehat{\theta}_t(\eta, \sigma) - \theta^*(\eta^2) \right\|_M^2 .
$$

Now, using Lemma 7 with resepct to the sequence $\left\| \widehat{\theta}_{t+1}(\eta, \sigma; \widehat{\theta}_t(\eta, \sigma)) - \theta^*(\eta^2) \right\|_M^2$ with the initialization $\widehat{\theta}_0 \leftarrow \vec{0}_d, c \leftarrow$

$2\chi^2$ and $e \leftarrow \frac{2}{\left(1 - \frac{\chi}{2}\right)^2} \cdot \frac{\sigma^2 \left(\sqrt{2d} + \sqrt{4\log(2T/\varrho)}\right)^2}{\lambda_{\min}^X + \eta^2}$ yields

$$
\left\| \widehat{\theta}_T(\eta, \sigma) - \theta^*(\eta^2) \right\|_M^2 \leq \frac{2}{(1 - 2\chi^2)(1 - \frac{\chi}{2})^2} \cdot \frac{\sigma^2 \left(\sqrt{2d} + \sqrt{4\log(2T/\varrho)}\right)^2}{\lambda_{\min}^X + \eta^2} + (\sqrt{2}\chi)^{2T} \left\| \widehat{\theta}_0 - \theta^*(\eta^2) \right\|_M^2
$$
$$
= \frac{2}{(1 - 2\chi^2)(1 - \frac{\chi}{2})^2} \cdot \frac{\sigma^2 \left(\sqrt{2d} + \sqrt{4\log(2T/\varrho)}\right)^2}{\lambda_{\min}^X + \eta^2} + (\sqrt{2}\chi)^{2T} \left\| \theta^*(\eta^2) \right\|_M^2
$$

and from which we conclude that

$$
\left\| \widehat{\theta}_T(\eta, \sigma) - \theta^*\left(\eta^2\right) \right\|_M^2 = O\left( \frac{\sigma^2 \max\{d, \log(T/\varrho)\}}{\lambda_{\min}^X + \eta^2} \right) + (\sqrt{2}\chi)^{2T} \left\| \theta^*\left(\eta^2\right) \right\|_M^2 .
$$

Then, applying the inequality $\|a + b\|^2 \leq 2\left(\|a\|^2 + \|b\|^2\right)$ and Lemma 5 together with our choice of $k$ yields

$$
\left\| \widehat{\theta}_T(\eta, \sigma) - \theta^* \right\|_M^2 \leq 2 \left\| \widehat{\theta}_T(\eta, \sigma) - \theta^*\left(\eta^2\right) \right\|_M^2 + 2 \left\| \theta^*\left(\eta^2\right) - \theta^* \right\|_M^2
$$
$$
\leq O\left( \frac{\sigma^2 \max\{d, \log(T/\varrho)\}}{\lambda_{\min}^X + \eta^2} \right) + (\sqrt{2}\chi)^{2T} \left\| \theta^*\left(\eta^2\right) \right\|_M^2 + \eta^2 \|\theta^*\|^2 . \tag{18a}
$$

Now, by our choice of $\gamma$ and $\sigma$ and under $\left\{ \widetilde{\lambda} \leq \lambda_{\min}^X \right\}$ it holds

$$
\gamma = \Theta\left( \frac{\sqrt{kT \log(1/\delta)}}{\varepsilon} \right), \quad \sigma^2 = \Theta\left( \frac{T \log(1/\delta) \mathbf{C}^2}{\varepsilon^2} \right), \quad \lambda_{\min}^X + \eta^2 \geq \max\left\{ \gamma, \lambda_{\min}^X \right\} . \tag{19}
$$

Furthermore,

$$
\left\| \widehat{\theta}_T(\eta, \sigma) - \theta^* \right\|_M^2 = \left\| \widehat{\theta}_T(\eta, \sigma) - \theta^* \right\|_{X^\top X}^2 + \eta^2 \left\| \widehat{\theta}_T(\eta, \sigma) - \theta^* \right\|^2
$$
$$
\geq \left\| \widehat{\theta}_T(\eta, \sigma) - \theta^* \right\|_{X^\top X}^2
$$

which holds by the inequality $X^\top X + \eta^2 \mathbb{I}_d \succeq X^\top X$ and moreover

$$
\left\| \theta^*\left(\eta^2\right) \right\|_{X^\top X + \eta^2 \mathbb{I}_d}^2 \leq \left(\lambda_{\max}^X + \eta^2\right) \left\| \theta^*(\eta^2) \right\|^2
$$
$$
\leq \kappa^X(\eta^2) \left(\lambda_{\min}^X + \eta^2\right) \|\theta^*\|^2
$$

which holds since $\|v\|_A \leq \lambda_{\max}(A) \|v\|$ and $\left\| \theta^*(\eta^2) \right\| \leq \|\theta^*\|$. We then note that whenever $\eta^2 = 0$ it holds that

$$
(\sqrt{2}\chi)^{2T} \left\| \theta^*(\eta^2) \right\|_M^2 + \eta^2 \|\theta^*\| \leq \kappa^X(0)(\sqrt{2}\chi)^{2T} \lambda_{\min}^X \|\theta^*\|^2
$$

and moreover whenever $\eta > 0$ it holds that

$$(\sqrt{2}\chi)^{2T} \left\|\theta^*(\eta^2)\right\|_M^2 + \eta^2 \|\theta^*\| \le (\sqrt{2}\chi)^{2T} \kappa^X(\eta^2) \left(\lambda_{\min}^X + \eta^2\right) \|\theta^*\|^2 + \eta^2 \|\theta^*\|^2.$$

Substituting these together with (19) and the $k$ from (15) and $\mathrm{C}^2 = \Theta\left(\mathrm{C}_Y^2 + \|\theta^*\|^2\right)$ inside (18a) yields

$$\left\|\widehat{\theta}_T(\eta, \sigma) - \theta^*\right\|_M^2$$
$$\le O\left(\frac{\sigma^2 \max\{d, \log(T/\varrho)\}}{\lambda_{\min}^X + \eta^2}\right) + (\sqrt{2}\chi)^{2T} \kappa^X(\eta^2) \left(\lambda_{\min}^X + \eta^2\right) \|\theta^*\|^2 + \eta^2 \|\theta^*\|^2$$
$$= O\left(\frac{\sigma^2 \max\{d, \log(T/\varrho)\}}{\lambda_{\min}^X + \eta^2}\right) + \left(1 + (\sqrt{2}\chi)^{2T} \kappa^X(\eta^2)\right) \left(\lambda_{\min}^X + \eta^2\right) \|\theta^*\|^2 - \lambda_{\min}^X \|\theta^*\|^2$$
$$= \begin{cases} O\left(\frac{\sigma^2 \max\{d, \log(T/\varrho)\}}{\gamma}\right) + \left(1 + (\sqrt{2}\chi)^{2T} \kappa^X(\eta^2)\right) \gamma \|\theta^*\|^2 - \lambda_{\min}^X \|\theta^*\|^2 & \text{if } \lambda_{\min}^X < \gamma \\ O\left(\frac{\sigma^2 \max\{d, \log(T/\varrho)\}}{\lambda_{\min}^X}\right) + (\sqrt{2}\chi)^{2T} \kappa^X(0) \lambda_{\min}^X \|\theta^*\|^2 & \text{otherwise} \end{cases}$$
$$= \begin{cases} O\left(\frac{\sqrt{T \max\{d, \log(T/\varrho), g_X\} \log(1/\delta)}(\mathrm{C}_Y^2 + \|\theta^*\|^2)}{\varepsilon} \left(1 + (\sqrt{2}\chi)^{2T} \kappa^X(\eta^2)\right)\right) & \text{if } \lambda_{\min}^X < \frac{\sqrt{T \max\{d, \log(T/\varrho), g_X\} \log(1/\delta)}}{\varepsilon} \\ O\left(\frac{T \max\{d, \log(T/\varrho), g_X\} \log(1/\delta)(\mathrm{C}_Y^2 + \|\theta^*\|^2)}{\varepsilon^2 \lambda_{\min}^X} + (\sqrt{2}\chi)^{2T} \kappa^X(0) \lambda_{\min}^X \|\theta^*\|^2\right) & \text{otherwise} \end{cases}$$

$$\le \begin{cases} O\left(\frac{\sqrt{T \max\{d, \log(T/\varrho), g_X\} \log(1/\delta)}(\mathrm{C}_Y^2 + \|\theta^*\|^2)}{\varepsilon}\right) & \text{if } \lambda_{\min}^X < \frac{\sqrt{T \max\{d, \log(T/\varrho), g_X\} \log(1/\delta)}}{\varepsilon} \\ O\left(\frac{T \max\{d, \log(T/\varrho), g_X\} \log(1/\delta)(\mathrm{C}_Y^2 + \|\theta^*\|^2)}{\varepsilon^2 \lambda_{\min}^X} + 4^{-T} \lambda_{\min}^X \|\theta^*\|^2\right) & \text{otherwise} \end{cases}$$

where all the steps are held by substitutions of the different parameters, and the last step holds since for our choice of $k$ it holds that $\chi\sqrt{\kappa^X(\eta^2)} \le \frac{1}{\sqrt{8}}$. We note that in the first case, the optimal $T$ is 1, corresponding to a constant in the non-asymptotic argument. In the second case, the minimum is upper bounded by using the choice $T^* = \log\left(\frac{(\varepsilon \lambda_{\min}^X)^2}{\max\{d, \log(1/\varrho), g_X\} \log(1/\delta)}\right)$ (up to constant factors). Substituting these minimizers and noting that they are logarithmic in terms that appear polynomial in the bound yields the stated guarantee. □

## C.2. Proof of Auxiliary Lemmas

**Proof of Lemma 5.** Recall first that by Lemma 4

$$\|b - A\theta\|^2 - \|b - A\theta^*(A, b)\| = \|\theta - \theta^*(A, b)\|_{A^\top A}^2. \tag{20}$$

Thus, we have

$$\|b - A\theta^*(A, b; a)\|^2 = \|b - A\theta^*(A, b)\|^2 + \|\theta^*(A, b; a) - \theta^*(A, b)\|_{A^\top A}^2$$
$$\|b - A\theta^*(A, b)\|^2 + a\|\theta^*(A, b)\|^2 = \|b - A\theta^*(A, b; a)\|^2 + a\|\theta^*(A, b; a)\|^2$$
$$+ \|\theta^*(A, b; a) - \theta^*(A, b)\|_{A^\top A + a\mathbb{I}_d}^2$$

where the second equality is by applying (20) with the matrix $A \leftarrow (A^\top, \sqrt{a}\mathbb{I}_d)^\top$, for which the base minimizer is $\theta^*(A, b; a)$, and with regard to the vector $\theta \leftarrow \theta^*(A, b)$. Adding both equations yields

$$a\|\theta^*(A, b)\|^2 = a\|\theta^*(A, b; a)\|^2 + \|\theta^*(A, b; a) - \theta^*(A, b)\|_{A^\top A}^2 + \|\theta^*(A, b; a) - \theta^*(A, b)\|_{A^\top A + a\mathbb{I}_d}^2.$$

The result of the lemma follows since $a\|\theta^*(A, b; a)\|^2 + \|\theta^*(A, b; a) - \theta^*(A, b)\|_{A^\top A}^2 \ge 0$.

**Proof of Lemma 6.** The proof follows directly by writing the closed-form of the minimizer in (10). The existence of this closed-form is by the condition $A^\top B^\top B A \succ 0$.

**Proof of Lemma 7.**    We first prove the next inequality:

$$a_T \leq \sum_{i=0}^{T-1} c^i \cdot e + a_0 \cdot c^T. \tag{21}$$

For $T = 1$, the desired result holds by the definition of the recursion (11). Assuming it holds for some $T > 0$, we have

$$a_{T+1} \leq c \cdot a_T + e \leq e + \sum_{i=0}^{T-1} c^{i+1} \cdot e + c^{T+1} \cdot a_0 = \sum_{i=0}^{T} c^i \cdot e + c^{T+1} \cdot a_0$$

and thus (21) holds for any $T \geq 1$ by induction. The proof is completed since for any $T \geq 1$

$$\sum_{i=0}^{T-1} c^i \leq \sum_{i=0}^{\infty} c^i = \frac{1}{1-c}.$$

**Proof of Lemma 8.**    The result follows by (Pilanci & Wainwright, 2015, Theorem 1), by the way it is applied in (Pilanci & Wainwright, 2015, Corollary 1). The final probability guarantee and the adjustment to $\chi/2$ in $\mathscr{A}_1$ is done similarly to (Pilanci & Wainwright, 2016, Lemma. 1).

**Proof of Lemma 9.**    The proof is due to (Laurent & Massart, 2000, Lem. 1), which states that

$$\mathbb{P}\left( \|\xi\|^2 > d \cdot \left( 1 + 2\sqrt{\frac{t}{d}} + 2\frac{t}{d} \right) \right) \leq e^{-t}$$

for $t > 0$. Note that $\left( \sqrt{d} + \sqrt{2t} \right)^2 \geq d \cdot \left( 1 + 2\sqrt{\frac{t}{d}} + 2\frac{t}{d} \right)$. Thus, setting $t = \log(1/\varrho)$ yields

$$\mathbb{P}\left( \|\xi\|^2 \geq \left( \sqrt{d} + \sqrt{2\log(1/\varrho)} \right)^2 \right) \leq \varrho.$$

The lemma is proved by replacing $\varrho \leftarrow \varrho^2$ and noting that increasing $d \leftarrow 2d$ reduces the probability.

**Proof of Lemma 10.**    The result is a direct consequence of the analysis in (Pilanci & Wainwright, 2016, Appendix. C), noting that $\theta_0$ takes the role of $x^t$ from (Pilanci & Wainwright, 2016, Equation. 42) and $B$ takes the role of the sketch $\mathsf{S}$. Then, the desired guarantee is established since $B \in \mathscr{A}_1(A, \chi, k) \cap \mathscr{A}_2(A, \chi, k)$ and noting that for $\chi \leq \frac{1}{2}$ it holds that $\frac{\chi/2}{1-\chi} \leq \chi$.

**Proof of Lemma 11.**    Let $D = \frac{BA}{\sqrt{k}}$ and note that for any $v \in \mathbb{R}^d$ it holds that

$$v^\top \left( D^\top D - \left( 1 - \frac{\chi}{2} \right) A^\top A \right) v = v^\top A^\top \left( \frac{1}{k} B^\top B - \left( 1 - \frac{\chi}{2} \right) \mathbb{I}_n \right) Av. \tag{22}$$

From the definition of $\mathscr{A}_1(A, \chi, k)$ and with $u \leftarrow Av$ it holds that

$$\left( 1 - \frac{\chi}{2} \right) \|Av\|^2 \leq v^\top A^\top \left( \frac{1}{k} B^\top B \right) Av \leq \left( 1 + \frac{\chi}{2} \right) \|Av\|^2. \tag{23}$$

Thus, using (23) in (22) we get that

$$v^\top A^\top \left( \frac{1}{k} B^\top B - \left( 1 - \frac{\chi}{2} \right) \mathbb{I}_n \right) Av \geq 0$$

which is further equivalent to

$$v^\top \left( D^\top D - \left( 1 - \frac{\chi}{2} \right) \mathbb{I}_n \right) v \geq 0$$

and thus proves that

$$D^\top D \succeq \left(1 - \frac{\chi}{2}\right) A^\top A$$

which further implies

$$\left(D^\top D\right)^{-1} \preceq \frac{1}{1 - \frac{\chi}{2}} \left(A^\top A\right)^{-1}. \tag{24}$$

Similarly, we note that

$$\left(D^\top D\right)^{-1} \succeq \frac{1}{1 + \frac{\chi}{2}} \left(A^\top A\right)^{-1}.$$

This proves (12). To prove (13), we first note that the Loewner order (24) is equivalent to

$$(A^\top A)^{1/2} \left(D^\top D\right)^{-1} (A^\top A)^{1/2} \preceq \frac{1}{1 - \frac{\chi}{2}} \mathbb{I}_d$$

and which, by explicitly substituting $D$, yields

$$k(A^\top A)^{1/2} \left((BA)^\top (BA)\right)^{-1} (A^\top A)^{1/2} = (A^\top A)^{1/2}(D^\top D)^{-1}(A^\top A)^{1/2} \preceq \frac{1}{1 - \frac{\chi}{2}} \mathbb{I}_d.$$

This implies that the operator norm of $k(A^\top A)^{1/2} \left((BA)^\top (BA)\right)^{-1} (A^\top A)^{1/2}$ is bounded by $\frac{1}{1 - \frac{\chi}{2}}$ since it is a symmetric positive definite matrix. As a consequence,

$$\begin{aligned}
k^2 \left\| \left((BA)^\top (BA)\right)^{-1} v \right\|_{A^\top A}^2 \\
&= \left\| k(A^\top A)^{1/2} \left((BA)^\top (BA)\right)^{-1} (A^\top A)^{1/2}(A^\top A)^{-1/2}v \right\|^2 \\
&\leq \left(1 - \frac{\chi}{2}\right)^{-2} \cdot \|(A^\top A)^{-1/2}v\|^2 \\
&= \left(1 - \frac{\chi}{2}\right)^{-2} \|v\|_{(A^\top A)^{-1}}^2 \\
&\leq \left(1 - \frac{\chi}{2}\right)^{-2} \frac{\|v\|^2}{\lambda_{\min}(A^\top A)}.
\end{aligned}$$

**Proof of Lemma 12.** We first note that by the row normalization assumption on $X$ we can use Cauchy-Schwartz to get the next upper bound for any vector $u \in \mathbb{R}^d$:

$$\|Xu\|_\infty = \max_{i \in [n]} \left| x_i^\top u \right| \leq \max_{i \in [n]} \|x_i\| \|u\| \leq \|u\|. \tag{25}$$

Moreover, for every matrix $A \succeq 0$ the next inequality holds by the definition of the operator norm

$$\|v\| = \left\| A^{-1/2} A^{1/2} v \right\| \leq \left\| A^{-1/2} \right\|_{\text{op}} \left\| A^{1/2} v \right\| \leq \frac{1}{\sqrt{\lambda_{\min}(A)}} \left\| A^{1/2} v \right\|. \tag{26}$$

Now, let $\widehat{M_t} := \frac{1}{k} X^\top B_t^\top B_t X$. Then, since $B_t \in \mathscr{A}_1(X, \chi, k)$ we can use Lemma 11 with $\chi \leftarrow \chi, A \leftarrow X$ and $B \leftarrow B_t$ to get

$$\frac{1}{1 + \frac{\chi}{2}} M^{-1} \preceq \widehat{M_t}^{-1} \preceq \frac{1}{1 - \frac{\chi}{2}} M^{-1}$$

and which further implies

$$\lambda_{\max}\left(\widehat{M_t}^{-1}\right) \leq \frac{\lambda_{\max}\left(M^{-1}\right)}{1 - \frac{\chi}{2}} = \frac{1}{\left(1 - \frac{\chi}{2}\right) \lambda_{\min}(M)}. \tag{27}$$

Our goal now is to obtain a recursive characterization of $\left\|\theta^* - \widehat{\theta}_t\left(\widehat{\theta}_{t-1}, \sigma v_{t-1}\right)\right\|$ in terms of $\left\|\theta^* - \widehat{\theta}_{t-1}\left(\widehat{\theta}_{t-2}, \sigma v_{t-2}\right)\right\|$. Then, we will use Lemma 7 to obtain an upper bound on $\left\|\theta^* - \widehat{\theta}_t\left(\widehat{\theta}_{t-1}, \sigma v_{t-1}\right)\right\|$. To that end, we use the following chain of inequalities

$$
\begin{aligned}
&\left\|\theta^* - \widehat{\theta}_t\left(\widehat{\theta}_{t-1}, \sigma v_{t-1}\right)\right\| \\
&= \left\|\theta^* - \widehat{\theta}_t\left(\widehat{\theta}_{t-1}, \vec{0}_d\right) - \sigma\widehat{M}_{t-1}^{-1}v_{t-1}\right\| \\
&\overset{(a)}{\leq} \left\|\theta^* - \widehat{\theta}_t\left(\widehat{\theta}_{t-1}, \vec{0}_d\right)\right\| + \sigma\left\|\widehat{M}_{t-1}^{-1}v_{t-1}\right\| \\
&\overset{(b)}{\leq} \left\|\theta^* - \widehat{\theta}_t\left(\widehat{\theta}_{t-1}, \vec{0}_d\right)\right\| + \frac{\sigma\left\|v_{t-1}\right\|}{\left(1 - \frac{\chi}{2}\right)\lambda_{\min}(M)} \\
&\overset{(c)}{\leq} \left\|\theta^* - \widehat{\theta}_t\left(\widehat{\theta}_{t-1}, \vec{0}_d\right)\right\| + \frac{\sigma}{1 - \frac{\chi}{2}}\cdot\frac{\sqrt{2d} + \sqrt{4\log\left(2T/\varrho\right)}}{\lambda_{\min}(M)} \\
&\overset{(d)}{\leq} \frac{1}{\sqrt{\lambda_{\min}(M)}}\left\|M^{1/2}\left(\theta^* - \widehat{\theta}_t\left(\widehat{\theta}_{t-1}, \vec{0}_d\right)\right)\right\| + \frac{\sigma}{1 - \frac{\chi}{2}}\cdot\frac{\sqrt{2d} + \sqrt{4\log\left(2T/\varrho\right)}}{\lambda_{\min}(M)} \\
&\overset{(e)}{\leq} \frac{\chi}{(2 - \chi)\sqrt{\lambda_{\min}(M)}}\left\|M^{1/2}\left(\theta^* - \widehat{\theta}_{t-1}\left(\widehat{\theta}_{t-2}, \sigma v_{t-2}\right)\right)\right\| + \frac{\sigma}{1 - \frac{\chi}{2}}\cdot\frac{\sqrt{2d} + \sqrt{4\log\left(2T/\varrho\right)}}{\lambda_{\min}(M)} \\
&\overset{(f)}{\leq} \frac{\chi}{2 - \chi}\cdot\sqrt{\frac{\lambda_{\max}(M)}{\lambda_{\min}(M)}}\left\|\theta^* - \widehat{\theta}_{t-1}\left(\widehat{\theta}_{t-2}, \sigma v_{t-2}\right)\right\| + \frac{\sigma}{1 - \frac{\chi}{2}}\cdot\frac{\sqrt{2d} + \sqrt{4\log\left(2T/\varrho\right)}}{\lambda_{\min}(M)}
\end{aligned}
$$

where (a) is by the triangle inequality, (b) is by (27), (c) is since $v_{t-1} \in \mathscr{A}_3(\varrho/2T, d)$, (d) is by (26), (e) is by Lemma 10 and (f) is since $\left\|M^{1/2}u\right\| \leq \lambda_{\max}\left(M^{1/2}\right)\|u\| = \sqrt{\lambda_{\max}(M)}\|u\|$ for all $u \in \mathbb{R}^d$. Then, since $v_t \in \mathscr{A}_3(\varrho/2T, d)$ for all $t \in \{0, 1, \ldots, T-1\}$ this upper bound holds for all $t \in \{0, 1, \ldots, T-1\}$, and we can use Lemma 7 with respect to the sequence $c_t := \left\|\theta^* - \widehat{\theta}_t\left(\widehat{\theta}_{t-1}, \sigma v_{t-1}\right)\right\|$ and with the initialization $c_0 := \|\theta^*\|$ to get

$$
\left\|\theta^* - \widehat{\theta}_t\left(\widehat{\theta}_{t-1}, \sigma v_{t-1}\right)\right\| \leq \frac{1}{1 - \frac{\chi\sqrt{\kappa(M)}}{2-\chi}}\cdot\frac{\sigma}{1 - \frac{\chi}{2}}\cdot\frac{\sqrt{2d} + \sqrt{4\log(2T/\varrho)}}{\lambda_{\min}(M)} + \left(\frac{\chi\sqrt{\kappa(M)}}{2 - \chi}\right)^t\|\theta^*\|. \tag{28}
$$

Then, we can use this to get the next guarantees on the $\ell_\infty$ norm

$$
\begin{aligned}
\left\|Y - X\widehat{\theta}_t\left(\widehat{\theta}_{t-1}, \sigma v_{t-1}\right)\right\|_\infty &\overset{(a)}{\leq} \|Y\|_\infty + \|X\theta^*\|_\infty + \left\|X\left(\theta^* - \widehat{\theta}_t\left(\widehat{\theta}_{t-1}, \sigma v_{t-1}\right)\right)\right\|_\infty \\
&\overset{(b)}{\leq} \mathsf{C}_Y + \|\theta^*\| + \left\|\theta^* - \widehat{\theta}_t\left(\widehat{\theta}_{t-1}, \sigma v_{t-1}\right)\right\| \\
&\overset{(c)}{\leq} \mathsf{C}_Y + \|\theta^*\| + \frac{1}{1 - \frac{\chi\sqrt{\kappa(M)}}{2-\chi}}\cdot\frac{\sigma}{1 - \frac{\chi}{2}}\cdot\frac{\sqrt{2d} + \sqrt{4\log(2T/\varrho)}}{\lambda_{\min}(M)} + \left(\frac{\chi\sqrt{\kappa(M)}}{2 - \chi}\right)^t\|\theta^*\|
\end{aligned}
$$

where (a) is by the triangle inequality, (b) is by (25) and the normalization assumption on $Y$, and (c) is by (28).

## D. Proof of Theorem 2

We first state the following proposition and helper lemma that we use in proving Theorem 2; Moreover, since Algorithm 5 is slightly different than the linear mixing algorithm of (Lev et al., 2025), we further prove that it also satisfies $(\varepsilon, \delta)$-DP. Throughout, similarly to Appendix C, we use the notation $\overline{X}_\eta := (X^\top, \eta\mathbb{I}_d)^\top$ and $\overline{Y} := (Y^\top, \vec{0}_d^\top)^\top$.

**Proposition 3.** *(Pilanci & Wainwright, 2015, Theorem 1) Let* $\mathsf{S} \sim \mathcal{N}(0, \mathbb{I}_{k\times n})$. *Then, there exist universal constants* $c_0, c_1, c_2$ *such that for any* $\chi \in (0, 1]$ *and* $\varrho \in (0, 1]$ *satisfying* $k\chi^2 \geq \max\left\{c_0\cdot\operatorname{rank}(X), c_2^{-1}\log\left(c_1/\varrho\right)\right\}$ *w.p. at least* $1 - \varrho$

$$
\mathcal{R}\left(\widehat{\theta}\right) \leq \left(2\chi + \chi^2\right)L\left(\theta^*\right).
$$

**Lemma 13.** *Given $X \in \mathbb{R}^{n \times d}, Y \in \mathbb{R}^n, \eta \geq 0$ and $k \in \mathbb{N}$, define*

$$\theta_{\text{Lin}} := \left( \left( (\mathsf{S}, \xi) \overline{X}_\eta \right)^\top \left( (\mathsf{S}, \xi) \overline{X}_\eta \right) \right)^{-1} \left( (\mathsf{S}, \xi) \overline{X}_\eta \right)^\top (\mathsf{S}Y + \eta\zeta)$$

*where $\mathsf{S} \sim \mathcal{N}(0, \mathbb{I}_{k \times n}), \xi \sim \mathcal{N}(0, \mathbb{I}_{k \times d}), \zeta \sim \mathcal{N}(\vec{0}_k, \mathbb{I}_k)$. Let $\theta^*$ be the OLS fit for $X$ over $Y$. Then, if $\text{rank}(\overline{X}_\eta) = d$, there exists universal constants $c_0, c_1, c_2$ such that for any $\chi \in (0, 1]$ satisfying $k\chi^2 \geq c_0 d$ w.p. at least $1 - c_1 \cdot \exp\left\{ -c_2 k\chi^2 \right\}$ it holds*

$$L\left(\theta_{\text{Lin}}\right) - L\left(\theta^*\right) \leq 3\chi \left( \eta^2 + L\left(\theta^*\right) \right) + 4\eta^2 \left\| \theta^* \right\|^2.$$

*Proof.* Define

$$\widetilde{Y}_\eta = \left( \overline{Y}^\top, \eta \right)^\top \in \mathbb{R}^{n+d+1}; \quad \widetilde{X}_\eta = \left( \overline{X}_\eta^\top, \vec{0}_d \right)^\top \in \mathbb{R}^{(n+d+1) \times d}.$$

We make two important observations. First, $\theta_{\text{Lin}}$ can be rewritten as

$$\theta_{\text{Lin}} = \underset{\theta}{\text{argmin}} \left\| (\mathsf{S}, \xi, \zeta) \left( \widetilde{Y}_\eta - \widetilde{X}_\eta \theta \right) \right\|^2. \tag{29}$$

Second, note that the minimizer of the objective

$$\|Y - X\theta\|^2 + \eta^2 \|\theta\|^2 = \left\| \widetilde{Y}_\eta - \widetilde{X}_\eta \theta \right\|^2 - \eta^2$$

which corresponds to the non-sketched version of the objective corresponding to $\theta_{\text{Lin}}$, (29), is $\theta^*(\eta^2)$. Since the elements of $\mathsf{S}, \xi, \zeta$ are i.i.d. according to $\mathcal{N}(0, 1)$, the matrix $(S, \xi, \zeta) \in \mathbb{R}^{k \times (n+d+1)}$ is distributionally equivalent to $\mathcal{N}(0, \mathbb{I}_{k \times (n+d+1)})$. Therefore, from Proposition 3 there exists constants $c_0, c_1, c_2$ such that for any $\chi \in (0, 1]$ satisfying $k\chi^2 \geq c_0 d$, it holds that

$$\left\| \widetilde{Y}_\eta - \widetilde{X}_\eta \theta_{\text{Lin}} \right\|^2 - \left\| \widetilde{Y}_\eta - \widetilde{X}_\eta \theta^*(\eta^2) \right\|^2 \leq \left( \chi^2 + 2\chi \right) \cdot \left\| \widetilde{Y}_\eta - \widetilde{X}_\eta \theta^*(\eta^2) \right\|^2 \tag{30}$$

with probability at least $1 - c_1 \cdot \exp\left\{ -c_2 k\chi^2 \right\}$. Expanding the squared norms in (30) results in

$$L(\theta_{\text{Lin}}) + \eta^2 \|\theta_{\text{Lin}}\|^2 + \eta^2 - \left( L(\theta^*(\eta^2)) + \eta^2 \left\| \theta^*(\eta^2) \right\|^2 + \eta^2 \right) \leq \left( 2\chi + \chi^2 \right) \cdot \left( L(\theta^*(\eta^2)) + \eta^2 \left\| \theta^*(\eta^2) \right\|^2 + \eta^2 \right).$$

Rearranging terms, we have

$$\begin{aligned}
L(\theta_{\text{Lin}}) - L(\theta^*) &\leq (1+\chi)^2 \cdot \left( L(\theta^*(\eta^2)) + \eta^2 \left\| \theta^*(\eta^2) \right\|^2 + \eta^2 \right) - \eta^2 \|\theta_{\text{Lin}}\|^2 - \eta^2 - L(\theta^*) \\
&\stackrel{(a)}{=} (1+\chi)^2 \cdot \left( L(\theta^*(\eta^2)) + \eta^2 \left\| \theta^*(\eta^2) \right\|^2 + \eta^2 - \eta^2 - L(\theta^*) \right) - \eta^2 \|\theta_{\text{Lin}}\|^2 \\
&\quad + (\chi^2 + 2\chi) \cdot (\eta^2 + L(\theta^*)) \\
&\stackrel{(b)}{\leq} (1+\chi)^2 \cdot \left( L(\theta^*(\eta^2)) + \eta^2 \left\| \theta^*(\eta^2) \right\|^2 - L(\theta^*) \right) + (\chi^2 + 2\chi) \cdot (\eta^2 + L(\theta^*)) \\
&\stackrel{(c)}{\leq} (1+\chi)^2 \cdot \eta^2 \|\theta^*\|^2 + (\chi^2 + 2\chi) \cdot (\eta^2 + L(\theta^*)) \\
&\stackrel{(d)}{\leq} 3\chi \cdot (\eta^2 + L(\theta^*)) + 4\eta^2 \|\theta^*\|^2.
\end{aligned}$$

In step $(a)$, we add and subtract $(\chi^2 + 2\chi) \cdot (\eta^2 + L(\theta^*))$, in step $(b)$, we cancel like term and use the non-negativity of the norm, in step $(c)$, we use the optimality of $\theta^*(\eta^2)$ as

$$L(\theta^*(\eta^2)) + \eta^2 \cdot \left\| \theta^*(\eta^2) \right\|^2 \leq L(\theta) + \eta^2 \cdot \|\theta\|^2 \qquad \forall\, \theta \in \mathbb{R}^d$$

and specifically for $\theta \leftarrow \theta^*$, which corresponds to

$$L(\theta^*(\eta^2)) + \eta^2 \cdot \left\| \theta^*(\eta^2) \right\|^2 - L(\theta^*) \leq \eta^2 \cdot \|\theta^*\|^2,$$

and in step (d), we use the fact that $\chi^2 + 2\chi \leq 3\chi$ and $(1+\chi)^2 \leq 4$ which holds since $\chi \in (0, 1]$. $\square$

**Lemma 14.** *Algorithm 5 is $(\varepsilon, \delta)$-DP with respect to $(X, Y)$.*

*Proof.* The proof follows similarly to the privacy proof from (Lev et al., 2025). To that end, note that the algorithm involves one private release of the minimal eigenvalue with the initial $k$, and another sketching step, with parameter $\widetilde{k}$ or $k$ and with noise scale of either $0$ or $\sqrt{\gamma \left( \mathrm{C}_X^2 + \mathrm{C}_Y^2 \right) - \widetilde{\lambda}}$. Following (Lev et al., 2025), the privacy guarantees of the eigenvalue test are $\left( \frac{\sqrt{2k \log(3.75/\delta)}}{\gamma}, \frac{\delta}{3} \right)$ and furthermore $\mathbb{P}\left( \lambda_{\min}^{XY} \geq \widetilde{\lambda} \right) \geq 1 - \frac{\delta}{3}$. Then, we note conditioned on the event $\left\{ \widetilde{\lambda} \leq \lambda_{\min}^{XY} \right\}$, the RDP guarantees of the sketching step are $\varphi\left( \alpha; k, \gamma \right)$ or $\varphi\left( \alpha; \widetilde{k}, \lambda_{\min}^{XY} \right)$. Thus, similarly to (Lev et al., 2025), the overall guarantees are $(\widehat{\varepsilon}, \widehat{\delta})$-DP where

$$\widehat{\delta} = \mathbb{P}\left( \widetilde{\lambda} \leq \lambda_{\min}^{XY} \right) + \frac{\delta}{3} + \frac{\delta}{3} \leq \delta$$

with the first two terms corresponds to the $\delta/3$ from the test of the minimal eigenvalue and the probability of the event $\left\{ \widetilde{\lambda} \leq \lambda_{\min}^{XY} \right\}$ and the last $\delta/3$ term is by the contribution of the sketching step, and where $\widehat{\varepsilon}$ is either

$$\widehat{\varepsilon} = \frac{\sqrt{2k \log(3.75/\delta)}}{\gamma} + \min_{1 < \alpha < \gamma} \left\{ \varphi(\alpha; k, \gamma) + \frac{\log(3/\delta) + (\alpha - 1)\log(1 - 1/\alpha) - \log(\alpha)}{\alpha - 1} \right\}$$

or

$$\widehat{\varepsilon} = \frac{\sqrt{2k \log(3.75/\delta)}}{\gamma} + \min_{1 < \alpha < \lambda_{\min}^{XY}} \left\{ \varphi(\alpha; \widetilde{k}, \lambda_{\min}^{XY}) + \frac{\log(3/\delta) + (\alpha - 1)\log(1 - 1/\alpha) - \log(\alpha)}{\alpha - 1} \right\}$$

$$\leq \frac{\sqrt{2\widetilde{k} \log(3.75/\delta)}}{\gamma} + \min_{1 < \alpha < \widetilde{\lambda}} \left\{ \varphi(\alpha; \widetilde{k}, \widetilde{\lambda}) + \frac{\log(3/\delta) + (\alpha - 1)\log(1 - 1/\alpha) - \log(\alpha)}{\alpha - 1} \right\}$$

and where the second inequality is since $k \leq \widetilde{k}$ and since $\widetilde{\lambda} \leq \lambda_{\min}^{XY}$ and the since the function $\varphi(\alpha; k, \cdot)$ is monotonically decreasing in its third argument. The first quantity is less than $\varepsilon$ by our choice of $\gamma$, and the second quantity is less than $\varepsilon$ by our choice of $\widetilde{k}$. $\qquad \square$

### D.1. Proof of Theorem 2

*Proof.* The estimator obtained through Algorithm 5 takes the form

$$\theta_{\mathrm{Lin}} := \left( (\mathsf{S}X + \widetilde{\kappa}\xi)^\top (\mathsf{S}X + \widetilde{\kappa}\xi) \right)^{-1} (\mathsf{S}X + \widetilde{\kappa}\xi)^\top (\mathsf{S}Y + \widetilde{\kappa}\zeta) \tag{31}$$

where $\widetilde{\kappa}$ is set internally to either $0$ or $\sqrt{\gamma - \widetilde{\lambda}}$ and $\mathsf{S} \sim \mathcal{N}(0, \mathbb{I}_{m \times n}), \xi \sim \mathcal{N}(0, \mathbb{I}_{m \times d}), \zeta \sim \mathcal{N}(0, \mathbb{I}_m)$ and with $m$ being either $k$ or $\widetilde{k}$. Throughout, we refer to the actual noise level set by the algorithm by $\widetilde{\eta}$, and to $\widetilde{\kappa}$ as a general noise level used to make calculations with estimators of the form (31). Moreover, note that by the calculation of $\widetilde{\lambda}_{\min}$ it holds that $\mathbb{P}\left( \widetilde{\lambda} \leq \lambda_{\min}^{XY} \right) \geq 1 - \frac{\varrho}{4}$.

Let $k = \max \left\{ c_0 d, \frac{1}{c_2} \log\left( \frac{2c_1}{\varrho} \right) \right\}$. From Lemma 13, we know that the excess empirical risk achieved by $\theta_{\mathrm{Lin}}$ with a general noise level $\widetilde{\kappa}$ is upper-bounded as

$$L(\theta_{\mathrm{Lin}}) - L(\theta^*) \leq 3\chi(\widetilde{\kappa}^2 + L(\theta^*)) + 4\widetilde{\kappa}^2 \|\theta^*\|^2 \tag{32}$$

with probability at least $1 - c_1 \cdot \exp\left\{ -c_2 k \chi^2 \right\}$ provided $\chi \in (0, 1]$ and $k$ satisfy $k\chi^2 \geq c_0 d$ for universal constants $c_0, c_1, c_2$. Note that by our choice of $k$ it holds that $k \geq c_2^{-1} \log\left( \frac{2c_1}{\varrho} \right)$. Furthermore, note that both of these conditions on $k$ allow for the guarantees to hold when $\chi$ is in the range

$$\sqrt{\frac{\max \left\{ c_0 d, \frac{1}{c_2} \log\left( \frac{2c_1}{\varrho} \right) \right\}}{k}} \leq \chi \leq 1 \tag{33}$$

and the tightest guarantee we obtain is thus with the lowest possible $\chi$, and which by these choices holds w.p. at least $1 - \frac{\varrho}{2}$. Given this, our goal is to simplify (32) using the values for $\widetilde{\kappa}^2$ and $\chi$ that are used in the different cases of the algorithm.

*Step 1: Utility For a General $\widetilde{\kappa}$:* We first rewirte (32) in the next form

$$L\left(\theta_{\mathrm{Lin}}\right) - L\left(\theta^*\right) \le 3\chi\left(\widetilde{\kappa}^2 + \widetilde{\lambda} + L\left(\theta^*\right) - \widetilde{\lambda}\right) + 4\widetilde{\kappa}^2 \left\|\theta^*\right\|^2 \tag{34}$$

and we recall that this guarantee holds w.p. at least $1 - \frac{\varrho}{2}$.

*Step 2: Establishing Privacy Parameters:* For the true level of the additive noise $\widetilde{\eta}$, following Proposition 2 it holds that for any sketch size $k$ the next connection between $\widetilde{\lambda}, k, \mathsf{C}_Y$ and $(\varepsilon, \delta)$ needs to be satisfied:

$$\frac{2\sqrt{2k\log(1/\delta)}}{\widetilde{\eta}^2 + \widetilde{\lambda}}\left(1 + \mathsf{C}_Y^2\right) + \frac{k}{2(\widetilde{\eta}^2 + \widetilde{\lambda})^2}\left(1 + \mathsf{C}_Y^2\right)^2 \le \varepsilon \tag{35}$$

and from which we calculate the minimal $\widetilde{\eta}^2$ that satisfy this equation, and as such ensures that the pair $(\mathsf{S}X + \widetilde{\eta}\xi, \mathsf{S}Y + \widetilde{\eta}\zeta)$ is $(\varepsilon, \delta)$-DP [5]. We note that (35) forms a quadratic equation in $\frac{\sqrt{k}}{\widetilde{\eta}^2 + \widetilde{\lambda}}(1 + \mathsf{C}_Y^2)$. Solving this equation for $\widetilde{\eta}$ by finding the roots of this quadratic equation implies the next condition:

$$\widetilde{\eta}^2 + \widetilde{\lambda} \ge \frac{\sqrt{k}\left(1 + \mathsf{C}_Y^2\right)}{2\sqrt{2\log(1/\delta)}\left(-1 + \sqrt{1 + \frac{\varepsilon}{4\log(1/\delta)}}\right)}. \tag{36}$$

We note that the algorithm solves for the exact $\gamma$ by minimizing the closed-form expression (8), thus the $\widetilde{\eta}^2 + \widetilde{\lambda}$ obtained from (35) is an upper bound on the exact value that is obtained in the algorithm. However, since our goal is to obtain an upper bound, and since (34) is monotonically increasing in the noise level $\widetilde{\kappa}$, we will use this upper bound in our analysis. In particular, we will be carrying out the analysis if (36) holds with equality in both of the settings of the algorithm, and the resulting guarantee will serve as an upper bound on the actual error of the algorithm.

*Step 3: Choice of $k$:* Following the setting of Algorithm 5 and the initial choice of $k$, using the connection between $\gamma, \varepsilon, \delta$ from (Lev et al., 2025) we note that

$$\widetilde{k} \ge \max\left\{c_0 d, \frac{1}{c_2}\log\left(\frac{2c_1}{\varrho}\right), \left(\frac{\widetilde{\lambda}\sqrt{2\log(1/\delta)}\left(-1 + \sqrt{1 + \frac{\varepsilon}{4\log(1/\delta)}}\right)}{1 + \mathsf{C}_Y^2}\right)^2\right\}.$$

*Step 4: Substituting Privacy Parameters in (34):* Using the previous value of $\widetilde{k}$, together with (33) and (36) we get that whenever $\widetilde{\eta} > 0$ it holds that

$$\chi\left(\widetilde{\eta}^2 + \widetilde{\lambda}\right) \le \frac{\sqrt{\max\left\{c_0 d, \frac{1}{c_2}\log\left(\frac{2c_1}{\varrho}\right)\right\}}\left(1 + \mathsf{C}_Y^2\right)}{2\sqrt{2\log(1/\delta)}\left(-1 + \sqrt{1 + \frac{\varepsilon}{4\log(1/\delta)}}\right)}.$$

---

[5] note that this further implies that, for large enough $\widetilde{\lambda}$, the resulting $\widetilde{\eta}^2$ can be set to 0

Substituting this back into (34) yields

$$
\begin{aligned}
L\left(\theta_{\mathrm{Lin}}\right)-L\left(\theta^{*}\right) &\le 3\chi\left(\widetilde{\eta}^{2}+\widetilde{\lambda}+L\left(\theta^{*}\right)-\widetilde{\lambda}\right)+4\widetilde{\eta}^{2}\left\|\theta^{*}\right\|^{2} \\
&= 3\chi\left(\widetilde{\eta}^{2}+\widetilde{\lambda}\right)\left(1+\frac{L(\theta^{*})-\widetilde{\lambda}}{\widetilde{\eta}^{2}+\widetilde{\lambda}}\right)+4\widetilde{\eta}^{2}\left\|\theta^{*}\right\|^{2} \\
&= 3\chi\left(\widetilde{\eta}^{2}+\widetilde{\lambda}\right)\left(\frac{L(\theta^{*})+\widetilde{\eta}^{2}}{\widetilde{\eta}^{2}+\widetilde{\lambda}}\right)+4\widetilde{\eta}^{2}\left\|\theta^{*}\right\|^{2} \\
&\le \frac{3\sqrt{\max\left\{c_0 d,\frac{1}{c_2}\log\left(\frac{2c_1}{\varrho}\right)\right\}}(1+\mathbf{C}_Y^2)}{2\sqrt{2\log(1/\delta)}\left(-1+\sqrt{1+\frac{\varepsilon}{4\log(1/\delta)}}\right)}\left(\frac{L(\theta^{*})+\widetilde{\eta}^{2}}{\widetilde{\eta}^{2}+\widetilde{\lambda}}\right)+4\widetilde{\eta}^{2}\left\|\theta^{*}\right\|^{2} \\
&\le O\left(\frac{\sqrt{\max\{d,\log(1/\varrho)\}\log(1/\delta)}\left(1+\mathbf{C}_Y^2\right)}{\varepsilon}\left(\frac{L\left(\theta^{*}\right)}{\widetilde{\eta}^{2}+\widetilde{\lambda}}+\frac{\widetilde{\eta}^{2}}{\widetilde{\eta}^{2}+\widetilde{\lambda}}\right)+\widetilde{\eta}^{2}\left\|\theta^{*}\right\|^{2}\right).
\end{aligned}
$$

Moreover, starting from (34), we note that for the case where $\widetilde{\eta}^{2}=0$ it further holds that

$$
\begin{aligned}
L\left(\theta_{\mathrm{Lin}}\right)-L\left(\theta^{*}\right) &\le 3\chi L(\theta^{*}) \\
&\le 3\frac{\sqrt{\max\left\{c_0 d,\frac{1}{c_2}\log\left(\frac{2c_1}{\varrho}\right)\right\}}}{\sqrt{2\log(1/\delta)}\left(-1+\sqrt{1+\frac{\varepsilon}{4\log(1/\delta)}}\right)}(1+\mathbf{C}_Y^2)\frac{L(\theta^{*})}{\widetilde{\lambda}} \\
&\le O\left(\frac{\sqrt{\max\{d,\log(1/\varrho)\}\log(1/\delta)}\left(1+\mathbf{C}_Y^2\right)}{\varepsilon}\cdot\frac{L\left(\theta^{*}\right)}{\widetilde{\lambda}}\right).
\end{aligned}
$$

*Step 5: Final Guarantees With $\lambda_{\min}^{XY}$:*   It remains to connect these guarantees to quantities that depend on $\lambda_{\min}^{XY}$ rather than $\widetilde{\lambda}$. First, we note that the noise added in the algorithm ensures that the condition $\mathrm{rank}\left(\widetilde{X}_{\widetilde{\eta}}\right)=d$ is satisfied whenever $\widetilde{\lambda}\le\lambda_{\min}^{XY}$, and by our construction it holds that $\mathbb{P}\left(\widetilde{\lambda}\le\lambda_{\min}^{XY}\right)\ge 1-\frac{\varrho}{4}$. Furthermore, note that it holds that

$$
\frac{2\sqrt{2k\log(1/\delta)}}{\gamma}\left(1+\mathbf{C}_Y^2\right)+\frac{k}{2\gamma^2}\left(1+\mathbf{C}_Y^2\right)^2\le\varepsilon \tag{37}
$$

whenever $\widetilde{\lambda}<\gamma$ and

$$
\frac{2\sqrt{2k\log(1/\delta)}}{\widetilde{\lambda}}\left(1+\mathbf{C}_Y^2\right)+\frac{k}{2\widetilde{\lambda}^2}\left(1+\mathbf{C}_Y^2\right)^2\le\varepsilon
$$

whenever $\widetilde{\lambda}\ge\gamma$ where $\gamma$ is defined in Line 1 of Algorithm 5. Moreover, note that it always holds that $\widetilde{\eta}^{2}+\widetilde{\lambda}\ge\gamma$ with equality whenever $\widetilde{\eta}>0$. Thus, we get similar guarantees while we replace $\widetilde{\eta}^{2}+\widetilde{\lambda}$ with $\max\left\{\widetilde{\lambda},\gamma\right\}$ and then further substituting $\gamma=\Theta\left(\frac{\sqrt{k\log(1/\delta)}(1+\mathbf{C}_Y^2)}{\varepsilon}\right)$, which is implied by (37). Finally, the final guarantees hold since, for our choice of $\tau$ and our construction of $\widetilde{\lambda}$

$$
\begin{aligned}
\mathbb{P}\left(\widetilde{\lambda}\ge\frac{\lambda_{\min}^{XY}}{2}\right) &= \mathbb{P}\left(\mathsf{z}\ge\tau-\frac{\lambda_{\min}^{XY}}{2\eta}\right) \\
&= \mathbb{P}\left(\mathsf{z}\ge\tau-\frac{\lambda_{\min}^{XY}\sqrt{k}}{2\gamma}\right)
\end{aligned}
$$

which holds w.p. at least $1-\frac{1}{2}\exp\left\{-\frac{\tau^2}{8}\right\}$ whenever $\lambda_{\min}^{XY}\ge\frac{3\tau\gamma}{\sqrt{k}}$, which, under our choice of $\tau$ and whenever $\varrho\le\delta$ included in the regime in which $\lambda_{\min}^{XY}\ge\gamma_{\mathrm{m}}\left(1+\mathbf{C}_Y^2\right)$ up to a constant factor. Substituting our choice of $\tau$, we note that

this further ensures that $\mathbb{P}\left(\frac{\lambda_{\min}^{XY}}{2} \leq \widetilde{\lambda} \leq \lambda_{\min}^{XY}\right) \geq 1 - \frac{\varrho}{2}$. Thus, we can replace $\widetilde{\lambda}$ with $\frac{\lambda_{\min}^{XY}}{2}$ in the guarantees. The proof is finished by dropping constant factors, and since the entire guarantee holds under the events ensuring that the guarantees with respect to sketch holds, together with the event $\left\{\frac{\lambda_{\min}^{XY}}{2} \leq \widetilde{\lambda} \leq \lambda_{\min}^{XY}\right\}$. By the union bound, this holds with probability at least $1 - \varrho$. $\qquad\square$

## E. Optimizing $k$ for Gaussian Sketch and Solve

We note that the final bound derived in Appendix D can be optimized over $\gamma$ in a data-dependent manner by tuning $k$. In particular, we note that for a general value of $k$, and by assuming that we have exact access to $\lambda_{\min}^{XY}$ (namely, replacing $\widetilde{\lambda}$ with this $\lambda_{\min}^{XY}$), similar developments lead to the final bound

$$
\mathcal{R}\left(\widehat{\theta}\right) \leq \begin{cases} O\left(\gamma_{\mathrm{m}} \cdot \left(1 + \mathrm{C}_Y^2\right) + \left(\gamma - \lambda_{\min}^{XY}\right)\|\theta^*\|^2 + \frac{\gamma_{\mathrm{m}}\left(L(\theta^*) - \lambda_{\min}^{XY}\right)}{\gamma}\left(1 + \mathrm{C}_Y^2\right)\right) & \text{if } \lambda_{\min}^{XY} < \gamma \\ O\left(\gamma_{\mathrm{m}} \cdot \left(1 + \mathrm{C}_Y^2\right) \cdot \frac{L(\theta^*)}{\lambda_{\min}^{XY}}\right) & \text{otherwise} \end{cases}
$$

which holds w.p. at least $1 - \varrho$ and where $\gamma$ corresponds to $\widetilde{\eta}^2 + \lambda_{\min}^{XY}$. Minimizing this under $k \geq \max\left\{c_0 d, \frac{1}{c_2}\log\left(\frac{2c_1}{\varrho}\right)\right\}$ (which corresponds to $\gamma > \gamma_{\mathrm{m}}\left(1 + \mathrm{C}_Y^2\right)$) yields

$$
\gamma^* = \max\left\{\lambda_{\min}^{XY}, \frac{1}{\|\theta^*\|}\sqrt{\gamma_{\mathrm{m}}\left(1 + \mathrm{C}_Y^2\right)\left(L\left(\theta^*\right) - \lambda_{\min}^{XY}\right)}, \gamma_{\mathrm{m}}\left(1 + \mathrm{C}_Y^2\right)\right\}.
$$

We note that whenever $\gamma^* = \frac{1}{\|\theta^*\|}\sqrt{\gamma_{\mathrm{m}}\left(1 + \mathrm{C}_Y^2\right)\left(L\left(\theta^*\right) - \lambda_{\min}^{XY}\right)}$ (which is the regime in which the optimization is different than (7)), the minimum is

$$
\gamma_{\mathrm{m}}\left(1 + \mathrm{C}_Y^2\right)\left(1 + \|\theta^*\|\sqrt{\frac{L(\theta^*) - \lambda_{\min}^{XY}}{\gamma_{\mathrm{m}}\left(1 + \mathrm{C}_Y^2\right)}}\right) + \gamma^*\|\theta^*\|^2 - \lambda_{\min}^{XY}\|\theta^*\|^2.
$$

This expression contains $\gamma_{\mathrm{m}}\left(1 + \mathrm{C}_Y^2\right) + \gamma^*\|\theta^*\|^2$, which is greater than $\gamma_{\mathrm{m}}\left(1 + \mathrm{C}_Y^2\right)\left(1 + \|\theta^*\|^2\right)$, and the additional term $\gamma^*\|\theta^*\|^2$ that grows with $L(\theta^*)$. Moreover, the entire bound is still monotonically decreasing in $\lambda_{\min}^{XY}$, and contain the irreducible term $\frac{L(\theta^*)}{\lambda_{\min}^{XY}\left(1 + \|\theta^*\|^2\right)}$. However, choosing $k$ to track $\gamma^*$ would require a data-dependent (and thus private) tuning rule and spending of an additional privacy budget. Since optimizing $\gamma$ does not change the asymptotic dependence of the term $\gamma_{\mathrm{m}}\left(1 + \mathrm{C}_Y^2\right)\left(1 + \|\theta^*\|^2\right)$—or our qualitative comparisons between the algorithms—we adopt the fixed choice $k = \max\left\{c_0 d, \frac{1}{c_2}\log\left(\frac{2c_1}{\varrho}\right)\right\}$ for simplicity and clarity.

## F. Relationship Between $\lambda_{\min}^X, \lambda_{\min}^{XY}$ and $L\left(\theta^*\right)$

We now prove the next proposition, which we use for comparing the different schemes

**Proposition 4.** *Let $X \in \mathbb{R}^{n \times d}, Y \in \mathbb{R}^n$, and $\theta^*$ as in (1). Then, it holds that $\lambda_{\min}^{XY} \leq \min\left\{\lambda_{\min}^X, \frac{L(\theta^*)}{1 + \|\theta^*\|^2}\right\}$.*

*Proof.* First, we show that $\lambda_{\min}^{XY} \leq \lambda_{\min}^X$. In particular, this holds since, given the unit-norm eigenvector $v$ of the matrix $X^\top X$ that corresponds to $\lambda_{\min}^X$, then

$$
\begin{aligned}
\lambda_{\min}^{XY} &= \min_{u:\|u\|\leq 1} u^\top((X, Y)^\top(X, Y))u \\
&\leq (v^\top, 0)((X, Y)^\top(X, Y))\begin{pmatrix} v \\ 0 \end{pmatrix} \\
&= v^\top(X^\top X)v \\
&= \lambda_{\min}^X.
\end{aligned}
$$

To prove the inequality $\lambda_{\min}^{XY} \le \frac{L(\theta^*)}{1+\|\theta^*\|^2}$, we note that it holds

$$
\begin{aligned}
\lambda_{\min}^{XY} &= \min_{u:\|u\|\le 1} u^\top((X,Y)^\top(X,Y))u \\
&\le \frac{1}{1+\|\theta^*\|^2}(-(\theta^*)^\top, 1)((X,Y)^\top(X,Y))\begin{pmatrix}-\theta^*\\1\end{pmatrix} \\
&= \frac{L(\theta^*)}{1+\|\theta^*\|^2}.
\end{aligned}
$$

$\square$

## G. Algorithms

---

**Algorithm 2** Calibrate Mixing Noise (Lev et al., 2025)

---

**Input:** Dataset $X \in \mathbb{R}^{n\times d}$, row bound $\mathsf{C}_X$, parameters $\gamma, \tau, \eta$.
1: Set $\widetilde{\lambda} = \max\left\{\lambda_{\min}^X - \eta\mathsf{C}_X^2(\tau - \mathsf{z}), 0\right\}$ for $\mathsf{z} \sim \mathcal{N}(0,1)$ and $\lambda_{\min}^X = \lambda_{\min}\left(X^\top X\right)$.
2: **Output:** $\widetilde{\eta} \leftarrow \sqrt{\max\left\{\gamma\mathsf{C}_X^2 - \widetilde{\lambda}, 0\right\}}$

---

**Algorithm 3** Gaussian Mixing Mechanism (Lev et al., 2025)

---

**Input:** Dataset $X \in \mathbb{R}^{n\times d}$, row bound $\mathsf{C}_X$, parameters $k, \gamma, \tau, \eta$.
1: $\widetilde{\eta} \leftarrow \texttt{CalibrateMixingNoise}(X, \mathsf{C}_X, \gamma, \tau, \eta)$  (Algorithm 2)
2: Sample $\mathsf{S} \sim \mathcal{N}(0, \mathbb{I}_{k\times n})$, $\xi \sim \mathcal{N}(0, \mathbb{I}_{k\times d})$
3: **Output:** $\mathsf{S}X + \widetilde{\eta}\mathsf{C}_X\xi$

---

**Algorithm 4** Linear Mixing (Lev et al., 2025)

---

**Input:** Dataset $(X, Y) \in \{\mathbf{A}_1, \mathbf{A}_2\}$, privacy parameters $(\varepsilon, \delta)$, sketch size $k$, failure probability $\varrho$.
1: Find the smallest $\gamma > \frac{5}{2}$ such that $\widetilde{\varepsilon}(\eta, \gamma, k, \delta)$ ((8)) is less than $\varepsilon$, while setting $\eta = \gamma/\sqrt{k}$.
2: Set $\left[\widetilde{X}, \widetilde{Y}\right] = \texttt{GaussianMixingMechanism}\left([X, Y], \sqrt{\mathsf{C}_X^2 + \mathsf{C}_Y^2}, k, \gamma, \sqrt{2\log\left(\max\left\{\frac{3}{\delta}, \frac{2}{\varrho}\right\}\right)}, \eta\right)$.
3: **Output:** $\widehat{\theta}_{\mathrm{mix}} := \left(\widetilde{X}^\top\widetilde{X}\right)^{-1}\widetilde{X}^\top\widetilde{Y}$.

---

**Algorithm 5** Linear Mixing: Dynamic $k$

---

**Input:** Dataset $(X, Y) \in \{\mathbf{A}_1, \mathbf{A}_2\}$, parameters $(\varepsilon, \delta)$, initial sketch size $k$, failure probability $\varrho$, bounds $(\mathsf{C}_X, \mathsf{C}_Y)$.
1: Set $\tau \leftarrow \sqrt{8\log\left(\max\left\{\frac{3}{\delta}, \frac{4}{\varrho}\right\}\right)}$.
2: Find the smallest $\gamma > \frac{5}{2}$ such that $\widetilde{\varepsilon}(\eta, \gamma, k, \delta)$ ((8)) is less than $\varepsilon$, while setting $\eta = \gamma/\sqrt{k}$.
3: Set $\widetilde{\lambda} \leftarrow \max\left\{\lambda_{\min}^{XY} - \eta\left(\mathsf{C}_X^2 + \mathsf{C}_Y^2\right)(\tau - \mathsf{z}), 0\right\}$ for $\mathsf{z} \sim \mathcal{N}(0,1)$, $\lambda_{\min}^{XY} = \lambda_{\min}((X,Y)^\top(X,Y))$.
4: **if** $\widetilde{\lambda} \ge \gamma$: **then**
5:     Find the largest $\widetilde{k} \ge k$ such that $\widetilde{\varepsilon}(\eta, \widetilde{\lambda}, \widetilde{k}, \delta)$ ((8)) is less than $\varepsilon$, while setting $\eta = \gamma/\sqrt{\widetilde{k}}$.
6:     Set $[\widetilde{X}, \widetilde{Y}] \leftarrow \mathsf{S}(X, Y)$ for $\mathsf{S} \sim \mathcal{N}(0, \mathbb{I}_{\widetilde{k}\times n})$.
7: **else**
8:     Set $[\widetilde{X}, \widetilde{Y}] \leftarrow \mathsf{S}(X, Y) + \sqrt{\gamma\left(\mathsf{C}_X^2 + \mathsf{C}_Y^2\right) - \widetilde{\lambda}} \cdot \xi$ for $\mathsf{S} \sim \mathcal{N}(0, \mathbb{I}_{k\times n})$ and $\xi \sim \mathcal{N}(0, \mathbb{I}_{k\times(d+1)})$.
9: **Output:** $\widehat{\theta}_{\mathrm{dmix}} := \left(\widetilde{X}^\top\widetilde{X}\right)^{-1}\widetilde{X}^\top\widetilde{Y}$.

---

In the next version of AdaSSP, we have included an additional $\sqrt{2}$ factor in the noise that is used throughout the algorithm. This is needed for ensuring that the private quantities are indeed $(\varepsilon/3, \delta/3)$-DP (see, for example, (Dwork et al., 2014a, Appendix. A)). In our experiments, and as described in Appendix H, we have used the tight formulation from (Balle & Wang, 2018).

---

**Algorithm 6** AdaSSP (Wang, 2018)

---

**Input:** Dataset $(X, Y)$; parameters $\varepsilon, \delta$; bounds: $\max_{i \in [n]} \|x_i\|^2 \leq \mathrm{C}_X^2, \max_{i \in [n]} |y_i|^2 \leq \mathrm{C}_Y^2$.

1: Calculate the minimum eigenvalue $\lambda_{\min}^X = \lambda_{\min}\left(X^\top X\right)$.

2: Privately release $\widetilde{\lambda}_{\min} = \max\left\{\lambda_{\min}^X + \frac{\sqrt{2\log(6/\delta)}\mathrm{C}_X^2}{\varepsilon/3}\left(\mathsf{z} - \sqrt{2\log(6/\delta)}\right), 0\right\}$ where $\mathsf{z} \sim \mathcal{N}(0,1)$.

3: Set $\widetilde{\lambda} = \max\left\{0, \frac{\sqrt{2d\log(6/\delta)\log(2d^2/\varrho)}\mathrm{C}_X^2}{\varepsilon/3} - \widetilde{\lambda}_{\min}\right\}$.

4: Release $\widetilde{X^\top X} = X^\top X + \frac{\sqrt{2\log(6/\delta)}\mathrm{C}_X^2}{\varepsilon/3}\xi$ for $\xi \sim \mathcal{N}_{\mathrm{sym}}(0, \mathbb{I}_d)$.

5: Release $\widetilde{X^\top Y} = X^\top Y + \frac{\sqrt{2\log(6/\delta)}\mathrm{C}_X\mathrm{C}_Y}{\varepsilon/3}\zeta$ for $\zeta \sim \mathcal{N}(\vec{0}_d, \mathbb{I}_d)$.

6: **return** $\widetilde{\theta} \leftarrow \left(\widetilde{X^\top X} + \widetilde{\lambda}\mathbb{I}_d\right)^{-1}\widetilde{X^\top y}$

---

**Algorithm 7** DP-GD

---

**Input:** Dataset $(X, Y)$; clipping threshold C; noise scale $\sigma > 0$; learning rate $b > 0$; number of iterations $T \in \mathbb{N}$; initialization $\widehat{\theta}_0 \in \mathbb{R}^d$.

1: **for** $t = 0, \ldots, T - 1$ **do**:

2: $\quad \overline{G}_t \leftarrow \frac{1}{n}\sum_{i=1}^n\left(-x_i\left(y_i - x_i^\top\widehat{\theta}_{t-1}\right)\right) \cdot \min\left\{1, \frac{\mathrm{C}}{\|x_i(y_i - x_i^\top\widehat{\theta}_t)\|}\right\}$;

3: $\quad$ Sample $\mathsf{Z}_t \sim \mathcal{N}(\vec{0}_d, \sigma^2\mathbb{I}_d)$;

4: $\quad$ Update $\widehat{\theta}_{t+1} \leftarrow \widehat{\theta}_t + b\left(\mathsf{Z}_t - \overline{G}_t\right)$

$\quad$ **return** $\widehat{\theta}_T$.

---

# H. Experimental Details

The experiments were run on 12th Gen Intel(R) Core(TM) i7-1255U. We used thirty-three different real datasets, all of which are from the UCI machine learning repository. For all the datasets, we have used a random train-test split of 80%/20% for generating a train and a test set. The plots throughout the paper are for the normalized (divided by $n$) excess empirical risk, as this quantity corresponds to our analysis. However, similar empirics hold for the excess test risk as well. For all the datasets, we have dropped any rows that contain missing values. In all cases, we normalized the training data so that the maximum $\ell_2$-norm of any training sample was 1 (namely, $\|x_i\|^2 \leq 1$ and $|y_i| \leq 1$ for all $i = 1, \ldots, n$, so $\mathrm{C}_X = \mathrm{C}_Y = 1$). The test data was scaled using the same normalization factor as the training data.

The baseline (non-private) estimator was computed as the minimum norm solution $\theta^*$ using the function `sklearn.linear_model.LinearRegression`. We report the normalized mean squared error (MSE) for the train set, computed as the squared error in predicting $y_i$ via $x_i^\top\widehat{\theta}$, averaged over the train set, and we call this quantity train MSE. Our plots contain the difference between this quantity and the base error $\|Y - X\theta^*\|^2$, normalized by the sample size $n$. The results are averaged over 500 independent trials, and we report both the empirical means and 95% confidence intervals, calculated via $\pm 1.96 \cdot \frac{\mathrm{std}}{\sqrt{\#\mathrm{runs}}}$.

Throughout the entire set of experiments, we have fixed the failure probability (for Hessian mixing, linear mixing, and for the AdaSSP algorithm) on $\varrho = \frac{\delta}{10}$, so the failure probability of the system is effectively only slightly larger than the additive slack from the DP definition, $\delta$. Moreover, we set $\delta = \frac{1}{n^2}$. For the linear mixing scheme, the value $2.5 \cdot \max\{d, \log(2/\varrho)\}$ was picked by performing a grid search over the set of multipliers $\{1.25, 2.5, 5.0, 7.5, 10.0\}$ applied to $\max\{d, \log(2/\varrho)\}$ at the target privacy level $\varepsilon = 0.5$, across all real datasets for which we expect our scheme to outperform the AdaSSP scheme (namely, datasets with low residual, low minimal eigenvalue, and $\mathrm{C}_Y^2 \approx \|\theta^*\|^2$). The choice $k = 2.5 \cdot \max\{d, \log(2/\varrho)\}$

delivered the best performance on this set of real datasets, and we adopt it throughout. For the iterative Hessian mixing, the value $k = 6 \cdot \max\{d, \log(4T/\varrho)\}$ was picked following similar rule of thumb choice presented in (Pilanci & Wainwright, 2016, Section. 3.1), and the value $T = 3$ was picked by finding the $T$ that minimizes the expression $3^{-T}n + \gamma_{\mathrm{hess}} C_{\mathrm{hess}}$ for a target sample size of $n = 2^{10}$, dimension $d = 2^7$ and with the choice $C_{\mathrm{hess}} = 2C_Y, \delta = 1/n^2, \varrho = \delta/10$ and such that $\log(1/\varrho) \leq d$ and $\frac{\sqrt{\max\{d, \log(1/\varrho)\} \log(1/\delta)}}{\lambda_{\min}^X} \geq 1$ and for $\varepsilon = 0.5$. Moreover, for the IHM, we further observed that omitting the term $-\eta^2 \widehat{\theta}_t$ from the computations of $\widetilde{G}_t$ improves the performance of the algorithm. Since this does not affect the privacy analysis of our algorithm, we omit this term in the empirical evaluations part.

For the linear-mixing scheme, setting $k = 2.5 \cdot \max\{d, \log(2/\varrho)\}$ resulted in a monotonically increasing (in $\varepsilon$) train MSE on the forest, pumadyn32nm, and tamielectric datasets, corresponding to datasets in which $L(\theta^*) \approx \|Y\|^2$ (and correspondingly $C_Y^2 \gg \|\theta^*\|^2$). This hints that this choice for $k$ is sub-optimal for these datasets. Increasing to $k = 12.5 \max\{d, \log(2/\varrho)\}$ remedied this. However, those datasets represent cases in which linear mixing is likely to underperform AdaSSP and, moreover, situations in which $\|Y\|^2 \approx \|Y - X\theta^*\|^2$, putting in question the relevance of the linear regression in these datasets. Furthermore, on these datasets, the IHM performed better with $T = 2$, since in these cases $\|\theta^*\| \ll 1$.

Whenever we used the Gaussian mechanism, we have calculated the amount of noise based on the analytic (and tight) formula from (Balle & Wang, 2018). This fixes the baseline for all the methods and allows us to compare only the algorithmic aspects of the different methods, rather than the tightness of any of the noise parameters. In AdaSSP, this corresponds to replacing the quantities $\frac{\sqrt{\log(6/\delta)} C_X^2}{\varepsilon/3}$ and $\frac{\sqrt{\log(6/\delta)} C_X C_Y}{\varepsilon/3}$ with the noise scale calculated inside the functions

`AnalyticGaussianMechanism`$\left(X^\top X, C_X^2, \varepsilon/3, \delta/3\right)$ and `AnalyticGaussianMechanism`$\left(X^\top Y, C_X C_Y, \varepsilon/3, \delta/3\right)$ where `AnalyticGaussianMechanism` is defined in (Balle & Wang, 2018, Algorithm. 1).

The next table provides key characteristic parameters for each of the datasets we have simulated. We draw relevant quantities (for example, high residuals) in bold, plum, serifed font. Following Appendix C.1.2, the condition on $k$ required for prevnting clipping is $\frac{\chi \sqrt{\kappa(M)}}{2 - \chi} < 1$. To evaluate this quantity, we substitute $\chi = \sqrt{\frac{d}{k}}$ and print this quantity in the last column of this table.

| Dataset | $n$ | $d$ | $\lambda_{\min}^X$ | $\lambda_{\min}^{XY}$ | $\frac{L(\theta^*)}{\gamma_{\mathrm{m}}(1+\mathrm{C}_Y^2)}$ | $\frac{1}{n}L(\theta^*)$ | $\frac{1}{n}\|Y\|^2$ | $\frac{\chi\sqrt{\kappa(M)}}{2-\chi}$ |
|---|---|---|---|---|---|---|---|---|
| 3droad | 391386 | 3 | 13141.3 | 9609.98 | 146.352 | **0.026** | **0.027** | 0.409 |
| Airfoil | 1202 | 5 | $<10^{-3}$ | $<10^{-3}$ | 1.38787 | 0.048 | 0.787 | 0.229 |
| AutoMPG | 313 | 7 | $<10^{-3}$ | $<10^{-3}$ | 0.315 | 0.04 | 0.362 | 0.250 |
| Autos | 143 | 25 | $<10^{-3}$ | $<10^{-3}$ | 0.03390 | 0.011 | 0.137 | 0.217 |
| Bike | 8708 | 18 | $<10^{-3}$ | $<10^{-3}$ | 3.53458 | 0.021 | 0.072 | 0.630 |
| BreastCancer | 155 | 33 | $<10^{-3}$ | $<10^{-3}$ | 0.11666 | 0.046 | 0.233 | 0.220 |
| Buzz | 524925 | 77 | $<10^{-3}$ | $<10^{-3}$ | **124.782** | 0.028 | 0.066 | 0.220 |
| Communities & Crime | 657 | 99 | $<10^{-3}$ | $<10^{-3}$ | 0.285 | 0.041 | 0.22 | 0.217 |
| Concrete | 824 | 8 | 0.00497 | 0.00495 | 0.32850 | 0.015 | 0.229 | 0.352 |
| Concrete Slump | 92 | 7 | **0.672** | **0.01** | 0.006 | 0.002 | 0.153 | 0.219 |
| Elevators | 14939 | 18 | $<10^{-3}$ | $<10^{-3}$ | 2.20847 | 0.008 | 0.037 | 0.216 |
| Energy | 614 | 8 | $<10^{-3}$ | $<10^{-3}$ | 0.08278 | 0.005 | 0.319 | 0.317 |
| Fertility | 90 | 9 | 1.50256 | 1.49546 | 0.25206 | **0.076** | **0.104** | 0.217 |
| Forest | 466 | 12 | 0.07957 | 0.07855 | 0.77175 | **0.055** | **0.057** | 0.216 |
| Gas | 2051 | 128 | $<10^{-3}$ | $<10^{-3}$ | 0.12730 | 0.006 | 0.112 | 0.229 |
| Housing | 404 | 13 | $<10^{-3}$ | $<10^{-3}$ | 0.11208 | 0.01 | 0.239 | 0.267 |
| KEGG (Directed) | 43944 | 20 | $<10^{-3}$ | $<10^{-3}$ | 6.41448 | 0.009 | 0.117 | 0.214 |
| KEGG (Undirected) | 57247 | 27 | $<10^{-3}$ | $<10^{-3}$ | 1.71694 | 0.002 | 0.069 | 0.224 |
| Kin40K | 36000 | 8 | 1910.7 | 1908.81 | 39.1509 | **0.064** | **0.064** | 0.219 |
| Machine | 188 | 7 | 0.16018 | 0.14595 | 0.13828 | 0.022 | 0.117 | 0.217 |
| Parkinsons | 4700 | 21 | $<10^{-3}$ | $<10^{-3}$ | 0.46978 | 0.004 | 0.105 | 0.434 |
| Pendulum | 567 | 9 | 4.24602 | 4.24482 | 0.25216 | **0.018** | **0.024** | 0.223 |
| Pol | 13500 | 26 | 0.004 | 0.004 | **42.11** | 0.183 | 0.345 | 0.230 |
| Protein | 41157 | 9 | 0.021 | 0.02 | **83.5075** | 0.12 | 0.167 | 0.232 |
| Pumadyn32nm | 7372 | 32 | 125.328 | 125.326 | 11.0211 | **0.093** | **0.093** | 0.222 |
| Servo | 150 | 4 | **2.09738** | **1.15909** | 0.35435 | 0.075 | 0.194 | 0.223 |
| Slice | 48150 | 385 | $<10^{-3}$ | $<10^{-3}$ | 5.29056 | 0.026 | 0.196 | 0.217 |
| SML | 3723 | 26 | $<10^{-3}$ | $<10^{-3}$ | 0.49965 | 0.007 | 0.210 | 0.223 |
| Solar | 960 | 10 | $<10^{-3}$ | $<10^{-3}$ | 0.247 | **0.01** | **0.012** | 0.223 |
| TamiElectric | 36624 | 3 | 32.4147 | 32.4137 | 209.37 | **0.334** | **0.334** | 0.782 |
| Tecator | 192 | 124 | $<10^{-3}$ | $<10^{-3}$ | $<10^{-3}$ | $<10^{-3}$ | 0.158 | 0.232 |
| Wine | 1087 | 11 | $<10^{-3}$ | $<10^{-3}$ | 0.18471 | 0.007 | 0.507 | 0.227 |
| Yacht | 277 | 6 | 0.17360 | 0.17309 | 0.03012 | 0.003 | 0.110 | 0.226 |

*Table 2.* Key parameters from the datasets simulated in this work. The values $\frac{L(\theta^*)}{\gamma_{\mathrm{m}}(1+\mathrm{C}_Y^2)}$ and $\frac{\chi\sqrt{\kappa(M)}}{2-\chi}$ were calculated for $\varepsilon = 0.65$.

# I. Additional Experiments

Results for the full set of simulated datasets are shown below. All experimental settings match those described in Appendix H.

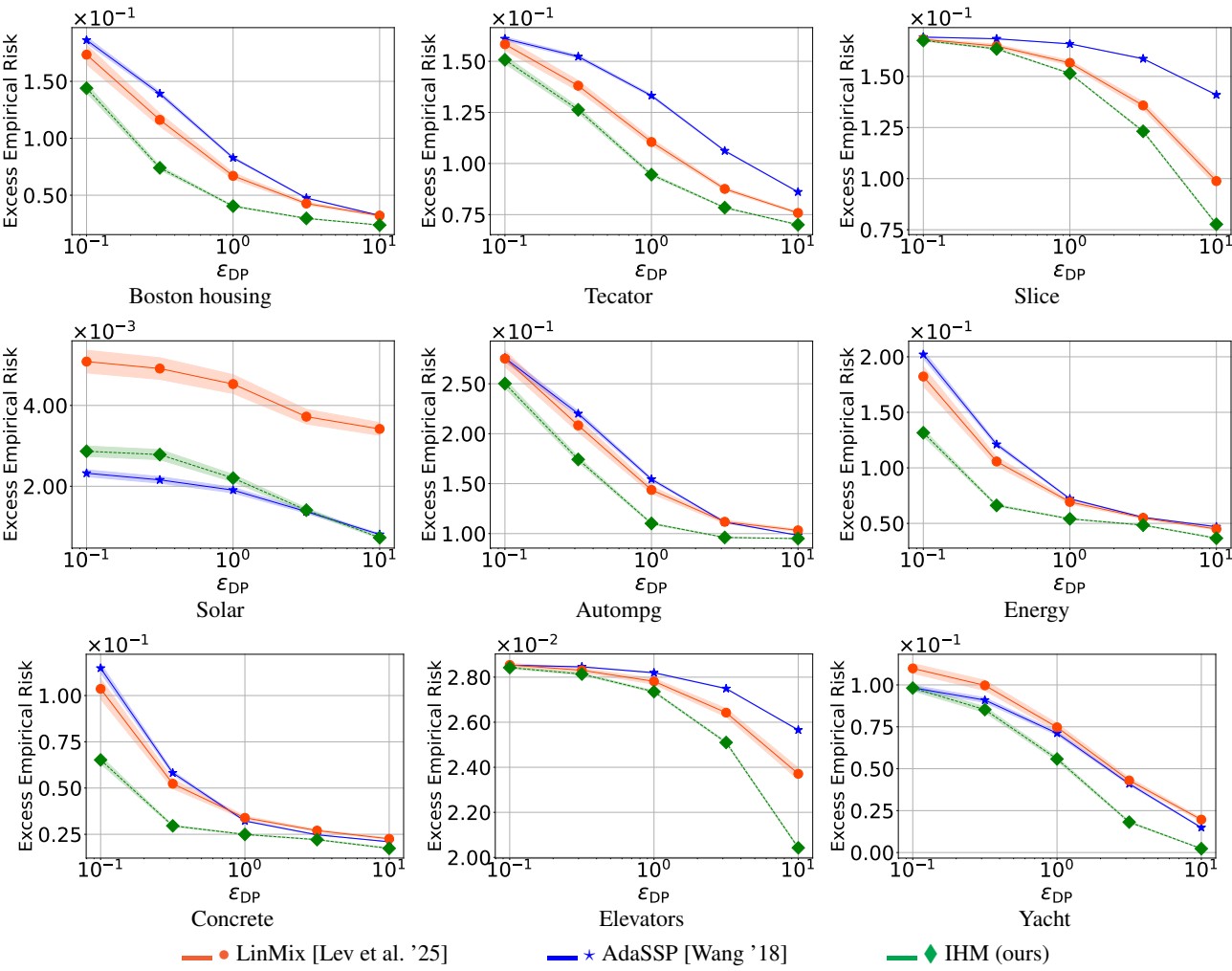

*Figure 3.* Overall performance comparison.

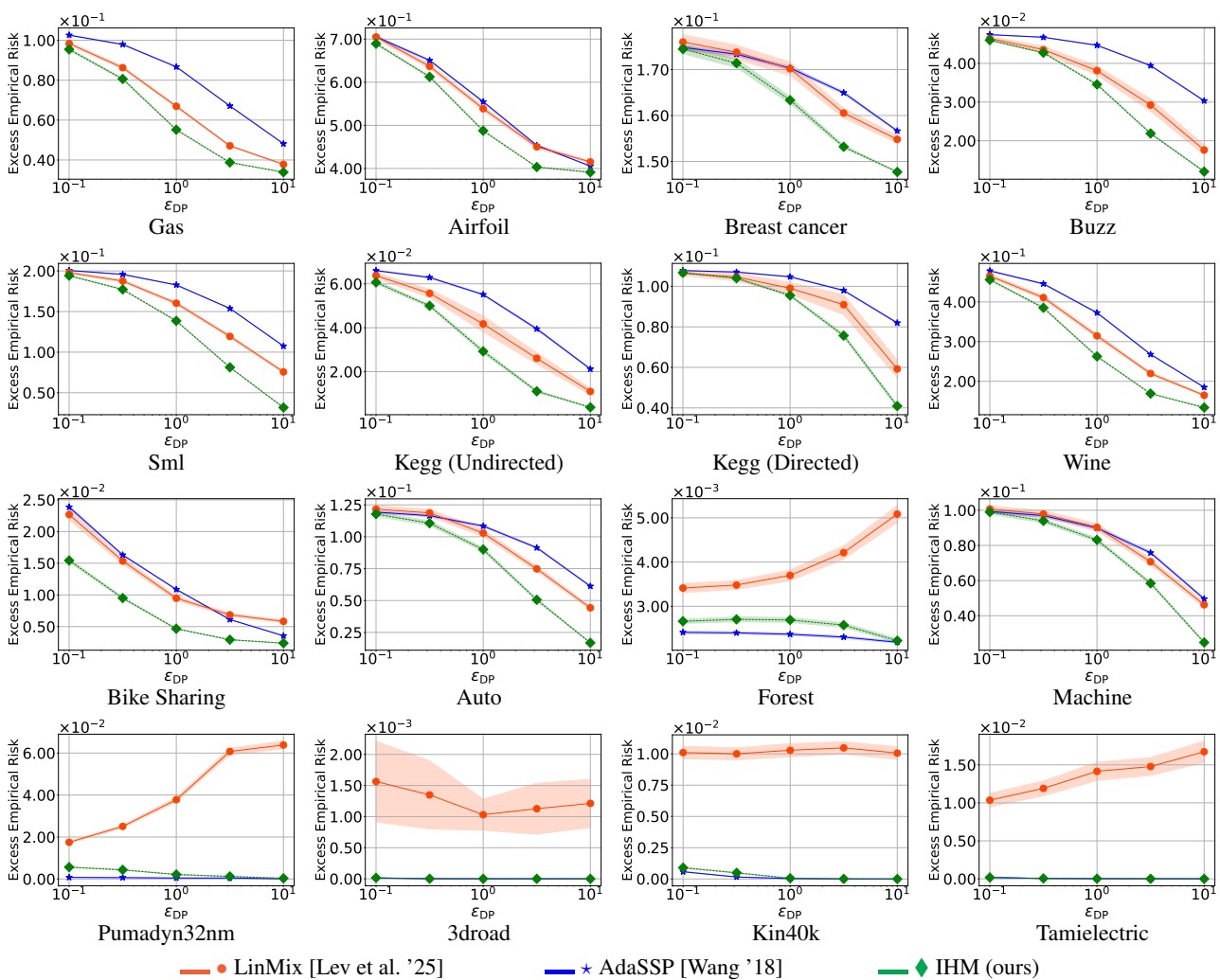

*Figure 4.* Overall performance comparison.

## J. Additional Experiments: Triggering the Clipping Operation

Our goal is to construct a setting that deliberately violates the theorem's assumption controlling $\kappa_X(\psi)$, so that clipping occurs with non-negligible probability. To this end, we generate low-rank covariates by first sampling a set of latent Gaussian variables and mapping them through a randomly initialized two-layer MLP whose output dimension is $2^7$. Denoting the resulting feature vectors by $\{\phi(x_i)\}_{i=1}^n$, we then generate responses according to the linear model

$$y_i = (\phi(x_i))^\top \theta_0 + \sigma z_i,$$

where $\theta_0$ is a randomly generated unit vector, $z_i \overset{\text{iid}}{\sim} \mathcal{N}(0, 1)$, and we set $\sigma = 0.2$ and $n = 2^{19}$. Because the features $\phi(x_i)$ concentrate around a low-dimensional manifold, we expect the empirical covariance to become increasingly ill-conditioned as $n$ grows, and in particular for $\lambda_{\max}^X$ (with $X$ the design matrix whose rows are $\phi(x_i)^\top$) to increase with $n$. Consequently, $\Delta\lambda$ becomes large—an effect we indeed observe empirically—which implies that the clipping threshold is exceeded with non-negligible probability. Nevertheless, as shown in Figure 5, our method consistently outperforms the baselines across the simulated range of privacy levels $\varepsilon$, suggesting that it can remain competitive even in regimes where clipping events are frequent, and the theorem's sufficient conditions do not hold.

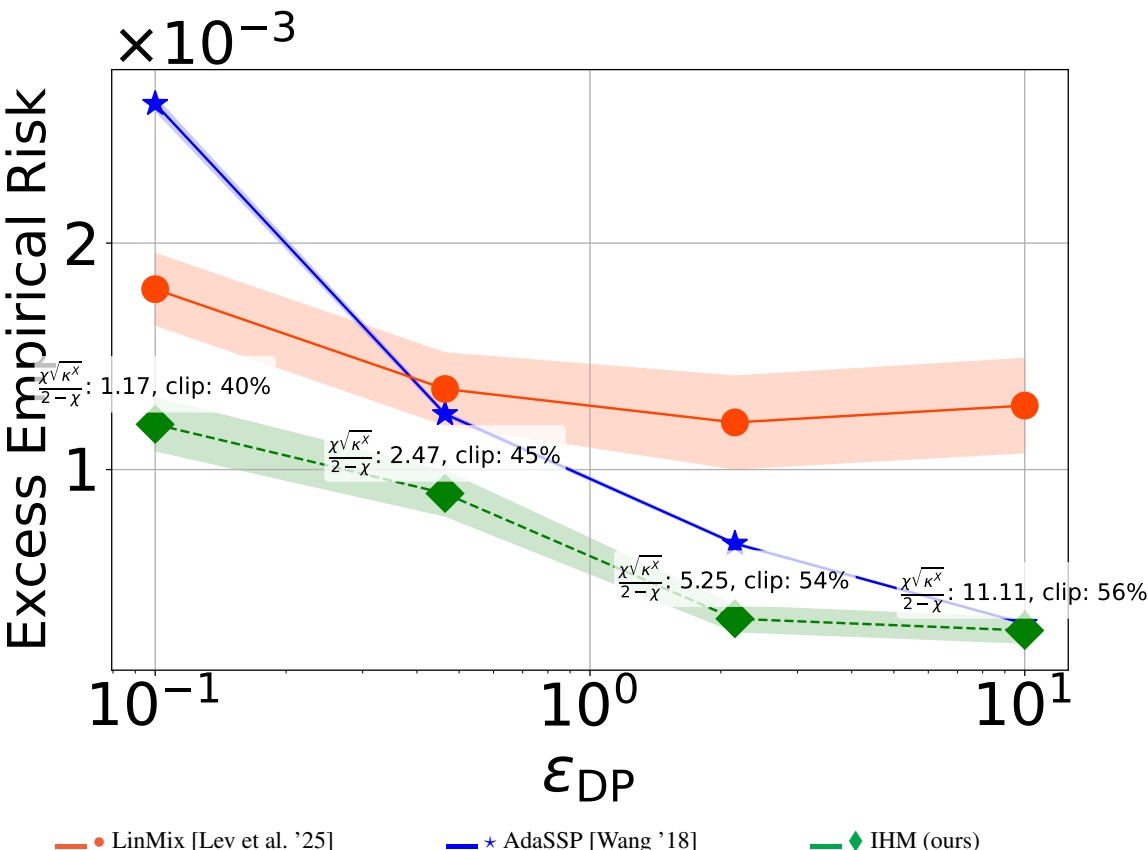

*Figure 5.* Synthetic dataset with non-negligible clipping probability. As demonstrated, for each simulated value of $\varepsilon$ more than $40\%$ of the iterations undergo a clipping operation, yet our method still outperforms the alternative techniques. Additionally, the frequency of the clipping events grows with $\frac{\chi\sqrt{\kappa^X}}{2-\chi}$, as expected.

## K. Additional Experiments: Extreme Minimal Eigenvalue

Our goal now is to construct a dataset where the extra term in (3) is deliberately large, to demonstrate when increasing $T$ is necessary. To that end, we generate covariates that are distributed i.i.d.and uniformly on the $d$-dimensional sphere. In that case, we note that

$$X^\top X = \sum_{i=1}^n x_i x_i^\top \approx \frac{n}{d}\mathbb{I}_d$$

so $\lambda_{\min}(X^\top X) \approx \frac{n}{d}$, which corresponds to its maximal attainable value since for any matrix $X$ the next bounds hold $d \cdot \lambda_{\min}(X^\top X) \le \mathrm{Tr}(X^\top X) \le n$. Then, we generate responses according to the linear model

$$y_i = x_i^\top \theta_0 + \sigma z_i$$

where $\theta_0$ is a randomly generated unit vector and $z_i \overset{iid}{\sim} \mathcal{N}(0,1)$ and $\sigma = 0.01$. We have fixed $n = 2^{15}$ and $d = 2^5$. We simulate the same setting of the IHM from the paper (with $T = 3$ and $k = 6 \cdot \max\{d, \log(^{4T}/_\varrho)\}$) and further another $T = 6$ and $k = 12 \cdot \max\{d, \log(^{4T}/_\varrho)\}$. As can be seen, in this extreme setup, slightly increasing the number of iterations and $k$ is needed for matching the performance of AdaSSP in the high $\varepsilon$ regime, as is predicted by the tuning presented in Appendix C.1.2.

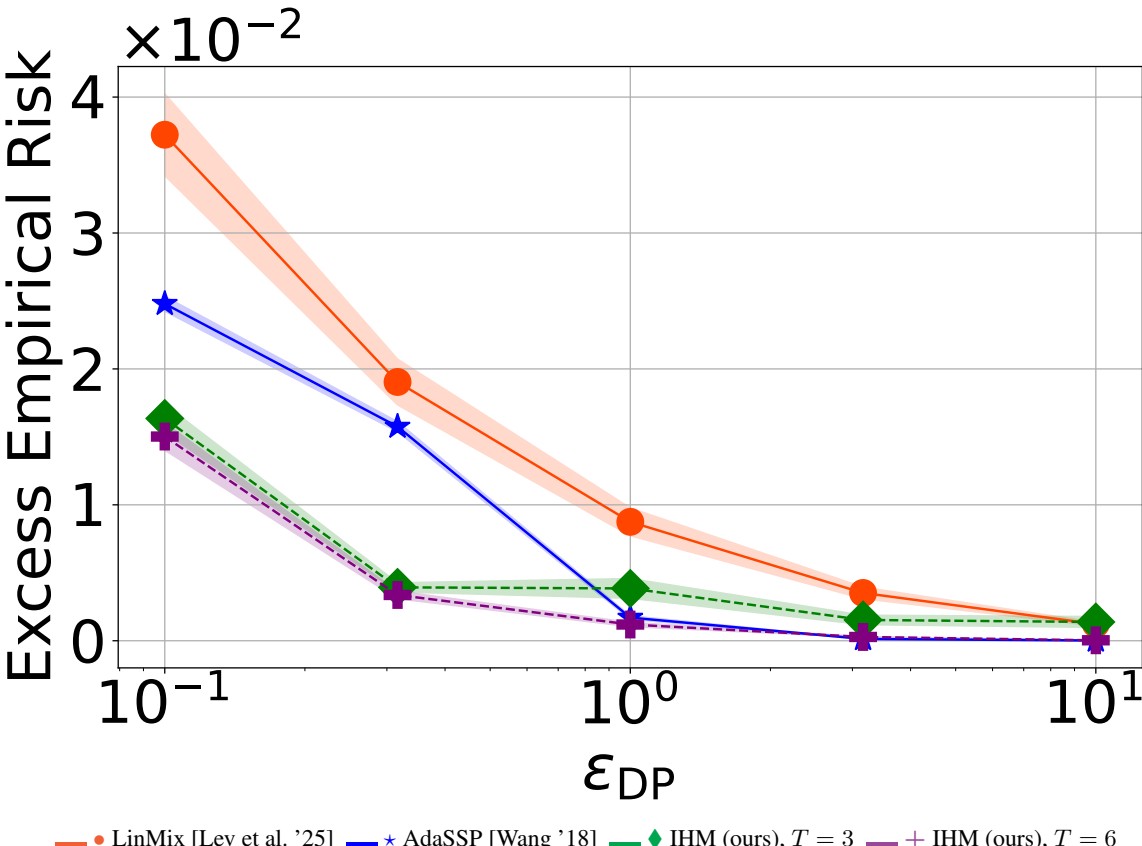

*Figure 6.* Synthetic dataset with $\lambda_{\min}^X \approx \frac{n}{d}$. As demonstrated, increasing $T$ and $k$ for IHM improves its performance in the regime where $\varepsilon$ is large.

## L. Additional Experiments: Different Number of Iterations

We note that on the solar, pendulum, fertility, forest, kin40k, tamielectric, 3droad, and pumadyn32nm datasets, the AdaSSP scheme outperforms IHM with $T = 3$ in the regime $\varepsilon \leq 1$. As reported in Table. 2, these datasets correspond to cases where $L(\theta^*) \approx \|Y\|$ and thus $\|\theta^*\| \ll 1$, corresponding to $C_Y^2 \|\theta^*\|^2 \ll 1$. Then, in this regime, the two terms in (3) can be balanced with a smaller number of iterations, so the choice $T = 3$ is suboptimal. In the simulations presented below, where we set $T = 2$, this gap is substantially reduced, and IHM achieves similar or improved performance, even when $\varepsilon \leq 1$.

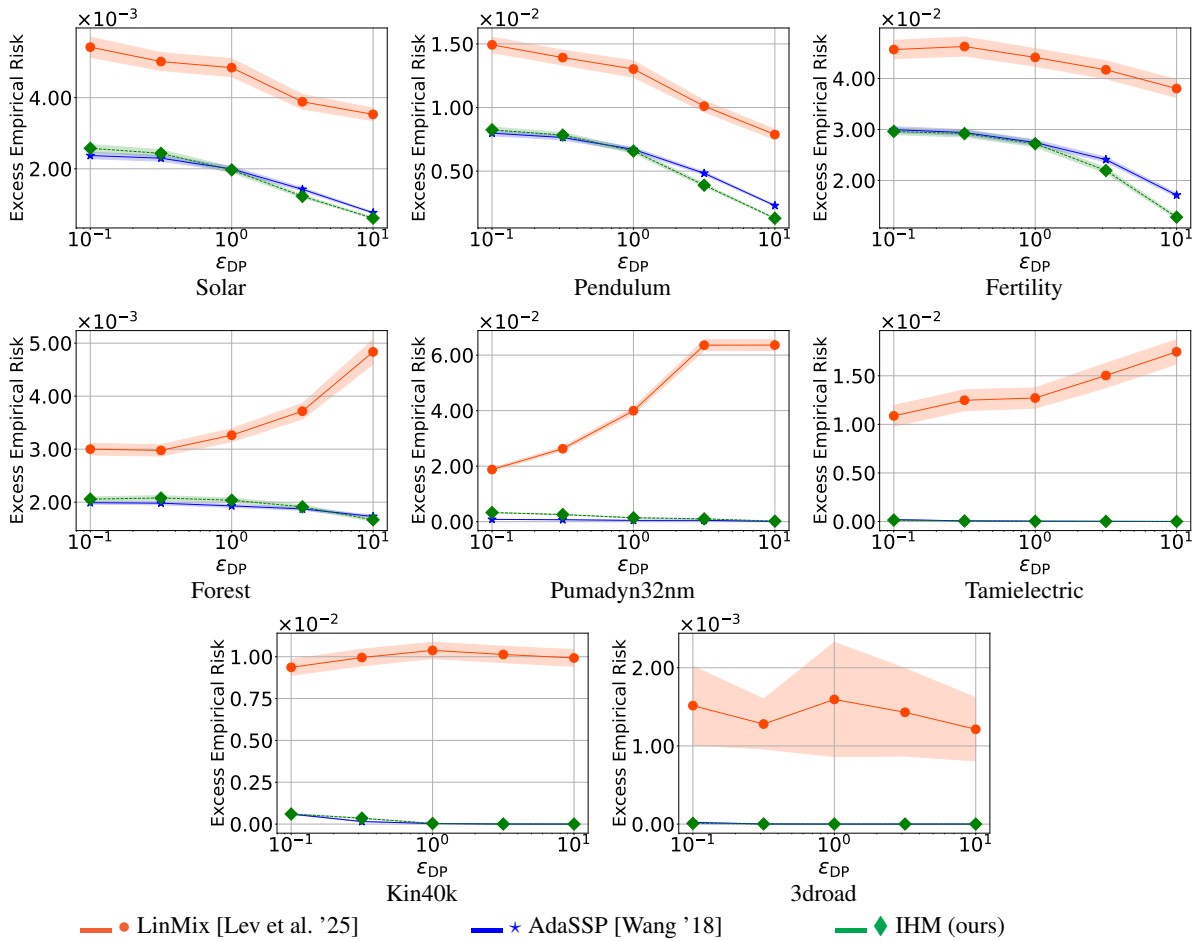

*Figure 7.* Overall performance comparison on datasets satisfying $L(\theta^*) \approx \|Y\|^2$. On these datasets, using $T = 2$ iterations improves the performance of IHM.

## M. Additional Experiments: Additional Baseline

This set of experiments contains an additional baseline that relies on a private, first-order iterative convex optimization method. In particular, we picked the DP-GD implementation from (Brown et al., 2024b). We have fixed the number of iterations and the clipping bounds similarly to those specified by IHM (namely, $T = 3$ and, since the data is normalized for $C_X = 1$, we set a similar clipping level of $C_Y$). The noise scale was chosen according to the zCDP analysis from (Brown et al., 2024b) converted to $(\varepsilon, \delta)$-DP via (Bun & Steinke, 2016) and was fixed on

$$\sigma^2 = \frac{2TC^2}{n^2 \left( \sqrt{\varepsilon + \log(1/\delta)} - \sqrt{\log(1/\delta)} \right)^2}.$$

To further emphasize the usefulness of the IHM, which works with hyperparameters that are set globally, we have fixed the learning rate at $0.25$ for all instances tested. This value was picked by testing learning rates out of the grid $\{0.0625, 0.125, 0.25, 0.5, 1, 2, 5\}$ and picking the one that yielded the best train MSE for target $\varepsilon = 0.5$ on the majority of instances. As shown below, in the large majority of instances, our algorithm works better than the DP-GD baseline that runs with a similar number of iterations and clipping bounds, further demonstrating the usefulness of our method against private iterative optimizers.

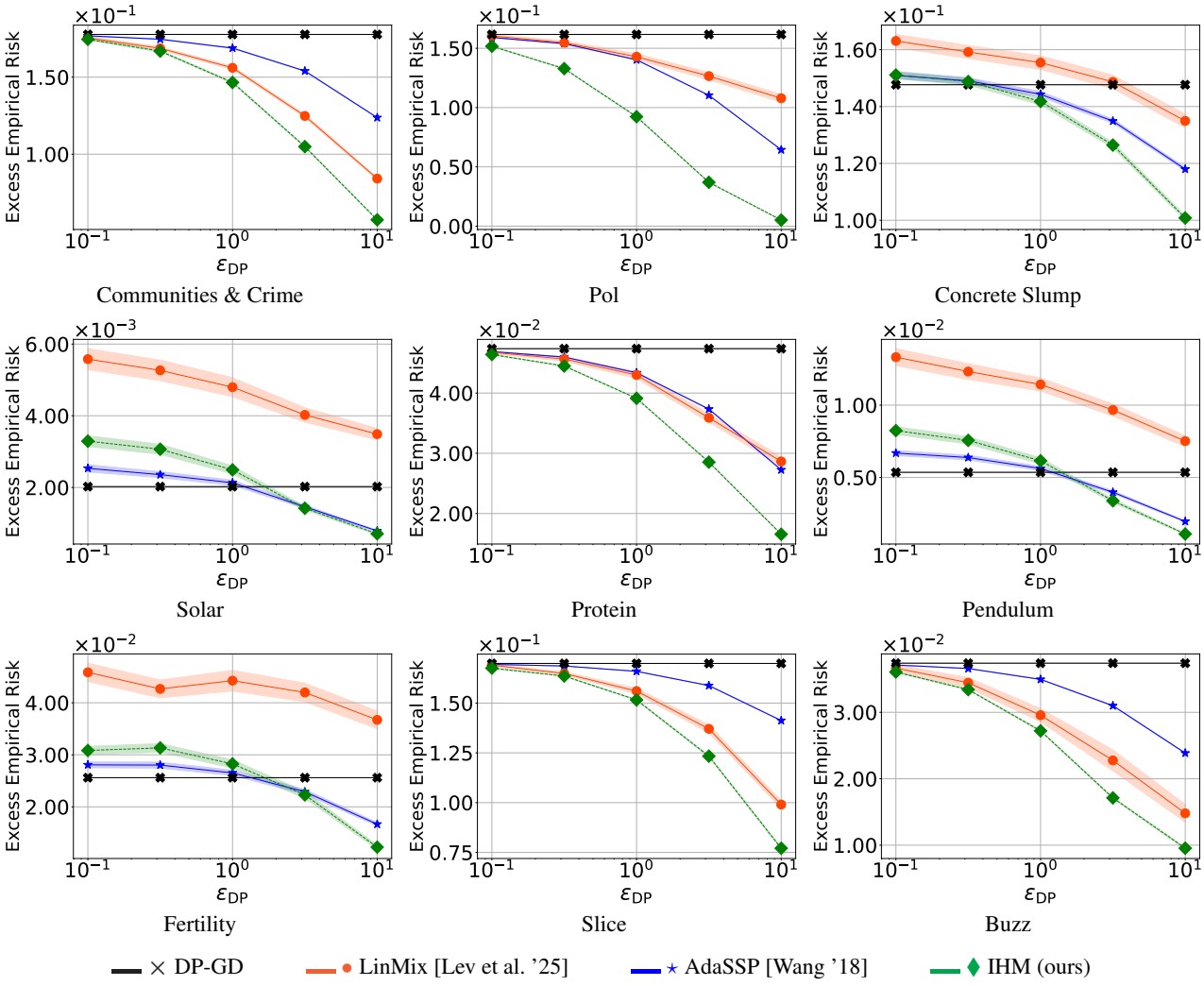

*Figure 8.* Overall performance comparison: together with DP-GD baseline.

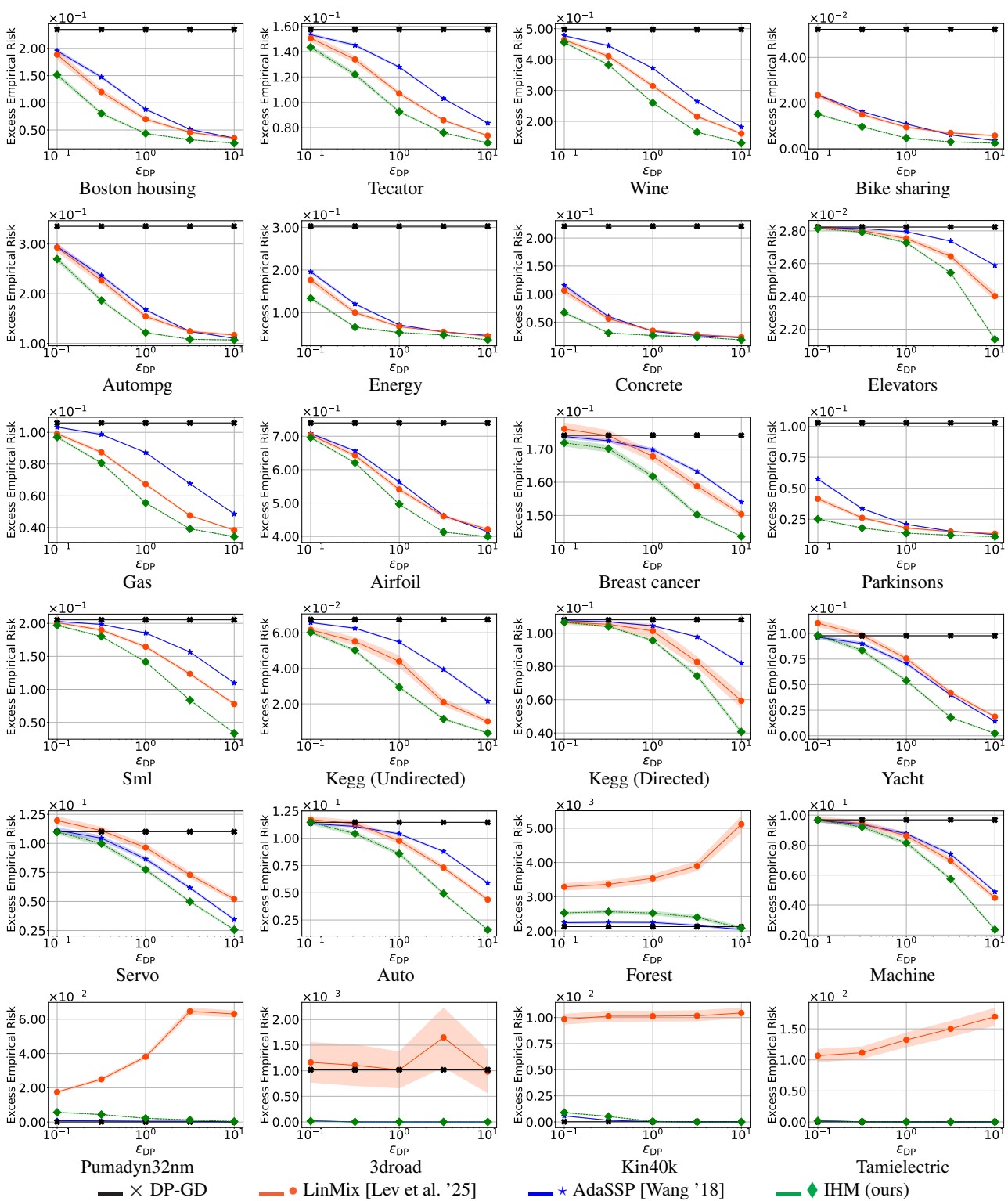

*Figure 9.* Overall performance comparison: together with DP-GD baseline.

