# OpenReview forum: "Near-Optimal Private Linear Regression via Iterative Hessian Mixing"
_ICML.cc/2026/Conference — ICML 2026 spotlight_

### Official Review · Reviewer_rGyE · 2026-03-07

**Soundness:** 3
**Presentation:** 3
**Significance:** 4
**Originality:** 3
**Overall Recommendation:** 5
**Confidence:** 4

**Summary:**

This paper studies centrally differentially private ordinary least squares (DP-OLS) under bounded covariates and responses. The authors propose Iterative Hessian Mixing (IHM), an iterative procedure inspired by the Iterative Hessian Sketch that sketches the design matrix inside a Hessian matrix surrogate while using a clipped-residual gradient term with Gaussian noise to ensure privacy.

The paper provides (i) a privacy guarantee for IHM under $(\varepsilon,\delta)$-DP, and (ii) high-probability excess empirical risk bounds for IHM that are compared against the leading sufficient-statistics perturbation baseline AdaSSP and Gaussian-sketch-based approaches. In addition, the paper presents a new utility analysis for a representative LinMix. Empirically, IHM is evaluated on 33 UCI regression datasets plus synthetic stress tests and is reported to consistently match or outperform prior baselines across a wide range of privacy levels.

**Compliance With Llm Reviewing Policy:**

Affirmed.

**Final Justification:**

The rebuttal addressed my main concerns satisfactorily. In particular, the authors clarified the role and limitations of the no-clipping analysis, gave a reasonable explanation for why omitting the $-\eta^2\hat\theta_t$ term can improve alignment with the OLS target in practice, and provided a clearer interpretation of the noise-calibration threshold. These responses did not change my overall view of the paper, but they did increase my confidence that the current weaknesses are mostly about conservativeness of the analysis and presentation/positioning rather than a fundamental flaw. I therefore maintain my positive recommendation.

**Key Questions For Authors:**

1. In Appendix H, you omit the $-\eta^2 \hat\theta_t$ term in $\tilde G_t$ for better empirical performance. Can you (a) comment on why this modification is helpful in practice, and (b) report key figures with the exact Algorithm 1 update as written? The appearance of $-\eta^2 \hat\theta_t$ seems tied to the “augmented design / ridge” view (i.e., treating the update as an IHS step on $[X;\eta I]$, where $-\eta^2\hat\theta_t$ corresponds to the gradient of an $\ell_2$ term). Dropping it changes the update/objective, so it would be helpful to explain this mismatch more directly (ideally in the main text).
2. I found the calibration step for Gaussian mixing particularly insightful. Algorithm 2 sets
$\tilde\eta=\sqrt{\max\{\gamma C_X^2-\tilde\lambda,\,0\}}$,
where $\tilde\lambda$ is a private noisy estimate of $\lambda_{\min}(X^\top X)$ that serves as a lower bound with high probability. This suggests a clear threshold-like behavior: when $\tilde\lambda$ exceeds $\gamma C_X^2$, no additional isotropic mixing noise is needed ($\tilde\eta=0$); once $\gamma C_X^2$ surpasses $\tilde\lambda$, $\tilde\eta$ becomes strictly positive, effectively adding a ridge term that raises the effective minimum eigenvalue used in the privacy analysis.
More precisely, under the good event used in the proof, the calibration ensures that $\lambda_{\min}(X^\top X)+\tilde\eta^2 \ge \gamma$, which is exactly what allows the authors to invoke the Gaussian mixing RDP guarantee in Lemma 2. Since the RDP cost of Gaussian mixing grows linearly with the sketch dimension $k$, and in IHM the privacy calibration is performed for the composed sketching budget across $T$ iterations, this creates a threshold-like dependence on the sketching budget. I think it would strengthen the paper to surface this phenomenon more explicitly, for example by giving an approximate characterization of the transition in terms of $k,T,\lambda_{\min}(X^\top X),\varepsilon,\delta$, or by plotting $\tilde\eta$ versus $k$ on representative datasets. This would provide useful intuition and practical guidance for choosing the sketch size.
3. Can you quantify $g_X$ (or proxies like $\lambda^X_{\max}-\lambda^X_{\min}$) on the real datasets and show how often $g_X$ actually drives the required sketch size $k$ (Sec. 3.1 / Appendix C.1.2)? This follows from the last comment about the choice of sketch size $k$, and I think it is a key relevant dataset property for understanding when IHM should help.

**Limitations:**

Yes

**Strengths And Weaknesses:**

- Strengths:
  - DP linear regression is a canonical problem in private statistics, and a core primitive/subroutine in many private analytics pipelines.
  - Theorem 2 provide a concrete, checkable decomposition of the LinMix/sketch-and-solve error into terms involving $\lambda^{XY}_{\min}$ and an explicit dependence on $L(\theta^\*)$. This is a useful conceptual contribution beyond the new algorithm itself.
  - I especially liked the way the experiments are structured to test the theory, rather than merely report a leaderboard. The authors go beyond average performance over many UCI datasets by explicitly examining dataset-specific properties that are central to the analysis (e.g., eigenvalue/conditioning statistics of $X^\top X$ versus $(X,Y)^\top(X,Y)$, clipping behavior, and regime indicators related to $L(\theta^\*)$ and $\|\theta^\*\|$). The additional synthetic and stress-test studies then vary these factors in isolation (e.g., well-conditioned designs, regimes where clipping is non-negligible, and settings where fewer iterations help), and the resulting performance trends closely track the qualitative predictions of the bounds. This kind of “mechanism-level” evaluation makes it much clearer why IHM improves over prior sketch-and-solve baselines in certain regimes, and when the gains should be expected to diminish.
  - The empirical results across many datasets suggest IHM could replace or complement AdaSSP and sketch-and-solve baselines in practice, especially when residual-related terms harm sketch-and-solve.
  - The writing is, like all submissions, clear and fluent. The narrative is also well structured, and the comparison framing is easy to follow.

- Weaknesses:
  - Theorem 1’s proof conditions on clipping not occurring with high probability, but Appendix J explicitly constructs settings with $\ge 40\%$ clipping frequency where the method still works empirically. This mismatch suggests the current theory may not predict behavior in challenging regimes where clipping is active, and the sufficient conditions may be conservative. Separately, the paper assumes a bounded response domain $|y_i|\le C_Y$, and moreover assumes $C_Y$ is known. This is a common modeling device in DP-OLS: many sensitivity-based DP mechanisms (including sufficient-statistics perturbation and gradient-based approaches) require bounded data to obtain finite worst-case sensitivity, and several prior DP linear regression works adopt comparable boundedness/normalization assumptions. That said, for many real regression tasks the response can be heavy-tailed or effectively unbounded, so it would strengthen the paper to more explicitly discuss (i) how $C_Y$ should be chosen in practice without violating privacy (e.g., DP scale/quantile estimation, or a fixed public clipping rule), and (ii) the utility implications/bias introduced by clipping/normalizing $Y$.
  - Appendix H states that in experiments the authors omit the term $-\eta^2 \hat\theta_t$ from $\tilde G_t$ (Algorithm 1, line 6) because it improves performance. While this may be benign, it means the empirical results do not exactly correspond to the stated Algorithm 1 / Theorem 1 pipeline, which weakens the tight linkage between theory and experiments unless further justified.

---

> ### Author Rebuttal · Authors · 2026-03-30
>
> We thank the reviewer for the careful and comprehensive review and for providing insightful suggestions towards improving our work.
>
> **No Clipping Conditions:**
>
> As stated, our guarantees work by specifying no-clipping conditions and performing the analysis conditioned on these. However, for example, in the experiments we report in Appendix J, we include experiments that deliberately violate these assumptions and where clipping occurs with non-negligible probability. In these experiments, and throughout the entire experimental suite, we note that our algorithm continues to consistently provide superior results even in the presence of clipping, suggesting that the actual performance of the algorithm might be better than that predicted by our asymptotic analysis. Characterizing the non-asymptotic performance of the algorithm, potentially even under the presence of clipping, is an important future direction of our work.
>
> Additionally, as stated by the reviewer, our algorithm assumes knowledge of a bound on the magnitude of the responses $C_Y$. Under such a bound, the performance of our algorithm is characterized by Theorem 1. Then, the effect of this clipping comes in the form of changing the baseline $L(\theta^{*})$, but rather does not affect the excess empirical risk guarantees derived. As clarified by the reviewer, picking this $C_Y$ is a current challenge in many practical and popular DP algorithms, and is not directly connected to the new algorithm developed in this work.
>
> **Dropping the Term $-\eta^2 \hat{\theta}_t$:**
>
> As mentioned by the reviewer, the introduction of the term $-\eta^2 \widehat{\theta}_t$ in the term $\widetilde{G}_t$ is since by viewing the matrix $\widetilde{X}_t$ as sketching the augmented design $ (\\mathsf{S}_t, \\xi_t) \\begin{pmatrix}
> X \\\\
> \\eta \\mathbb{I}_d
> \\end{pmatrix} $ the underlying baseline solution is the Ridge regressor $\theta^{\star}(\eta^2)$. Thus, the IHS (and consequently the IHM) geometrially converges to the solution $\theta^{\star}(\eta^2)$ rather than to the (desired) baseline $\theta^{\star}$. Intuitively, discarding the terms $\sigma \zeta_t$ in the IHM, we note that convergence of the algorithm (corresponding to having $\widetilde{G}_t \approx 0$) corresponds to the solution $(X^{\top}X + \eta^2\mathbb{I}_d)^{-1}X^{\top}Y$, while without this term we would get $(X^{\top}X)^{-1}X^{\top}Y$ which will be typically closer to the desired baseline $\theta^{\star}$. The current experimental results do not contain this correction term throughout. In the final version of this work, we will include simulations with this correction term as well as this previous intuition on why omitting it from the guarantees improves the actual performance.
>
> **Noise Calibration Step:**
>
> We thank the reviewer for finding the noise-calibration procedure insightful. A similar intuition appears in [1], and our algorithm builds on this prior result. In particular, note that $\\gamma = \\Theta\\left(\\frac{\\sqrt{kT\\log(1/\\delta)}}{\\epsilon}\\right)$ (Line 895). Hence, the threshold for requiring zero additional noise is
> $$C_X^{-2}\\lambda_{\\min}(X^{\\top}X) \\geq \\Omega\\left(\\frac{\\sqrt{kT\\log(1/\\delta)}}{\\epsilon}\\right).$$
> We agree that this point is important, and we will add this clarification to the paper. Thank you for raising it.
>
> **Empirically Quantifying $g^{X}$:**
>
> Following the asymptotic analysis from Appendix C.1.2, the condition on $g^{X}$ in $k$ comes from the constraint $\frac{\chi\sqrt{\kappa^{X}(\eta^2)}}{2-\chi} < 1.$ As reported in Table 2, this is satisfied throughout the simulated $\epsilon$ range when using the sketch size $k = 6\\cdot \\max\\left\\{d, \\log(T/\\rho)\\right\\}$, so practically over the experimental suite we used the constraint $k\geq g^{X}$ is practically already active by tuning $k$ in a way agnostic to this $g^{X}$.
>
> Moreover, note that our experiments indicate that IHM continues to outperform the competing methods even in regimes where clipping occurs. This suggests that the practical behavior of IHM is stronger than what is captured by our current asymptotic analysis, and that the present theory may be conservative. Developing a full non-asymptotic analysis of IHM and sharpening the corresponding performance guarantees is an important direction for future work.
>
> [1] Lev, Omri, et al. "The Gaussian Mixing Mechanism: Rényi Differential Privacy via Gaussian Sketches." NeurIPS 2025

---

> > ### Author Rebuttal · Reviewer_rGyE · 2026-04-01
> >
> > The authors have well addressed my questions, and I therefore maintain my positive recommendation for acceptance.

---

### Official Review · Reviewer_ZFEC · 2026-03-10

**Soundness:** 3
**Presentation:** 3
**Significance:** 2
**Originality:** 3
**Overall Recommendation:** 3
**Confidence:** 3

**Summary:**

This paper studies the problem of differentially private ordinary least squares (DP-OLS). The authors propose a new algorithm, Iterative Hessian Mixing (IHM), which combines ideas from Gaussian sketching and iterative Hessian sketch techniques. The method iteratively constructs privatized sketches of the design matrix and performs updates using approximate second-order information.The paper provides theoretical guarantees on the excess empirical risk of the proposed estimator under $(\varepsilon,\delta)$-differential privacy. In addition, the authors present a theoretical comparison between IHM, AdaSSP, and previous Gaussian sketch–based methods. The analysis suggests that IHM can remove a multiplicative factor (potentially up to $\sqrt{d}$) that appears in the leading term of AdaSSP’s error bound, while also avoiding the residual sensitivity issue present in earlier sketching approaches.

**Compliance With Llm Reviewing Policy:**

Affirmed.

**Final Justification:**

The discussion on hyperparameters is helpful, but the selection of $k$, $T$, and $C$ still appears largely heuristic. A more systematic guideline would improve the usability of the method.

While the potential extensions are interesting, the current work remains focused on DP-OLS in the overdetermined regime ($n \ge d$), which is somewhat narrow. A clearer discussion of this limitation would strengthen the paper.

**Key Questions For Authors:**

1. How does the computational cost of IHM compare to AdaSSP and LinMix in practice? In particular, how does the runtime scale with $n$ and $d$?

2.  How sensitive is the performance of IHM to the choice of sketch dimension $k$ and iteration number $T$?

3.  Can the iterative Hessian mixing framework be extended to other differentially private learning problems, such as logistic regression or generalized linear models?

**Limitations:**

Yes.

**Strengths And Weaknesses:**

Strengths:

1. The paper provides theoretical guarantees on the excess empirical risk of the estimator under differential privacy. The comparison with AdaSSP and Gaussian sketch–based methods helps clarify the relative advantages of the proposed approach and contributes to the theoretical understanding of DP-OLS algorithms.

2.The discussion comparing AdaSSP, LinMix, and the proposed method is informative. In particular, the analysis of how residual sensitivity and eigenvalue dependence affect the error bounds provides useful insights into when different algorithms may perform better.

Weakness:

1. While the algorithm involves iterative updates and repeated sketching operations, the paper provides limited discussion of the computational complexity compared to existing methods. A clearer analysis of the runtime and scalability would help readers better understand the practical cost of the method.

2. The algorithm involves several parameters, such as the sketch dimension $k$, the number of iterations $T$, and the clipping parameter $C$. Although some guidance is provided, the sensitivity of the algorithm to these parameters is not thoroughly studied. Additional empirical analysis or practical guidelines would strengthen the paper.

3.  The paper focuses specifically on ordinary least squares (OLS) regression under differential privacy. While OLS is a classical problem, its applicability in modern machine learning settings is somewhat limited, especially compared to more widely used models such as logistic regression, generalized linear models, or regularized estimators. Moreover, the theoretical analysis in the paper assumes a relatively favorable high-sample regime **(e.g., $n \ge d$)**. As a result, the practical impact of the proposed method may be restricted to a relatively narrow class of problems.

---

> ### Author Rebuttal · Authors · 2026-03-30
>
> We thank the reviewer for their feedback.
> Below, we answer the questions posed in the review and provide an extended discussion about the points of weakness highlighted.
>
> **Computational Complexity:**
>
> The cost of our algorithm is typically $T$ times higher than AdaSSP, due to the iterative nature of our algorithm. However, since practically $T$ can be chosen to be a small constant, this increase in cost is usually negligible. Mathematically, AdaSSP involves solving one regression problem of size $n\times d$ and thus is given as $O(nd^2)$. The IHM, in contrast, involves $T$ Gaussian sketching steps with a Gaussian matrix of size $k\times n$, where each step is followed by solving a linear regression problem of size $k\times d$. Thus, the overall complexity is evaluated as $O\left(T\cdot \left(kd^2 + knd\right)\right)$. Subsistuting $k = \\Theta\\left(\\max\\left\\{d, \\log(1/\\rho)\\right\\}\\right)$ yields overall complexity of $O\\left(T\\cdot \\max\\left\\{d, \\log(1/\\varrho)\\right\\}\\cdot nd\\right)$, which is approximately $T$ times higher than that of AdaSSP. Since usually $T$ can be set to a small constant (in our experiments, $T=3$), this overhead does not appear to preclude practical deployment. Additionally, our theoretical analysis, performed in Appendix.C.1.2 (Lines.1071-1077) suggests that the optimal $T$ grows at most like $T = \\Theta\\left(\\log\\left(\\frac{(\\epsilon\\lambda^X_{\\min})^2}{\\max\\left\\{d, \\log(1/\\rho), g^X\\right\\}\\log(1/\\delta)}\\right)\\right)$, which might further be reasonably small for most practical settings. Extending the method to use sketches that admit faster computations and reducing this cost is an important direction for future work. We would like to emphasize that the central goal of our paper is to provide a near-optimal construction that is both (a) practically implementable, and (b) provably achieves \((\epsilon,\delta)\)-DP.
>
> **Sensitivity to Hyperparameters Tuning:**
>
> The choices of $k$, $T$, and $C$ were guided in practice by principles similar to those used in the classical sketching literature [1].
> Moreover, our simulations suggest that changing these by constant factors typically has only a minor effect on performance, and the values used in the paper were selected by numerical tuning over the evaluated datasets, as we discuss in Appendix.H. Two important edge cases for calibrating $T$ are the low-SNR regime and the regime in which $\lambda_{\text{min}}^{X} \gg 1$, as clarified in Appendix.L and Appendix.K, where using fewer or more iterations may improve performance. We will add these practical guidelines to the discussion of hyperparameter selection in Section 3.1.
>
> **Extension to Other Problem Instances:**
>
> Our focus in this paper is on DP-OLS, as our primary goal is to demonstrate that sketching-based methods can offer better performance than classical DP constructions. DP-OLS provides a natural starting point: it is a fundamental problem, theoretically well understood, and rich enough to support meaningful guarantees and algorithmic insights. Nevertheless, the scope of these ideas is broader. The classical Hessian sketch of [1], as well as its extension in [2], applies beyond the unconstrained OLS setting; indeed, the original Hessian sketch was introduced for general constrained least-squares problems. The Hessian mixing framework studied in our work can likewise be extended to accommodate such settings. More generally, related ideas may be developed in the spirit of the Newton sketch from [2], opening the door to DP second-order methods for broader classes of convex optimization problems. In this sense, IHM may be viewed as a first theoretically grounded and empirically successful step toward a more general private Newton sketch framework, which we leave for future investigation. Lastly, we note that the overdetermined regime ($n\geq d$) is usually a standard assumption in many classical works around OLS and DP-OLS (see, for example, [3,4]), which was followed in our work.
>
> [1] Pilanci, Mert, and Martin J. Wainwright. "Iterative Hessian sketch: Fast and accurate solution approximation for constrained least-squares." JMLR.
>
> [2] Pilanci, Mert, and Martin J. Wainwright. "Newton sketch: A near linear-time optimization algorithm with linear-quadratic convergence." SIAM Journal on Optimization.
>
> [3] Yu-Xiang Wang. Revisiting differentially private linear regression: Optimal and adaptive prediction \& estimation in unbounded domain. In the Conference on Uncertainty in Artificial Intelligence.
>
> [4] Sheffet, Or. "Differentially private ordinary least squares." ICML, 2017.

---

> > ### Author Rebuttal · Reviewer_ZFEC · 2026-04-04
> >
> > Thank you for the clarifications.
> >
> > The discussion on hyperparameters is helpful, but the selection of $k$, $T$, and $C$ still appears largely heuristic. A more systematic guideline would improve the usability of the method.
> >
> > While the potential extensions are interesting, the current work remains focused on DP-OLS in the overdetermined regime ($n \ge d$), which is somewhat narrow. A clearer discussion of this limitation would strengthen the paper.

---

> > > ### Author Response · Authors · 2026-04-04
> > >
> > > We thank the reviewer for the helpful follow-up.
> > >
> > > Regarding the choice of hyperparameters, the non-asymptotic selections were made according to the following principles, which are already reflected in the manuscript, although we agree that they should be explained more explicitly.
> > >
> > > 1. The sketch size was set to $k = 6 \\cdot \\mathrm{max}\\{d, \\mathrm{log}(4T/\\varrho)\\}$, following the same constant used in the classical Hessian sketch for OLS; see Section 3.1 of [1]. In particular, [1] uses $6d$, whereas in our setting we further inflate $k$ to account for the desired overall failure probability.
> > >
> > > 2. The choice $T=3$ was made to balance two sources of error: the non-asymptotic residual term, bounded by $(\\sqrt{2}\\chi)^{2T} n$, and the noise-induced term, of order $\\gamma_h C_h$, for typical values of $n,d,\\epsilon,\\delta$, as discussed in Appendix H. Empirically, this choice gives near-optimal performance on most problem instances. At the same time, as we explain in the paper, there are two important regimes in which a modified choice of $T$ yields better performance, and we provide both mathematical and empirical guidance for selecting $T$ in those cases (in particular, the low-SNR regime and the extreme minimal eigenvalue regime).
> > >
> > > 3. Our asymptotic analysis shows that choosing $C = a \\cdot \\mathrm{max}\\{C_Y, ||\\theta^*||\\}$ avoids clipping events. In practice, we set $C = C_Y$, based on the assumed bound on $Y$, and because under this choice clipping appears to be negligible; see also our response to Reviewer poR3.
> > >
> > > We therefore agree with the reviewer that these choices, and especially the choice of $C$, should be better motivated in the presentation, and we will revise the manuscript accordingly.
> > >
> > > Regarding the focus on overdetermined DP-OLS, we agree that this restricts the current scope of the work. However, this setting is also the central focus of the paper, as reflected in the title and framing of the manuscript. Our primary goal was to develop a near-optimal algorithm for DP-OLS and to show that it improves over existing methods in this setting, rather than to demonstrate applicability across a broader family of problems. Extending the framework beyond DP-OLS is an important direction for future work, but it was not the main objective of the present paper.
> > >
> > > We thank the reviewer again for the careful reading and constructive suggestions. We agree that these points should be stated more explicitly in the manuscript, as we will do in our revised version.
> > >
> > > [1] Pilanci, Mert, and Martin J. Wainwright. "Iterative Hessian Sketch: Fast and Accurate Solution Approximation for Constrained Least-Squares." Journal of Machine Learning Research 17.53 (2016): 1-38.

---

### Official Review · Reviewer_poR3 · 2026-03-13

**Soundness:** 4
**Presentation:** 3
**Significance:** 3
**Originality:** 3
**Overall Recommendation:** 5
**Confidence:** 4

**Summary:**

This paper studies differentially private ordinary least squares (DP-OLS) regression under bounded data assumptions, where the goal is to output a regression vector that is accurate (in excess empirical risk) while satisfying $(\varepsilon,\delta)$-differential privacy.

The central contribution is a new algorithm, Iterative Hessian Mixing (IHM), which blends two ideas: (i) Gaussian sketching / mixing mechanisms for privacy, and (ii) the Iterative Hessian Sketch (IHS) approach for solving least squares via repeated (Hessian) sketches and geometric error decay.

Concretely, the method iterates a small number of times, forming a privatized “Hessian-like” sketch (via Gaussian sketching applied to an augmented matrix involving $X$ and a scaled identity) and a privatized gradient-like term (via clipping plus Gaussian noise), then updating the parameter vector by solving a sketched linear system.

On the theory side, the paper provides:
 - a full $(\varepsilon,\delta)$-DP proof by composing privacy guarantees for (a) calibrating the mixing noise, (b) Gaussian sketches, and (c) Gaussian mechanism calls, using Rényi DP tools and conversion to approximate DP,

 - a high-probability excess empirical risk bound for IHM that features (a) a geometrically decaying term across iterations, and (b) a remaining statistical/privacy error term comparable in structure to prior near-optimal baselines, but with improved dependence on certain parameters (notably the failure probability and dimension in some regimes).

In addition, the paper contributes a new analysis of prior Gaussian sketch-and-solve DP-OLS methods (e.g., sketching both $X$ and $Y$), highlighting an intrinsic limitation in how their error bounds can scale with the residual at the optimum.

Empirically, the authors report broad experiments over many standard regression datasets and privacy budgets, showing IHM consistently matches or improves over strong baselines such as adaptive sufficient-statistics perturbation (AdaSSP) and Gaussian sketch baselines, aligning with the theoretical narrative.

**Compliance With Llm Reviewing Policy:**

Affirmed.

**Key Questions For Authors:**

1. Parameter robustness and default guidance. The experiments appear to use fairly simple default settings (e.g., fixed small $T$ and straightforward choices of sketch dimension/clipping). How sensitive is IHM to these choices across datasets and privacy budgets? If there are regimes where performance is brittle, what diagnostics or tuning rules do you recommend? A strong, empirically validated “how to set $T,k,C$” story would increase my confidence in the method’s practical impact.


2. Role and prevalence of the “improvement regime.” The main theoretical comparison highlights improved dependence on the failure probability/dimension in some regimes. How often do real datasets fall into those regimes (e.g., when the term controlling the max in $\gamma_{\text{hess}}$ behaves favorably relative to AdaSSP)? A concise empirical breakdown (“X% of datasets satisfy condition Y, and in those we see Z improvements”) would strengthen the significance argument.

3. Computational cost vs. baselines. IHM solves a sketched linear system for multiple iterations. What is the wall-clock/runtime overhead relative to AdaSSP and the sketch-and-solve baseline at comparable privacy/accuracy points? If IHM is slower, can this be mitigated (e.g., reuse factorizations, fewer iterations, structural sketches) without materially harming utility? This could affect the overall recommendation if the intended setting is practical deployment.

4. Clipping behavior and accuracy. The algorithm uses clipping to control sensitivity of the gradient-like term. In the benchmark suite, how frequently does clipping actually trigger (as a function of (\varepsilon), dataset, and iteration)? If clipping activates often, does it systematically bias results on certain tasks? Clear statistics here would help assess whether the “clipping rarely triggers under typical conditions” intuition holds.

5. Reproducibility artifacts. Is there a public implementation (or plan to release one) with exact hyperparameters and preprocessing steps for the full benchmark suite? Since the empirical claim is broad (“consistently outperforms”), having a clean reference implementation would materially strengthen the case for acceptance, especially for a fast-moving DP empirical literature.

**Limitations:**

Yes.

**Strengths And Weaknesses:**

# Soundness
The work is technically careful in how it constructs privacy: (i) Gaussian sketching mechanisms are used in a way that leverages known privacy analyses for Gaussian sketches, and (ii) gradient-like terms are privatized via clipping to control sensitivity followed by Gaussian noise addition, a standard recipe for the Gaussian mechanism and its calibrations.

A particularly strong aspect is that the utility analysis matches the algorithmic structure: by aligning IHM with the IHS-style contraction behavior (geometric reduction of an initialization-dependent term), the bound captures an explicit “optimization-style” decay term plus a privacy/statistical noise term. This mirrors what makes IHS attractive in non-private sketching contexts.

The soundness risks are mostly about assumptions and tuning rather than obvious proof gaps:

 - The analysis (like much DP-OLS work) leans on boundedness assumptions and a particular neighboring relation (zero-out style). This is reasonable and common in this line of work but does narrow direct applicability.

 - Several conditions in the theorems involve quantities tied to the unknown optimum (e.g., controlling clipping so it does not trigger with high probability). While the experiments suggest the defaults work well, the “gap” between theory-driven parameter constraints and practice-driven fixed hyperparameters is an area where more explicit guidance could improve confidence in general deployment settings.

# Presentation
The paper is well structured: it motivates the problem and prior dominant baselines (e.g., adaptive sufficient-statistics perturbation) and positions Gaussian sketching as a widely used tool that has been less common in DP-OLS.

The algorithmic description is concrete and theorems are clearly stated early; the appendices provide support for privacy lemmas and theorems, and the experimental details are separated in a way that should aid verification.

Presentation weaknesses are mostly about reader ergonomics:
 - There are multiple “gamma/parameter” quantities and regime conditions; the paper would benefit from a slightly more didactic “one-page guide” that directly maps (problem regime → suggested $T,k,C$ → expected behavior), beyond what is already in the theorem statements and experimental defaults.
 - Because the work makes comparative claims (“circumvents intrinsic limitations” of sketching baselines), it would help to include a short, explicit “when do we expect improvements?” section in the main body (even if it is a condensed version of what is currently elaborated in appendices).

# Significance
DP-OLS is a classical and practically relevant primitive—used both for prediction and for explanatory/statistical purposes (e.g., estimating effects under privacy constraints).

The proposed method is significant because it offers a new, competitive DP-OLS design pattern: iterate with sketched Hessian information under privacy, rather than releasing privatized sufficient statistics once (AdaSSP) or sketching both the design and outcomes. This seems broadly reusable for other private second-order / sketch-based methods and, at minimum, deepens our understanding of how sketching interacts with privacy and utility in regression.

The impact is likely specialized but meaningful: the work sits at the intersection of DP theory and practical private regression. If the reported empirical consistency across many benchmarks holds up in community reproduction, it should influence what practitioners choose as “default” DP-OLS baselines and inspire follow-on work in private sketching algorithms.

# Originality
On novelty: the algorithm is not merely a rebranding of known methods; it is a principled adaptation of IHS-style iteration to the private setting, with a careful decomposition into (a) a privacy-preserving Hessian sketch mechanism and (b) a privacy-preserving gradient term.

The theoretical contribution is also original in two ways:
 - It provides a new utility guarantee for the proposed IHM algorithm with the explicit geometric-decay term, and
 - It offers a new analysis of prior Gaussian sketch DP-OLS methods, clarifying an inherent limitation in their dependence on residual quantities and eigen-structure (as framed by the paper).

Overall, the originality seems to come from a creative recombination (iterative Hessian sketching + privacy-preserving Gaussian sketching + DP accounting) coupled with nontrivial new analysis and extensive experiments.

---

> ### Author Rebuttal · Authors · 2026-03-30
>
> We thank the reviewer for the careful review and insightful suggestions.
>
> **Choosing Hyperparameters.**
>
> The choices of $k$ and $T$ were guided by principles similar to those in the classical sketching literature [1]. Our simulations suggest that changing them by constant factors has only a minor effect on performance, and the values used in the paper were selected by numerical tuning over the evaluated datasets (Appendix H). Two important edge cases for calibrating $T$ are the low-SNR regime and the regime where $\\lambda_{\\min}^X \\gg 1$, as clarified in Appendices L and K, where fewer or more iterations may improve performance. We will add these guidelines to Section 3.1. The parameter $\\mathsf{C}$ was set to the magnitude bound on $Y$, $\\mathsf{C}_Y$. Although the no-clipping guarantees suggest choosing $\\mathsf{C}=a_0\\cdot\\mathsf{C}_Y$ for some $a_0>1$, our empirical study shows that a small amount of clipping is tolerable, and that choosing $\\mathsf{C}_Y$ suffices for close-to-optimal performance on both the tested and synthetic stress-test datasets.
>
> **Improvement Regime.**
>
> Theorem 1 shows that IHM improves over AdaSSP whenever (a) $T$ is finite and small, and (b) $g^X \\leq \\max\\{d,\\log(T/\\varrho)\\}$. Step 2 of Appendix C.1.2 suggests that satisfying $\\frac{\\chi\\sqrt{\\kappa^X(\\eta^2)}}{2-\\chi}<1$ requires $k \\geq c g^X$. Empirically, this is automatically satisfied under our choice of $k$ (see Table 2, last column), so we expect the same theoretical improvement on these datasets, consistent with our results. We use the dataset collection of [2], which is reasonably diverse and representative of practical settings. Appendix K also shows that our method outperforms AdaSSP even when the theoretical assumptions are violated, suggesting that its scope is broader than our current theory predicts. As suggested by the reviewer, we will revise Sections 3.1 and 4.3 to give a more intuitive explanation of the improvement regime and clearer guidance for selecting the hyperparameters.
>
> **Computational Complexity.**
>
> Our algorithm is typically about $T$ times more expensive than AdaSSP due to its iterative structure. Since in practice $T$ can be chosen as a small constant, this overhead is usually negligible. AdaSSP solves one regression problem of size $n \\times d$, yielding complexity $O(nd^2)$. In contrast, IHM performs $T$ Gaussian sketching steps with a $k \\times n$ matrix, each followed by solving a $k \\times d$ regression problem, for a total complexity of $O\\left(T(kd^2+knd)\\right)$. Substituting $k=\\Theta\\left(\\max\\{d,\\log(1/\\varrho)\\}\\right)$ gives $O\\left(T\\max\\{d,\\log(1/\\varrho)\\}nd\\right)$, i.e., roughly a factor of $T$ over AdaSSP. Since $T$ is small in practice ($T=3$ in our experiments), this overhead does not appear to preclude deployment. Moreover, Appendix C.1.2 (Lines 1071--1077) suggests that the optimal $T$ grows at most as $T=\\Theta\\left(\\log\\left(\\frac{(\\epsilon\\lambda_{\\min}^X)^2}{\\max\\{d,\\log(1/\\varrho),g^X\\}\\log(1/\\delta)}\\right)\\right)$, which may remain small in many practical settings. Improving this complexity by incorporating faster sketches is an important direction for future work. Our main goal in this work was to provide a near-optimal algorithm that is practically implementable.
>
> **Clipping Behavior and Accuracy.**
>
> Our no-clipping guarantees are part of an asymptotic accuracy analysis and omit several constant factors. In practice, choosing $\\mathsf{C}=\\mathsf{C}_Y$ leads to a negligible, though nonzero, number of clipping events while retaining the performance advantage over AdaSSP. This suggests that the practical performance is stronger than our asymptotic analysis currently captures, and that a more refined non-asymptotic analysis could yield stronger guarantees. For the eight datasets simulated in the main text, the fraction of clipped coordinates divided by $nT$ for $T=3$ and at $\\epsilon=2.5$ is given below.
>
> | Dataset | Crime | Pol | Concreteslump | Fertility | Servo | Pendulum | Parkinsons | Protein |
> |---|---:|---:|---:|---:|---:|---:|---:|---:|
> | | $5.58 \\cdot 10^{-5}$ | $2.46 \\cdot 10^{-4}$ | $1.99 \\cdot 10^{-3}$ | $3.33 \\cdot 10^{-3}$ | $5.96 \\cdot 10^{-4}$ | $7.17 \\cdot 10^{-4}$ | $0$ | $1.21 \\cdot 10^{-3}$ |
>
> **Practicality of Assumptions.**
>
> As discussed throughout, our analysis assumes bounded covariates and responses, which restricts applicability when such bounds are unavailable. However, existing approaches that relax these assumptions (e.g., [3]) remain, to our knowledge, impractical, while our method performs well on several practical benchmarks and can serve as a drop-in replacement for DP-OLS practitioners.
>
> **Open-Source Implementation.**
>
> Our current code is included in the supplementary material for this submission, and we plan to release the full codebase as open source upon publication.
>
> [1] Pilanci and Wainwright, JMLR 2016.
>
> [2] Wang, UAI 2018.
>
> [3] Brown et al., COLT 2024.

---

> > ### Author Rebuttal · Reviewer_poR3 · 2026-04-04
> >
> > I thank the author for the thorough explanations to my concerns in the detailed rebuttal, and it basically solves my concerns. I will keep my positive score.

---

### Official Review · Reviewer_PM6M · 2026-03-13

**Soundness:** 4
**Presentation:** 4
**Significance:** 3
**Originality:** 3
**Overall Recommendation:** 5
**Confidence:** 3

**Summary:**

This paper studies the problem of differentially private ordinary least squares in the fixed-design, bounded-data setting where $x_i$ and $y_i$ are known to be bounded. The authors propose Iterative Hessian Mixing (IHM), an algorithm inspired by the non-private Iterative Hessian Sketch (IHS) of Pilanci & Wainwright (2016), adapted to satisfy ($\epsilon$, $\delta$)-differential privacy. The authors interpret the IHM algorithm as applying IHS to an augmented matrix $[X;\eta I]$, which enables them to leverage existing IHS contraction guarantees. They provide privacy guarantees and high-probability excess empirical risk bounds. The main theoretical claim is an improvement over AdaSSP (Wang, 2018), the leading sufficient-statistics perturbation method, with a dimension-dependent improvement that can be as large as $\sqrt{d}$, along with improved dependence on the failure probability.

**Compliance With Llm Reviewing Policy:**

Affirmed.

**Final Justification:**

This paper proposes a differentially private linear regression method based on iterative Hessian sketching, with accompanying theoretical guarantees and empirical evaluation. The work is technically well grounded, building on established ideas from sketching-based optimization and differential privacy, and the analysis appears sound. The empirical results demonstrate consistent improvements over strong baselines such as AdaSSP.

In terms of originality, the integration of iterative Hessian sketching into the differentially private setting is interesting and nontrivial, although it can be viewed as a natural extension of existing techniques. The contribution is meaningful in that it provides a near-optimal construction with practical performance gains under differential privacy constraints.

The rebuttal addressed my main concerns satisfactorily. In particular, the authors provided a clear comparison of computational complexity with AdaSSP and included additional experimental results for AdaOPS, which strengthen the empirical evaluation and clarify the positioning of the method. These clarifications improve the overall presentation and reinforce my assessment of the paper’s validity and practicality.

The discussion of the sketch dimension $k$ could be further elaborated in the final version. For example, by providing additional intuition or empirical sensitivity analysis. But I view this as a minor point rather than a substantive limitation.

Overall, the rebuttal improves clarity and confirms my initial assessment. I consider the paper to be technically sound with solid empirical support and a meaningful contribution to differentially private regression.

**Key Questions For Authors:**

1. Wang (2018) proposes both AdaSSP and AdaOPS. Could the authors clarify whether AdaOPS applies in the fixed-design bounded-data setting considered here? If so, how does IHM compare theoretically or empirically?
2. IHM involves iterative sketching steps across multiple iterations, whereas AdaSSP is a one-shot sufficient-statistics method. Could the authors provide additional discussion of the computational cost of IHM relative to AdaSSP, particularly in terms of scaling with $n$ and $d$? It is especially concerning when $d$ is relatively large because the sketching dimension $k$ depends on $d$. Even a brief comparison or runtime comment would help clarify the practical tradeoffs between the methods.

**Limitations:**

Yes. The paper discusses the main assumptions underlying the theoretical guarantees, such as data normalization and conditions on the design matrix. The discussion appropriately acknowledges the limitations of the analysis and the contexts in which the guarantees apply.

**Strengths And Weaknesses:**

Strengths

1. The reinterpretation of the IHM algorithm as IHS applied to an augmented matrix is elegant and technically effective. The analysis carefully controls the impact of mixing noise, gradient noise, and clipping.

2. The main theoretical contribution is a refinement of excess empirical risk bounds, improving upon AdaSSP in certain regimes by a factor that can scale as $\sqrt{d}$.

3. The work contributes conceptual clarity regarding when sketch-based approaches can compete with sufficient-statistics perturbation methods. The discussion of failure probability dependence and the role of Gaussian sketching is useful.

Weaknesses

1. Appendix E discusses optimizing the sketch dimension $k$, so the theoretically optimal choice depends on data-dependent quantities and thus would require additional privacy budget. But the authors adopted a fixed heuristic choice of $k$ in practice. A clearer empirical discussion on the method's performance using the optimal choice of $k$ and the fixed choice of $k$ would further strengthen the practical evaluation.

---

> ### Author Rebuttal · Authors · 2026-03-30
>
> We thank the reviewer for the positive review and thoughtful questions.
>
> **Optimizing the Sketching Dimension.**
>
> Appendix E discusses the Gaussian Sketch-and-Solve method (Algorithm 5). For IHM, however, the lower bound on $k$ comes from two sources: classical sketching requirements and clipping control (Section 3.1), yielding $k \\geq \\Omega\\left(\\max\\{d,\\log(T/\\varrho),g^X\\}\\right)$. Empirically, Table 2 (last column) shows that $\\frac{\\chi\\sqrt{\\kappa(\\eta^2)}}{2-\\chi}<1$ throughout, which is sufficient for the asymptotic no-clipping condition in Theorem 1. We therefore omit this term in practice and set $k=6\\cdot \\max\\{d,\\log(T/\\varrho)\\}$, which is data-independent, and where the constant $6$ was picked based on calibration done in [1].
>
> For Algorithm 5 in Theorem 2, the optimized $k$ depends on $\\theta^{\star}$ and $L(\\theta^{\star})$. These are usually difficult to estimate privately, since the sensitivity of $\\theta^{\star}$ depends on $(X^\\top X)^{-1}$ and may be large. One can instead control local sensitivity, but this requires privatizing that quantity and then using it to calibrate the noise added to $\\|\\theta^{\star}\\|$, which is typically expensive in privacy cost. Since the resulting asymptotic guarantee still faces limits similar to the classical sketch-and-solve bounds discussed in the paper, we did not include this calibration in our experiments.
>
> **Comparing to AdaOPS.**
>
> AdaSSP is generally regarded as the stronger baseline among the two methods in [2], both empirically and theoretically in the fixed-design setting (see [3,4]), which is why we focused on it in the main paper. To strengthen the empirical comparison, we also evaluated AdaOPS. As expected, AdaOPS performs worse than AdaSSP, and IHM therefore also outperforms AdaOPS.
>
> | Dataset | AdaOPS | AdaSSP | LinMix | IHM |
> |---|---:|---:|---:|---:|
> | Crime | $1.53\\cdot10^{-1}\\pm6.60\\cdot10^{-4}$ | $1.54\\cdot10^{-1}\\pm4.90\\cdot10^{-4}$ | $1.30\\cdot10^{-1}\\pm2.30\\cdot10^{-4}$ | $1.11\\cdot10^{-1}\\pm1.00\\cdot10^{-4}$ |
> | Pol | $1.22\\cdot10^{-1}\\pm2.76\\cdot10^{-4}$ | $1.16\\cdot10^{-1}\\pm3.57\\cdot10^{-4}$ | $1.31\\cdot10^{-1}\\pm6.16\\cdot10^{-3}$ | $4.51\\cdot10^{-2}\\pm8.77\\cdot10^{-4}$ |
> | Concreteslump | $1.44\\cdot10^{-1}\\pm1.69\\cdot10^{-3}$ | $1.40\\cdot10^{-1}\\pm1.88\\cdot10^{-3}$ | $1.53\\cdot10^{-1}\\pm5.73\\cdot10^{-3}$ | $1.31\\cdot10^{-1}\\pm2.37\\cdot10^{-3}$ |
> | Fertility | $2.51\\cdot10^{-2}\\pm9.41\\cdot10^{-4}$ | $2.43\\cdot10^{-2}\\pm9.78\\cdot10^{-4}$ | $4.15\\cdot10^{-2}\\pm3.74\\cdot10^{-3}$ | $2.45\\cdot10^{-2}\\pm1.72\\cdot10^{-3}$ |
> | Servo | $9.46\\cdot10^{-2}\\pm2.92\\cdot10^{-3}$ | $7.13\\cdot10^{-2}\\pm2.43\\cdot10^{-3}$ | $8.17\\cdot10^{-2}\\pm5.46\\cdot10^{-3}$ | $6.06\\cdot10^{-2}\\pm2.07\\cdot10^{-3}$ |
> | Pendulum | $5.69\\cdot10^{-3}\\pm2.82\\cdot10^{-4}$ | $5.11\\cdot10^{-3}\\pm2.59\\cdot10^{-4}$ | $1.03\\cdot10^{-2}\\pm1.02\\cdot10^{-3}$ | $4.37\\cdot10^{-3}\\pm3.17\\cdot10^{-4}$ |
> | Parkinsons | $1.76\\cdot10^{-2}\\pm2.10\\cdot10^{-4}$ | $1.59\\cdot10^{-2}\\pm1.01\\cdot10^{-4}$ | $1.56\\cdot10^{-2}\\pm5.28\\cdot10^{-4}$ | $1.22\\cdot10^{-2}\\pm1.28\\cdot10^{-4}$ |
> | Protein | $4.14\\cdot10^{-2}\\pm1.06\\cdot10^{-4}$ | $3.85\\cdot10^{-2}\\pm1.21\\cdot10^{-4}$ | $3.80\\cdot10^{-2}\\pm1.40\\cdot10^{-3}$ | $3.06\\cdot10^{-2}\\pm1.60\\cdot10^{-4}$ |
>
> **Computational Complexity.**
>
> Our algorithm is typically about $T$ times more expensive than AdaSSP due to its iterative structure. Since $T$ can usually be chosen as a small constant, this overhead is modest in practice. AdaSSP solves one regression problem of size $n\\times d$, yielding complexity $O(nd^2)$. In contrast, IHM performs $T$ Gaussian sketching steps with a $k\\times n$ matrix, each followed by solving a $k\\times d$ regression problem, for total complexity $O\\left(T\\cdot(kd^2+knd)\\right)$. Substituting $k=\\Theta\\left(\\max\\{d,\\log(1/\\varrho)\\}\\right)$ gives $O\\left(T\\cdot\\max\\{d,\\log(1/\\varrho)\\}\\cdot nd\\right)$, i.e., roughly a factor of $T$ over AdaSSP. In our experiments, $T=3$. Appendix C.1.2 (Lines 1071--1077) also suggests that the optimal $T$ grows at most as $T=\\Theta\\left(\\log\\left(\\frac{(\\epsilon\\lambda_{\\min}^X)^2}{\\max\\{d,\\log(1/\\varrho),g^X\\}\\log(1/\\delta)}\\right)\\right)$, which may remain small in practical settings. Extending the method to faster sketches is an important direction for future work. Our main goal is a near-optimal construction that is both practically implementable and provably achieves $(\\varepsilon,\\delta)$-DP.
>
> [1] Pilanci & Wainwright, JMLR 2016.
>
> [2] Wang, UAI 2018.
>
> [3] Brown et al., ICML 2024.
>
> [4] Liu, Kong, and Oh, COLT 2022.

---

> > ### Author Rebuttal · Reviewer_PM6M · 2026-04-02
> >
> > The rebuttal provides helpful clarifications and addresses several of my concerns. In particular, the authors give a clear comparison of computational complexity between IHM and AdaSSP, showing that the additional overhead is modest when the number of iterations $T$ is small. The inclusion of additional experimental results for AdaOPS also strengthens the empirical evaluation and clarifies its relative performance.
> >
> > My main remaining concern relates to the choice of the sketch dimension $k$. While the authors explain that the theoretically optimal, data-dependent choice of $k$ is difficult to implement under differential privacy constraints, the practical use of a fixed heuristic $k$ remains only qualitatively justified. It would strengthen the paper to provide more explicit insight into how close this heuristic is to the optimal choice, or to include empirical sensitivity analysis demonstrating robustness to different values of $k$.
> >
> > Overall, the rebuttal improves clarity and addresses most of the practical concerns, but the theoretical–practical tradeoff in choosing $k$ remains only partially clarified.

---

> > > ### Author Response · Authors · 2026-04-02
> > >
> > > Our constant choice of $k$ for LinMix was made so that the leading term in the guarantee improves over that of AdaSSP. As explained in Section 4.3, this choice yields $\\gamma_{\\mathrm{m}}<\\gamma_{\\mathrm{a}}$ and avoids the $g^{X}$ term appearing in $\\gamma_{\\mathrm{h}}$. Moreover, even under the optimized data-dependent choice of $k$ from Appendix E, the leading multiplier does not improve asymptotically. In particular, in the favorable regime for LinMix, namely $L(\\theta^{*})=O(\\gamma_{\\mathrm{m}}C_{\\mathrm{m}}^{2})$ and $\\lambda_{\\min}^{X}\\ll 1$, the asymptotic guarantee remains unchanged. We will clarify this point more explicitly. Appendix E concerns only the LinMix scheme, and is included to support our claim that these limitations are intrinsic to this approach, even with an optimized choice of $k$.
> > >
> > > For IHM, the constant choice of $k$ was instead guided by the constraints arising in the proof of Theorem 1. We used this simplified calibration because, as shown in Table 2, it already implies the required no-clipping condition in the simulated regime. We agree that this rationale should be stated more clearly and will add it in the revision.
> > >
> > > We thank the reviewer for raising this point.

---

### Decision · Program_Chairs · 2026-04-30

**Decision:**

Accept (spotlight)

**Comment:**

This is a strong paper on differentially private ordinary least squares. Reviewers found the Iterative Hessian Mixing idea technically well motivated, liked the reinterpretation through iterative Hessian sketching, and found the resulting theory meaningful both as a new guarantee for IHM and as a clarification of the limitations of prior Gaussian-sketching approaches. The empirical evaluation is also convincing, and supports the paper’s claim that IHM is competitive with strong baselines such as AdaSSP.

The main remaining concerns were about practical guidance for choosing hyperparameters, runtime relative to simpler baselines, and the paper’s scope on the bounded, overdetermined OLS regime. The rebuttal clarified computational overhead, added comparison to AdaOPS, and discussed clipping and hyperparameter choices. I think that these are good pointers for things to discuss in the final version, but do not remove from the contribution itself.

I therefore recommend acceptance.